# Regional variation in the effectiveness of methane-based and land-based climate mitigation options

Garry D. Hayman[1,*], Edward Comyn-Platt[1], Chris Huntingford[1], Anna B. Harper[2], Tom Powell[2], Peter M. Cox[2], William Collins[3], Christopher Webber[3], Jason Lowe[4,5], Stephen Sitch[2], Joanna I. House[6], Jonathan C. Doelman[7], Detlef P. van Vuuren[7,8], Sarah E. Chadburn[2], Eleanor Burke[5], Nicola Gedney[9].

[1] Centre for Ecology & Hydrology, Wallingford, OX10 8BB, U.K.

[2] University of Exeter, Exeter, EX4 4QF, U.K.

[3] University of Reading, Reading, RG6 6BB, U.K.

[4] University of Leeds, Leeds, LS2 9JT, U.K.

[5] Met Office Hadley Centre, FitzRoy Road, Exeter, EX1 3PB, U.K.

[6] Cabot Institute for the Environment, University of Bristol, Bristol, BS8 1SS, U.K.

[7] Department of Climate, Air and Energy, Netherlands Environmental Assessment Agency (PBL), PO Box 30314, 2500 GH The Hague, Netherlands

[8] Copernicus Institute of Sustainable Development, Utrecht University, Heidelberglaan 2, 3584 CS, the Netherlands

[9] Met Office Hadley Centre, Joint Centre for Hydrometeorological Research, Wallingford, OX10 8BB, U.K.

*Correspondence to*: Garry Hayman (garr@ceh.ac.uk)

**Abstract.** Scenarios avoiding global warming greater than 1.5 or 2°C, as stipulated in the Paris Agreement, may require the combined mitigation of anthropogenic greenhouse gas emissions alongside enhancing negative emissions through approaches such as afforestation/reforestation (AR) and biomass energy with carbon capture and storage (BECCS). We use the JULES land-surface model coupled to an inverted form of the IMOGEN climate emulator to investigate mitigation scenarios that achieve the 1.5 or 2°C warming targets of the Paris Agreement. Specifically, within this IMOGEN-JULES framework, we focus on and characterise the global and regional effectiveness of land-based (BECCS and/or AR) and anthropogenic methane ($CH_4$) emission mitigation, separately and in combination, on the anthropogenic fossil fuel carbon dioxide ($CO_2$) emission budgets (AFFEBs) to 2100. We use consistent data and socio-economic assumptions from the IMAGE integrated assessment model for the second Shared Socioeconomic Pathway (SSP2). The analysis includes the effects of the methane and carbon-climate feedbacks from wetlands and permafrost thaw, which we have shown previously to be significant constraints on the AFFEBs.

Globally, mitigation of anthropogenic $CH_4$ emissions has large impacts on the anthropogenic fossil fuel emission budgets, potentially offsetting (i.e. allowing extra) carbon dioxide emissions of 188-212 GtC. This is because of (a) the reduction in the direct and indirect radiative forcing of methane in response to the lower emissions and hence atmospheric concentration of methane; and (b) carbon-cycle changes leading to increased uptake by the land and ocean by $CO_2$-based fertilisation. Methane mitigation is beneficial everywhere, particularly for the major $CH_4$-emitting regions of India, USA and China. Land-based mitigation has the potential to offset 51-100 GtC globally, the large range reflecting assumptions and uncertainties associated with BECCS. The ranges for $CH_4$ reduction and BECCs implementation are valid for both the 1.5° and 2°C warming targets. That is the mitigation potential of the $CH_4$ and of the land-based scenarios is similar for whether society aims for one or other of the final stabilised warming levels. Further, both the effectiveness and the preferred land-management strategy (i.e., AR or BECCS) have strong regional dependencies. Additional analysis shows extensive BECCS could adversely affect water security for several regions. Although the primary requirement remains mitigation of fossil fuel emissions, our results highlight the potential for the mitigation of $CH_4$ emissions to make the Paris climate targets more achievable.

## 1 Introduction

The stated aims of the Paris Agreement of the United Nations Framework Convention on Climate Change (UNFCCC, 2015) are "to hold the increase in global average temperature to well below 2°C and to pursue efforts to limit the increase to 1.5°C". The global average surface temperature for the decade 2006-2015 was 0.87°C above pre-industrial levels and is likely to reach 1.5°C between the years 2030 and 2052, if global warming continues at current rates (IPCC, 2018). The IPCC Special Report on Global Warming of 1.5°C (IPCC, 2018) gives the median remaining carbon budgets between 2018 and 2100 as 770 $GtCO_2$ (210 GtC) and 1690 $GtCO_2$ (~461 GtC) to limit global warming to 1.5°C and 2°C, respectively. These budgets represent ~20 and ~41 years at present-day emission rates. The actual budgets could however be smaller, as they exclude Earth system feedbacks such as $CO_2$ released by permafrost thaw or $CH_4$ released by wetlands. Meeting the Paris Agreement goals will, therefore, require sustained reductions in sources of fossil carbon emissions, other long-lived anthropogenic greenhouse gases (GHGs) and some short-lived climate forcers (SLCFs) such as methane ($CH_4$), alongside increasingly extensive implementations of carbon dioxide removal (CDR) technologies (IPCC, 2018). Accurate information is needed on the range and efficacy of options available to achieve this.

Biomass energy with carbon capture and storage (BECCS) and afforestation/reforestation (AR) are among the most widely considered CDR technologies in the climate and energy literature (Minx et al., 2018) . For scenarios consistent with a 2°C warming target, the review by Smith et al. (2016) finds this may require (1) a median removal of 3.3 GtC $yr^{-1}$ from the atmosphere through BECCS by 2100 and (2) a mean CDR through AR of 1.1 GtC $yr^{-1}$ by 2100, giving a total CDR equivalent to 47% of present-day emissions from fossil fuel and other industrial sources (Le Quéré et al., 2018). Although there are fewer scenarios that look specifically at the 1.5°C pathway, BECCS is still the major CDR approach (Rogelj et al., 2018). For the default assumptions in Fuss et al. (2018), BECCS would remove a median of 4 GtC $yr^{-1}$ by 2100 and a total of 41-327 GtC from the atmosphere during the twenty-first century, equivalent to about 4-30 years of current annual emissions. The land requirements for BECCS will be greater for the 1.5°C target within a given shared socio-economic pathway (e.g., SSP2), although published estimates are similar for the two warming targets, with between 380-700 Mha required for the 2°C target (Smith et al., 2016) and greater than 600 Mha for the 1.5°C target (van Vuuren et al., 2018). This is because the land requirements for bioenergy production differ strongly across the different SSPs, depending on assumptions about the contribution of residues, assumed yields and yield improvement, start dates of implementation and the rates of deployment. While the CDR figures assume optimism about the mitigation potential of BECCS, concerns have been raised about the potentially detrimental impacts of BECCS on food production, water availability and biodiversity, e.g., (Heck et al., 2018; Krause et al., 2017). Others note the risks and query the feasibility of large-scale deployment of BECCS e.g. (Anderson and Peters, 2016; Vaughan and Gough, 2016; Vaughan et al., 2018).

Harper et al. (2018) find the overall effectiveness of BECCS to be strongly dependent on the assumptions concerning yields, the use of initial above-ground biomass that is replaced and the calculated fossil-fuel emissions that are offset in the energy system. Notably, if BECCS involves replacing ecosystems that have higher carbon contents than energy crops, then AR and avoided deforestation can be more efficient than BECCS for atmospheric $CO_2$ removal over this century (Harper et al., 2018).

Mitigation of the anthropogenic emissions of non-$CO_2$ GHGs such as $CH_4$ and of SLCFs such as black carbon have been shown to be attractive strategies with the potential to reduce projected global mean warming by 0.22-0.5°C by 2050 (Shindell et al., 2012; Stohl et al., 2015). It should be noted that these were based on scenarios with continued use of fossil fuels. Through the link to tropospheric ozone ($O_3$), there are additional co-benefits of $CH_4$ mitigation for air quality, plant productivity and food production (Shindell et al., 2012) and carbon sequestration (Oliver et al., 2018). Control of anthropogenic $CH_4$ emissions

leads to rapid decreases in its atmospheric concentration, with an approximately 9-year removal lifetime (and as such is an SLCF). Furthermore, many $CH_4$ mitigation options are inexpensive or even cost negative through the co-benefits achieved (Stohl et al., 2015), although expenditure becomes substantial at high levels of mitigation (Gernaat et al., 2015). The extra "allowable" carbon emissions from $CH_4$ mitigation can make a substantial difference to the feasibility or otherwise of achieving the Paris climate targets (Collins et al., 2018).

Some increases in atmospheric $CH_4$ are not related to direct anthropogenic activity, but indirectly to climate change triggering natural carbon and methane-climate feedbacks. These effects could act as positive feedbacks, and thus in the opposite direction to the mitigation of anthropogenic $CH_4$ sources. Wetlands are the largest natural source of $CH_4$ to the atmosphere and these emissions respond strongly to climate change (Gedney et al., 2019; Melton et al., 2013). A second natural feedback is from permafrost thaw. In a warming climate, the resulting microbial decomposition of previously frozen organic carbon is potentially one of the largest feedbacks from terrestrial ecosystems (Schuur et al., 2015). As the carbon and $CH_4$ climate feedbacks from natural wetlands and permafrost thaw could be substantial, this causes a reduction in anthropogenic $CO_2$ emission budgets compatible with climate change targets (Comyn-Platt et al., 2018a; Gasser et al., 2018).

This paper models the potential for mitigation of greenhouse gases to contribute to meeting the Paris targets of limiting global warming to 1.5°C and 2°C respectively. Specifically, we investigate the effectiveness of mitigation of anthropogenic methane emissions and land-based mitigation (e.g., implementation of BECCS and AR), combining results from three recent papers (Collins et al., 2018; Comyn-Platt et al., 2018a; Harper et al., 2018). We determine the effectiveness of these approaches in terms of their impact on the anthropogenic fossil fuel $CO_2$ emissions budget consistent with stabilising temperature at 1.5°C and 2° C of warming. The more effective the mitigation option, the larger the fossil fuel $CO_2$ emissions budget consistent with stabilisation at a given level. We estimate the impact of these mitigation scenarios relative to an existing scenario of greenhouse gas concentrations (based on the IMAGE SSP2 baseline), spanning uncertainties in both climate model projections (both global warming and regional climate change), process representation and the efficacy of BECCS. Sect. 2 provides a brief description of the models, the experimental set-up and the key datasets used in the model runs and subsequent analysis. Sect. 3 presents and discusses the results, starting with a global perspective before addressing the regional dimension. For BECCS, we additionally investigate the sensitivity to key assumptions and consider the implications for water security. Sect. 4 contains our conclusions.

## 2   Approach and Methodology

Our overall modelling strategy is as follows. The starting point is the prescription of global temperature profiles that match the historical record, followed by a transition to a future stabilisation at either 1.5 or 2.0°C above pre-industrial levels. For these profiles, we then determine the related pathways in atmospheric radiative forcing by inversion of the global energy balance component of the IMOGEN impacts model. IMOGEN "Integrated Model Of Global Effects of climatic aNomalies" (Sect. 2.2) (Comyn-Platt et al., 2018a; Huntingford et al., 2010) is an intermediate complexity climate model, which emulates 34 models in the CMIP5 climate model ensemble. Hence our radiative forcing (RF) trajectories have uncertainty bounds, reflecting the different climate sensitivities of existing climate models.

For each radiative forcing pathway, we subtract the individual RF components for non-$CO_2$ and non-$CH_4$ radiatively-active gases that are perturbed by human activity, using baseline and mitigation scenarios taken from the IMAGE integrated assessment model. Then, for $CH_4$, we represent its atmospheric chemistry by a single atmospheric lifetime to translate the methane emissions into atmospheric concentrations. The related RF for $CH_4$ is also subtracted from the overall value. Hence

the remaining RF is that available for changes to atmospheric $CO_2$ concentration. The IMOGEN model uses pattern-scaling, again fitted to the same 34 climate models, to estimate local changes in near-surface meteorology. Combined with our global temperature pathways, these pattern-based changes (as well as atmospheric $CO_2$ concentration) drive the Joint UK Land-Environment Simulator land surface model (JULES, Sect. 2.1) (Best et al., 2011; Clark et al., 2011). JULES estimates atmosphere-land $CO_2$ exchange, and similarly, IMOGEN contains a single global description of oceanic $CO_2$ draw-down. These two estimates of carbon exchanges with the land and ocean respectively, in conjunction with atmospheric storage being linear in the $CO_2$ pathway, finally determine by simple summation compatible $CO_2$ emissions from fossil fuel burning. We call this the anthropogenic ($CO_2$) fossil fuel emission budgets (AFFEB) compatible with the warming pathway, subject to the assumptions made for non-$CO_2$ forcings.

Our numerical simulation structure allows us to investigate the implications of three different key changes on AFFEB, for stabilisation at both 1.5 and 2.0°C, and in a structure that captures features of a full set of climate models. First and maybe most importantly, we work to understand how regional reductions in $CH_4$ emissions allow higher values of AFFEB. Second, we consider how alternative scenarios of BECCS implementation alter atmosphere-land $CO_2$ exchanges, and again presented as the resultant implications for AFFEB. Third, we determine how the newer understanding of warming impacts on wetland methane emissions also affects AFFEB. Figure 1 captures the modelling framework, derivation of AFFEB, and our numerical experiments in a single overall schematic diagram.

Each of the scenarios investigated using the IMOGEN-JULES framework comprises 2 ensembles of 136 members, one ensemble for each of the warming targets. We make use of these ensembles to derive an "uncertainty" in the derived carbon budgets, specifically from climate change (as given by the 34 CMIP5 models) and from key land-surface processes (methane emissions from wetlands and the ozone vegetation damage). The climate change uncertainty comprises both the range of climate sensitivities of the CMIP5 models and the different regional patterns in the models. We use the median of the 136-member ensemble as the central value to derive the carbon budgets and the interquartile range (25-75%) for the uncertainty.

## 2.1 The JULES model

We use the JULES land surface model (Best et al., 2011; Clark et al., 2011), release version 4.8, but with a number of additions required specifically for our analysis:

1. Land use: We adopt the approach used by Harper et al. (2018) and prescribe *managed* land-use and land-use change (LULUC). On land used for agriculture, C3 and C4 grasses are allowed to grow to represent crops and pasture. The land-use mask consists of an annual fraction of agricultural land in each grid cell. Historical LULUC is based on the HYDE 3.1 dataset (Klein Goldewijk et al., 2011), and future LULUC is based on two scenarios (SSP2 RCP-1.9 and SSP2 baseline), which were developed for use in the IMAGE integrated assessment model (IAM) (Doelman et al., 2018; van Vuuren et al., 2017) (see also Sect. 2.3).

   Natural vegetation is represented by nine plant functional types (PFTs): broadleaf deciduous trees, tropical broadleaf evergreen trees, temperate broadleaf evergreen trees, needle-leaf deciduous trees, needle-leaf evergreen trees, C3 and C4 grasses, deciduous and evergreen shrubs (Harper et al., 2016). These PFTs are in competition for space in the non-agricultural fraction of grid cells, based on the TRIFFID (Top-down Representation of Interactive Foliage and Flora Including Dynamics) dynamic vegetation module within JULES (Clark et al., 2011). A further four PFTs are used to represent agriculture (C3 and C4 crops, and C3 and C4 pasture), and harvest is calculated separately for food and bioenergy crops (see Sect. 2.4.3, where we describe the modelling of carbon removed via bioenergy with CCS). When natural vegetation is converted to managed agricultural land, the vegetation carbon removed is placed into woody

product pools that decay at various rates back into the atmosphere (Jones et al., 2011). Hence, the carbon flux from
LULUC is not lost from the system. There are also four non-vegetated surface types: urban, water, bare soil and ice.
2. Soil carbon: Following Comyn-Platt et al. (2018b), we also use a 14 layered soil column for both hydro-thermal
(Chadburn et al., 2015) and carbon dynamics (Burke et al., 2017b). Burke et al. (2017a) demonstrated that modelling
the soil carbon fluxes as a multi-layered scheme improves estimates of soil carbon stocks and net ecosystem exchange.
In addition to the vertically discretised respiration and litter input terms, the soil-carbon balance calculation also
includes a diffusivity term to represent cryoturbation/bioturbation processes. The freeze-thaw process of
cryoturbation is particularly important in cold permafrost-type soils (Burke et al., 2017a). Following Burke et al.
(2017b), we diagnose permafrost wherever the deepest soil layer is below 0°C (assuming that this layer is below the
depth of zero annual amplitude, i.e., where seasonal changes in ground temperature are negligible ($\leq$0.1 °C)). Further,
for permafrost regions, there is an additional variable to trace or diagnose "old" carbon and its release from permafrost
as it thaws.
The multi-layered methanogenesis scheme improves the representation of high latitude $CH_4$ emissions, where
previous studies underestimated production at cold permafrost sites during "shoulder seasons" (Zona et al., 2016).
Figure 2 shows the annual cycle in the observed and modelled wetland $CH_4$ emissions at the Samoylov Island field
site (panel a) and a comparison of observed and modelled annual mean fluxes at this and other sites (panel b). The
range of uncertainty used in our study (JULES low $Q_{10}$ - JULES high $Q_{10}$) captures the range of uncertainty in the
observations (In Fig. 2b, the error bars denote the lower and upper estimates from the low and high $Q_{10}$ simulations.
The symbols represent the mean value between these estimates). Further, the layered methane scheme used in this
work gives a better description of the shoulder season emissions when compared with the original, non-layered
methane scheme in JULES. The multi-layered scheme allows an insulated sub-surface layer of active methanogenesis
to continue after the surface has frozen. These model developments not only improve the seasonality of the emissions,
but more importantly for this study capture the release of carbon as $CH_4$ from deep soil layers, including thawed
permafrost. Further evaluation of the multi-layer scheme can be found in Chadburn et al. (2020).
3. Methane from wetlands: Following Comyn-Platt et al. (2018b), we also use the multi-layered soil carbon scheme
described in (2) above to give the local land-atmosphere $CH_4$ flux, $E_{CH4}$ (kg C m$^{-2}$ s$^{-1}$):
$$E_{CH4} = k \cdot f_{wetl} \cdot \sum_{i=1}^{n\,C_s\,pools} \kappa_i \cdot \sum_{z=0m}^{z=3m} e^{-\gamma z} C_{s_{i,z}} \cdot Q_{10}\left(T_{soil_z}\right)^{0.1\left(T_{soil_z}-T_0\right)} \qquad (1)$$
where $k$ is a dimensionless scaling constant such that the global annual wetland $CH_4$ emissions are 180 Tg $CH_4$ in
2000 (as described in Comyn-Platt et al. (2018b)), $z$ is the depth in soil column (in m), $i$ is the soil carbon pool, $f_{wetl}$
(-) is the fraction of wetland area in the grid cell, $\kappa_i$ (s$^{-1}$) is the specific respiration rate of each pool (Table 8 of Clark
et al. (2011)), $C_s$ (kg m$^{-2}$) is soil carbon, $T_{soil}$ (K) is the soil temperature. The decay constant $\gamma$ (= 0.4 m$^{-1}$) describes
the reduced contribution of $CH_4$ emission at deeper soil layers due to inhibited transport and increased oxidation
through overlaying soil layers. This representation of inhibition and of the pathways for $CH_4$ release to the atmosphere
(e.g., by diffusion, ebullition and vascular transport) is a simplification. However, previous work which explicitly
represented these processes showed little to no improvement when compared with in-situ observations (McNorton et
al., 2016). We do not model $CH_4$ emissions from freshwater lakes (and oceans).
Comyn-Platt et al. (2018b) varied $Q_{10}$ in Eq. (1) to encapsulate a range of methanogenesis process uncertainty. They
derive $Q_{10}$ values for each GCM configuration to represent two wetland types identified in Turetsky et al. (2014)

('poor-fen' and 'rich-fen'). They also include a third 'low-$Q_{10}$', which gives increased importance to high latitude emissions. Their ensemble spread was able to describe the magnitude and distribution of present-day $CH_4$ emissions from natural wetlands, according to the models used in the then-current global methane assessment (Saunois et al., 2016). Here, we use the 'low-$Q_{10}$' value of Comyn-Platt et al. (2018b) (=2.0) and adopt a 'high-$Q_{10}$' value of ~4.8 from the rich-fen parameterisation. The two $Q_{10}$ values used here still capture the full range of the methanogenesis process uncertainty.

4. Ozone vegetation damage: We use a JULES configuration including ozone deposition damage to plant stomata, which affects land-atmosphere $CO_2$ exchange (Sitch et al., 2007). JULES requires surface atmospheric ozone concentrations, $O_3$ (ppb), for the duration of the simulation period (1850-2100). As in Collins et al. (2018), we do not model tropospheric ozone production from $CH_4$ explicitly in IMOGEN. Instead, we use two sets of monthly near-surface $O_3$ concentration fields (January-December) from HADGEM3-A GA4.0 model runs, with the sets corresponding to low (1285 ppbv) and high (2062 ppbv) global mean atmospheric $CH_4$ concentrations (Stohl et al., 2015). We assume that the atmospheric $O_3$ concentration in each grid cell responds linearly to the atmospheric $CH_4$ concentration. We derive separate linear relationships for each month and grid cell, and use these to calculate the surface $O_3$ concentration from the corresponding global atmospheric $CH_4$ concentration as it evolves during the IMOGEN run (Sect. 2.2.1). We use $CH_4$ concentration profiles from the IMAGE SSP2 Baseline and RCP-1.9 scenarios (Sect. 2.3.1), adjusted for natural methane sources (see 3 above and Sect. 2.3.3). We undertake runs using both the 'high' and 'low' vegetation ozone-damage parameter sets (Sitch et al., 2007).

**2.2 The IMOGEN intermediate complexity climate model**

**2.2.1 IMOGEN**

The IMOGEN climate impacts model (Huntingford et al., 2010) uses "pattern-scaling" to estimate changes to the seven meteorological variables required to drive JULES. Huntingford et al. (2010) assume that changes in local temperature, precipitation, humidity, wind-speed, surface shortwave and longwave radiation and pressure are linear in global warming. Spatial patterns of each variable (based on the 34 GCM simulations in CMIP5, Comyn-Platt et al. (2018b)) are multiplied by the amount of global warming over land, $\Delta T_L$, to give local monthly predictions of climate change. When using IMOGEN in forward mode, $\Delta T_L$ is calculated with an Energy Balance Model (EBM) as a function of the overall changes in radiative forcing, $\Delta Q$ (W m$^{-2}$). $\Delta Q$ is the sum of the atmospheric greenhouse gas contributions (Eq. (2)) (Etminan et al., 2016), which in the forward mode are either calculated ($CO_2$ and $CH_4$) or prescribed (for other atmospheric contributors) on a yearly time step.

$$\Delta Q(total) = \Delta Q(CO_2) + \Delta Q(non\ CO_2\ GHGs) + \Delta Q(aerosols\ and\ other\ climate\ forcers) \qquad (2)$$

The EBM includes a simple representation of the ocean uptake of heat and $CO_2$ and uses a separate set of four parameters for each climate and Earth system model emulated (Huntingford et al., 2010): the climate feedback parameters over land and ocean, $\lambda_l$ and $\lambda_o$ (W m$^{-2}$ K$^{-1}$) respectively, the oceanic "effective thermal diffusivity", $\kappa$ (W m$^{-1}$ K$^{-1}$) representing the ocean thermal inertia and a land-sea temperature contrast parameter, $\nu$, linearly relating warming over land, $\Delta T_l$ (K) to warming over ocean, $\Delta T_o$ (K), as $\Delta T_l = \nu \Delta T_o$. The climate feedback parameters ($\lambda_l$ and $\lambda_o$) are calibrated using model-specific data for the top of the atmosphere radiative fluxes, the mean land and ocean surface temperatures, along with an estimate of the radiative forcing modelled for the $CO_2$ changes.

The atmospheric CH$_4$ concentrations available from the IMAGE database (see Sect. 2.3.1) assume a constant annual
wetland CH$_4$ emission (van Vuuren et al., 2017). However, these emissions have interannual variability and a positive climate
feedback (Comyn-Platt et al., 2018a; Gedney et al., 2019), and their correct representation is a central part of our study. We
follow the same approach that we used in our previous studies (Collins et al., 2018; Comyn-Platt et al., 2018a; Gedney et al.,
2019). As the IMOGEN-JULES modelling framework does not have an explicit representation of the atmospheric chemistry
of methane, we represent the oxidation and hence loss of CH$_4$ by a single lifetime ($\tau$).
$$\frac{d([CH_4] - [CH_4]_{IMAGE})}{dt} = C \left\{ \sum F [CH_4] - \sum F [CH_4]_{IMAGE} \right\} - \frac{[CH_4] - [CH_4]_{REF}}{\tau} \qquad (3)$$
where [CH$_4$] and [CH$_4$]$_{IMAGE}$ are the atmospheric methane concentrations using our new wetland-based, time varying
($F$[CH$_4$]) and the constant IMAGE ($F$[CH$_4$]$_{IMAGE}$) wetland emissions, respectively. Parameter C is a constant to convert from
Tg CH$_4$ to a mixing ratio in parts per billion by volume (ppbv). Further, higher atmospheric concentrations of CH$_4$ and its
oxidation product (carbon monoxide) lower the concentration of hydroxyl radicals, the major removal reaction for CH$_4$, thereby
increasing the atmospheric lifetime of CH$_4$. Conversely, lower CH$_4$ concentrations will shorten its atmospheric lifetime. We
take account of this feedback of CH$_4$ on its lifetime ($\tau$), using Eq. (4) (Collins et al., 2018; Comyn-Platt et al., 2018a; Gedney
et al., 2019), as.
$\ln (\tau/\tau_o) = s.\ln ([CH_4]/[CH_4]_o)$, i.e., $\tau = \tau_o \exp (s [CH_4]/[CH_4]_o)$ \qquad (4)
In Eq. (4), $[CH_4]_o$ and $\tau_o$ are the contemporary atmospheric CH$_4$ concentration and lifetime, and s is the CH$_4$-OH feedback
factor, defined by $s = \partial \ln(\tau)/\partial \ln (CH_4)$. We take values of $\tau_o = 8.4$ years, $[CH_4]_o = 1{,}745$ ppbv and s = 0.28 from Ehhalt et al.
(2001) (pages 248 and 250). In our earlier study on the climate-wetland methane feedback (Gedney et al., 2019), we investigate
the sensitivity to the methane lifetime and the feedback factor, in addition to an analysis of the main drivers on the wetland
methane-climate feedback and the main sources of uncertainty. Gedney et al. (2019) conclude that the limited knowledge of
contemporary global wetland emissions is a larger source of uncertainty than that from the projected climate spread of the 34
GCMs. We quantify this uncertainty in our experimental design by using two values of Q$_{10}$ (see Sect. 2.1).
In response to our dynamic interactive calculations of atmospheric CH$_4$ concentrations, we derive the related change in
methane radiative forcing (RF). We use the formulation from Etminan et al. (2016), which accounts for the short-wave
absorption by CH$_4$ and the overlap with N$_2$O. The atmospheric oxidation of methane (by the hydroxyl radical) leads to the
production of tropospheric ozone and stratospheric water vapour. We calculate these indirect contributions of methane to the
overall radiative forcing, following the approach for methane adopted in our previous work (Collins et al., 2018; Comyn-Platt
et al., 2018a; Gedney et al., 2019). Collins et al. (2018) represent the forcing contributions from O$_3$ and stratospheric water
vapour as linear functions of the CH$_4$ mixing ratio, based on the analysis presented in IPCC AR5 Myhre et al. (2013). The
indirect methane forcings amount to $2.36 \times 10^{-4} \pm 1.09 \times 10^{-4}$ W m$^{-2}$ per ppb CH$_4$ (i.e., 0.65±0.3 times the CH$_4$ radiative
efficiency). Hence we incorporate the indirect effects of these CH$_4$ emission changes by an approximation, multiplying the
CH$_4$ radiative forcing by 1.65.
In this study, we use the inverse version of IMOGEN, which follows prescribed temperature pathways (Fig. 3(a)), to
derive the total radiative forcing ($\Delta Q$ [total]) and then the CO$_2$ radiative forcing ($\Delta Q$ [CO$_2$]), using Eq. (2). Comyn-Platt et al.
(2018b) describe the changes made to the EBM to create the inverse version. As each of the 34 GCMs that IMOGEN emulates
has a different set of EBM parameters, each GCM has a different time-evolving radiative forcing ($\Delta Q$) estimate for a given
temperature pathway, $\Delta T_G (t)$. When IMOGEN is forced with a historical record of $\Delta T_G$, the range of $\Delta Q$ for the near present
day (year 2015) from the 34 GCMs is 1.13 W m$^{-2}$. To ensure a smooth transition to the modelled future, we require the historical

period, 1850-2015, to match observations of both $\Delta T_G$ and atmospheric composition for all GCMs. As we have a model-specific estimate of the radiative forcing modelled for the $CO_2$ changes (see above), we, therefore, attribute the spread in $\Delta Q$ to the uncertainty in the non-$CO_2$ radiative forcing component, particularly the atmospheric aerosol contribution, which has an uncertainty range of -0.5 to -4 $Wm^{-2}$ (Stocker et al., 2013). Apart from our modelled $CH_4$ and $CO_2$ radiative forcings and the potential future balances between them, we use the projections from the IMAGE SSP2 baseline or RCP1.9 scenario for the radiative forcing of other atmospheric contributors (Fig. 3(b)).

### 2.2.2 Temperature Profile Formulation

Huntingford et al. (2017) define a framework to create trajectories of global temperature increase, based on two parameters, and which model the efforts of humanity to limit emissions of greenhouse gases and short-lived climate forcers, and, if necessary, capture atmospheric carbon. These profiles have the mathematical form of:

$$\Delta T(t) = \Delta T_0 + \gamma t - (1 - e^{-\mu(t)t})[\gamma t - (\Delta T_{Lim} - \Delta T_0)] \tag{5}$$

where $\Delta T(t)$ is the change in temperature from pre-industrial levels at year $t$, $\Delta T_0$ is the temperature change at a given initial point (in this case $\Delta T_0 = 0.89°C$ for 2015), $\Delta T_{Lim}$ is the final prescribed warming limit and

$$\mu(t) = \mu_0 + \mu_1 t \text{ and } \gamma = \beta - \mu_0(\Delta T_{Lim} - \Delta T_0) \tag{6}$$

where $\beta$ (= 0.00128) is the current rate of warming and $\mu_0$ and $\mu_1$ are tuning parameters which describe anthropogenic attempts to stabilise global temperatures (Huntingford et al., 2017). The parameter values used for the two profiles are: (a) 1.5°C profile: $\Delta T_{lim} = 1.5°C$; $\mu_0 = 0.1$ and $\mu_1 = 0.0$; (b) 2°C profile: $\Delta T_{lim} = 2°C$; $\mu_0 = 0.08$ and $\mu_1 = 0.0$.

### 2.3 Scenarios and model runs

We undertake a control run and other simulations with anthropogenic $CH_4$ mitigation or land-based mitigation, stabilising at either 1.5°C or 2.0°C warming without a temperature overshoot. We denote the control run as "CTL", the anthropogenic $CH_4$ mitigation scenario, a land-based mitigation scenario using BECCS and a variant land-based scenario focussing on AR, as "$CH_4$", "BECCS", "Natural", respectively. We also undertake runs combining the $CH_4$ and land-based mitigation scenarios (coupled "BECCS+$CH_4$" and coupled "Natural+$CH_4$") to determine if there are any non-linearities when we combine these mitigation scenarios. We summarise the key assumptions of these scenarios in Table 1.

We use future projections of atmospheric $CH_4$ concentrations and LULUC (specifically, the areas assigned to agriculture and within that to BECCS) from the IMAGE SSP2 projections (Doelman et al., 2018) as input or prescribed data for both the methane and land-based mitigation strategies (Table 1). This ensures that all projections are consistent and based on the same set of IAM model and socio-economic pathway assumptions. The SSP2 socio-economic pathway is described as "middle of the road" (O'Neill et al., 2017), with social, economic, and technological trends largely following historical patterns observed over the past century. Global population growth is moderate and levels off in the second half of the century. The intensity of resource and energy use declines. We define the upper and lower limits of anthropogenic mitigation as the lowest (RCP1.9, denoted "IM-1.9") and highest ("baseline", denoted "IM-BL") total radiative forcing pathways, respectively, within the IMAGE SSP2 ensemble (Riahi et al., 2017). As described in Section 2.2.1, we modify the atmospheric concentrations of $CH_4$ in the IMOGEN-JULES modelling as the IMAGE scenarios assume constant natural and hence wetland methane emissions.

### 2.3.1 Methane: baseline and mitigation scenario

The anthropogenic $CH_4$ emission increase from 318 Tg yr$^{-1}$ in 2005 to 484 Tg yr$^{-1}$ in 2100 in the IMAGE SSP2 baseline

scenario, but fall to 162 Tg yr$^{-1}$ in 2100 in the IMAGE SSP2 RCP1.9 scenario. The sectoral $CH_4$ emissions in 2005 (Energy
Supply & Demand: 113; Agriculture: 136; Other Land Use (primarily burning): 18; Waste 52, all in Tg yr$^{-1}$) are in agreement
with the latest estimates of the global methane cycle (Saunois et al., 2020). As summarised in Supplementary Information,
Table SI.1, the reduction in $CH_4$ emissions from specific source sectors is achieved as follows: (a) coal production by
maximising $CH_4$ recovery from underground mining of hard coal; (b) oil/gas production & distribution, through control of
fugitive emissions from equipment and pipeline leaks, and from venting during maintenance and repair; (c) enteric
fermentation, through change in animal diet and the use of more productive animal types; (d) animal waste by capture and use
of the $CH_4$ emissions in anaerobic digesters; (e) wetland rice production, through changes to the water management regime
and to the soils to reduce methanogenesis; (f) landfills by reducing the amount of organic material deposited and by capture of
any $CH_4$ released; (g) sewage and wastewater, through using more wastewater treatment plants and also recovery of the $CH_4$
from such plants, and through more aerobic wastewater treatment. The levels of reduction vary between sectors, from 50%
(agriculture) to 90% (fossil-fuel extraction and delivery). The abatement costs are between US$ 300-1000 (1995 US$)
(Supplementary Information, Table SI.1). Figure 4 presents the IMAGE baseline and RCP1.9 $CH_4$ emission pathways globally
and for selected IMAGE regions, including the major-emitting regions of India, USA and China (Supplementary Information,
Figure SI.1 shows the emission pathways for all 26 IMAGE regions). These two methane emission pathways (IMAGE SSP2
baseline and RCP1.9) define our "CTL" and "$CH_4$" scenarios, respectively.
**2.3.2 Land-based mitigation: baseline, BECCS and Natural scenarios**

For our land-based mitigation scenarios, we take time series of the annual areas assigned to agriculture (crops and pasture)

and within that, the area allocated to bioenergy crops, from the IM-BL and IM-1.9 scenarios (defined at the start of Sect. 2.3).
We use the dynamic vegetation module in JULES to calculate the evolution of the natural plant functional types and the non-
vegetated surface on the remaining land area in the grid cell (see Land use in Sect. 2.1).

The IM-BL LULUC scenario assumes (a) moderate land-use change regulation; (b) moderately effective land-based

mitigation; (c) the current preference for animal products; (d) moderate improvement in livestock efficiencies; and (e) moderate
improvement in crop yields (Table 1 in (Doelman et al., 2018)). It represents a control scenario within which agricultural land
is accrued to feed growing populations associated with the SSP2 pathway and with no deployment of BECCS. Three types of
land-based climate change mitigation are implemented in the IMAGE land use mitigation scenarios (Doelman et al., 2018):
(1) bioenergy; (2) reducing emissions from deforestation and degradation (REDD or avoided deforestation); and (3)
reforestation of degraded forest areas. For the IM-1.9 scenario, there are high levels of REDD and full reforestation. The
scenario assumes a food-first policy (Daioglou et al., 2019) so that bioenergy crops are only implemented on land not required
for food production (e.g., abandoned agricultural crop land, most notably, in central Europe, southern China and eastern USA,
and on natural grasslands in central Brazil, eastern and southern Africa, and Northern Australia (Doelman et al., 2018)). The
IM-1.9 scenario also requires bioenergy crops to replace forests in temperate and boreal regions (notably Canada and Russia).
The demand for bioenergy is linked to the carbon price required to reach the mitigation target (Hoogwijk et al., 2009). In this
scenario, the area of land used for bioenergy crops expands rapidly from 2030 to 2050, reaching a maximum of 550 Mha in
2060, and then declining to 430 Mha by 2100. Table 2 gives the maximum area of BECCS deployed in each IMAGE region
for the IM-1.9 scenario. This defines the land use in the "BECCS" scenario.

We define a third LULUC pathway, which is identical to the "BECCS" scenario, except that any land allocated to

bioenergy crops is allocated instead to natural vegetation, i.e., areas of natural land, which are converted to bioenergy crops,

remain as natural vegetation, and areas, which are converted from food crops or pasture to bioenergy crops, return to natural vegetation. We make no allowance for any changes in the energy generation system, as this would require energy sector modelling that is beyond the scope of this study. We denote this scenario as "Natural". Table 2 also summarises the main differences in land use between the BECCS and Natural scenarios for each IMAGE region.

Figure 5 presents time series of the land areas calculated for trees and prescribed for agriculture (including bioenergy crops) and bioenergy crops for the "BECCS" and "Natural" scenarios for the Russia and Brazil IMAGE regions, each as a difference to the baseline scenario (IM-BL). Supplementary Information, Figure SI.2 is equivalent to Fig. 5 for all the IMAGE regions.

### 2.3.3 Model runs

For each temperature pathway (1.5°C or 2.0°C) and for the baseline and each mitigation scenario, the set of scenario runs comprises a 136-member ensemble (34 GCMs x 2 ozone damage sensitivities x 2 methanogenesis $Q_{10}$ temperature sensitivities). In all model runs, we include the effects of the methane and carbon-climate feedbacks from wetlands and permafrost thaw, which we have shown previously to be significant constraints on the AFFEBs (Comyn-Platt et al., 2018a).

As shown in Fig. 1, we use a number of input or prescribed datasets: (a) time series of the annual area of land used for agriculture, including that for BECCS if appropriate; (b) time series of the global annual mean atmospheric concentrations of $CH_4$ (and $N_2O$ for the radiative forcing calculations of $CO_2$ and $CH_4$); (c) time series of the overall radiative forcing by SLCFs and non-$CO_2$ GHGs (corrected for the radiative forcing of $CH_4$); and (d) time series of annual anthropogenic $CH_4$ emissions (used in the post-processing step). We take these from the IMAGE database for the relevant IMAGE SSP2 scenario (baseline or SSP2-1.9). Table 1 lists the main scenario runs, their key features and the prescribed datasets used (for agricultural land and BECCS, anthropogenic emissions and atmospheric concentrations of $CH_4$ and the non-$CO_2$ radiative forcing).

Figure 6 presents the effect of these scenarios on the modelled atmospheric $CH_4$ and $CO_2$ concentrations. We adjust the input atmospheric $CH_4$ concentrations to allow for the interannual variability in the wetland $CH_4$ emissions, as described in Sect. 2.2.1. As we use the same input datasets for the two warming targets, the major control on the modelled atmospheric $CH_4$ concentrations is the $CH_4$ emission pathway followed, with the temperature pathway (1.5° versus 2°C warming) having a minor effect. For $CO_2$, on the other hand, the temperature and the $CH_4$ emission pathways both lead to increased atmospheric $CO_2$ concentrations, with the temperature pathway having a slightly larger effect.

### 2.4  Post-processing

#### 2.4.1 Anthropogenic Fossil Fuel Emission Budget and Mitigation Potential

Following Comyn-Platt et al. (2018b), we define the anthropogenic fossil fuel $CO_2$ emission budget (AFFEB) for scenario $i$ as the change in carbon stores from present to the year 2100:

$$AFFEB_i = [\, C^{land}(2100) - C^{land}(2015)]_i + [C^{ocean}(2100) - C^{ocean}(2015)]_i$$

$$+ [C^{atmos}(2100) - C^{atmos}(2015)]_i + BECCS(2015:2100)_i \qquad (7)$$

where $C^{land}(t)$, $C^{ocean}(t)$ and $C^{atmos}(t)$ are the carbon stored in the land, ocean and atmosphere, respectively, in year $t$ and $BECCS(t_1:t_2)$ is the carbon sequestered via BECCS between the years $t_1$ and $t_2$. The atmospheric carbon store does not include

CH₄. This is a reasonable approximation, however, given the relative magnitudes of the atmospheric concentrations of $CH_4$
(~2 ppmv at the surface) and $CO_2$ (400 ppmv).
Within the IMOGEN-JULES modelling framework, we use (a) the IMOGEN climate emulator to derive the changes in
the ocean and atmosphere carbon stores, and (b) JULES for the changes in the land carbon store and carbon sequestered
through BECCS. We discuss the changes in the carbon stores for the baseline and different mitigation scenarios in Sect. 3.1.
For brevity in the subsequent discussion, we use the following shorthand where the terms on the RHS of Eq. (7) are
equivalent to those on the RHS of Eq. (8):
$$AFFEB_i = \Delta C_i^{land} + \Delta C_i^{ocean} + \Delta C_i^{atmos} + BECCS_i \qquad (8)$$
We define the mitigation potential (MP) for a mitigation strategy, *j*, as the difference between a control AFFEB (AFFEB$_{ctl}$)
and the AFFEB resulting from applying the strategy i.e.:
$$MP_j = AFFEB_j - AFFEB_{ctl} \qquad (9)$$
which can be broken down into its component parts as:
$$MP_j = MP_j^{land} + MP_j^{ocean} + MP_j^{atmos}$$
$$MP_j = \left(\Delta C_j^{land} - \Delta C_{ctl}^{land}\right) + \left(\Delta C_j^{ocean} - \Delta C_{ctl}^{ocean}\right) + \left(\Delta C_j^{atmos} - \Delta C_{ctl}^{atmos}\right) + BECCS_j \qquad (10)$$

**2.4.2 Optimisation of the land-based mitigation**
Harper et al. (2018) find that the land-use pathways do not provide a clear choice for the preferred mitigation pathway.
The key issue is that replacing natural vegetation with bioenergy crops often results in large emissions of soil carbon and the
loss of the benefits of maintaining forest carbon stocks. In such circumstances, Harper et al. (2018) find that the loss of soil
carbon in regions with high carbon density makes it difficult for BECCS to deliver a net negative emission of $CO_2$. Hence, to
optimise the land-based mitigation (LBM), we compare the land-carbon stocks in the BECCS and Natural scenarios. We then
select the optimum land-management option for each grid cell simulated as that, which maximises the *AFFEB* by year 2100.
That is:
$$AFFEB_{LBM} = \Delta C_{BECCS}^{atmos} + \Delta C_{BECCS}^{ocean} + \Delta C_{LBM}^{land} \qquad (11)$$
with
$$\Delta C_{LBM}^{land} = \begin{cases} \sum_l^{grid\ cells} \Delta C_{BECCS}^{land} + BECCS & where\ \Delta C_{BECCS}^{land} < \Delta C_{BECCS}^{land} + BECCS \\ & or \\ \sum_l^{grid\ cells} \Delta C_{Natural}^{land} & where\ \Delta C_{Natural}^{land} > \Delta C_{BECCS}^{land} + BECCS \end{cases} \qquad (12)$$
where $\Delta C_{scenario}^{store}$ is the change in carbon between 2015 and 2100 for the 'store' (= atmosphere, ocean or land) for the LULUC
scenario. We use the ocean and atmosphere contributions from the BECCS simulations as the changes in store size between
the BECCS and Natural simulations are negligible (i.e. <2GtC).


### 2.4.3 Assumptions about BECCS efficiency

The efficacy of the BECCS scheme implemented in JULES is significantly lower than that of other implementations (Harper et al., 2018), reflecting the importance of assumptions about the efficiency of the BECCS process and bioenergy crop yields in determining their ability to contribute to climate mitigation. More specifically, there is (1) large uncertainty in carbon losses from farm to final storage (Harper et al. (2018) assumed a 40% loss compared to 13-52% loss found in other studies); and (2) a large range in potential productivity of second-generation lignocellulosic bioenergy crops, with JULES falling on the low end. JULES in this study and in Harper et al. (2018) simulated median average yields of ~4.8 and ~4.6 tDM ha$^{-1}$ yr$^{-1}$ , respectively, compared to measured median of 11.5 tDM ha$^{-1}$ yr$^{-1}$ and simulated average of 15.8 tDM ha$^{-1}$ yr$^{-1}$ in IMAGE. The JULES yield of ~4.8 tDM ha$^{-1}$ yr$^{-1}$ corresponds to ~59 EJ yr$^{-1}$ of primary energy, using the maximum area for BECCS from Table 2 of 637.7 Mha and an energy yield of 19.5 GJ t DM$^{-1}$ (Daioglou et al., 2017). Bioenergy supplied 55.6 EJ yr$^{-1}$ or ~10% of primary energy requirement worldwide in 2017 (WBA, 2019). According to Smith et al. (2016), this would increase to ~170 EJ yr$^{-1}$ of primary energy in 2100, for negative emissions of 3.3 Gt Ceq yr$^{-1}$ from BECCS (as required for a 2°C warming target).

As both of these components are assumed to be diagnostics of the simulations, we can modify the contribution of BECCS to the AFFEB via a post-processing scaling factor, $\kappa$, which represents the efficiency of (1) and (2) with respect to the JULES parameterisation. That is, Eq. (12) becomes:

$$\Delta C_{LBM}^{land} = \begin{cases} \sum_l^{grid\ cells} \Delta C_{BECCS}^{land} + \kappa\ BECCS & \text{where } \Delta C_{Natural}^{land} < \Delta C_{BECCS}^{land} + \kappa\ BECCS \\ & or \\ \sum_l^{grid\ cells} \Delta C_{Natural}^{land} & \text{where } \Delta C_{Natural}^{land} > \Delta C_{BECCS}^{land} + \kappa\ BECCS \end{cases} \qquad (13)$$

Figure 7 presents maps of the scaling factor required for BECCS to be the preferable mitigation option, as opposed to natural land carbon uptake, for each grid cell for warming of 1.5°C or 2°C. There are large factors in the northern temperate and boreal regions, parts of Africa and Australia. As discussed in Harper et al. (2018), this follows from the loss of soil carbon in the tropics and at high northern latitude leading to long recovery or payback times (10-100+ years and >100 years, respectively, Fig. 6(c) in their paper). The payback time is however insignificant when bioenergy crops replace existing agriculture, for example in Europe and eastern North America.

Additionally, we define a threshold efficiency factor, $\kappa^{*}$, which represents the required BECCS efficiency for BECCS to be a preferable mitigation strategy for a given grid-cell, i.e.:

$$\kappa^{*} = \frac{\Delta C_{INatural}^{land} - \Delta C_{BECCS}^{land}}{BECCS} \qquad (14)$$

This increased efficiency can be considered to be the additional bioenergy harvest (H) and/or the reduced carbon losses from farm to storage needed to pay back the carbon debt accrued due to land-use change (since carbon removed via BECCS = Hε, where ε is the assumed efficiency factor for farm to storage carbon conservation and H is the simulated biomass harvest). In addition, $\kappa^{*}$ implies a new threshold (or break-even) level of BECCS:

$$BECCS^{*} = \kappa^{*} * BECCS \qquad (15)$$

In other words, BECCS$^{*}$ is equivalent to the carbon loss due to the land use change to grow the bioenergy crops. Our IMOGEN-JULES simulations assume a 40% carbon loss from farm to final storage, although other studies have assumed this to be as low as 13% (Harper et al., 2018). To assess the feasibility of meeting this break-even level of BECCS, we calculate

the harvest (H*) that would be needed if carbon losses are to be minimised, i.e. by increasing ε from 0.6 to 0.87, and assuming
in Eq. (15) that:
$BECCS^* = 0.87 \, H^*$ and $BECCS = 0.60 \, H$
So:
$H^* = \kappa^* * \frac{0.6}{0.87} * H$          (16)
We discuss this further in Sect. 3.2.
**3    Results and Discussion**
**3.1   Global Perspective**
We calculate the anthropogenic fossil fuel emission budget to limit global warming to a particular temperature target as
the sum of the changes in the carbon stores of the atmosphere, land (vegetation and soil) and ocean between 2015 and 2100
(Sect. 2.4.1, Eq. (7) and (8)). We use a BECCS scale factor ($\kappa$) of unity. In Fig. 8, we present the median and spread of the
AFFEB (as box and whiskers) from the 136-member ensemble, and the individual GCM/ESM contributions to the AFFEBs
from the four carbon pools shown (points), for each of the main scenarios modelled using the IMOGEN-JULES or derived in
the post-processing optimisation step (see Table 1 for description of the scenarios).
In all the scenarios apart from the BECCS scenario, there is an increase in the land carbon store (shown as positive changes
for Coupled (Natural) and Coupled (Optimised) but as smaller negative changes for "CH₄", "Natural" and "Optimised"
scenarios. In the "BECCS" scenario, the land carbon change becomes more negative than in the "CTL" scenario, as bioenergy
crops replace ecosystems with higher carbon content. In the combined ('coupled') CH₄ and land-based mitigation scenarios,
the reduction in the emissions and hence atmospheric concentrations of CH₄ allow increased atmospheric concentrations of
$CO_2$ (Fig. 6). There is increased uptake of carbon by the land, directly because of the increased atmospheric $CO_2$ concentration
and indirectly through the reduction in $O_3$ damage. In the coupled "BECCS" scenario, this increased uptake of atmospheric
$CO_2$ is again offset by the land carbon lost through conversion of the land to bioenergy crops. We also find that there is
increased uptake of $CO_2$ by the oceans for all scenarios. A further co-benefit of reducing the CH₄ emissions and allowing more
$CO_2$ emissions is that the oceanic drawdown of $CO_2$ rises (although it eventually falls to zero under climate stabilisation and
there would also be implications for ocean acidification). In Fig. 9(a), we compare the AFFEBs for both the 1.5°C and 2°C
temperature pathways. We find that the absolute AFFEBs are 200-300 GtC larger for the 2°C target than the 1.5°C target.
These budgets are in agreement with other estimates, which include corrections to the historical period (Millar et al., 2017). In
both Figs. 8 and 9, it should be noted that the land carbon store for the "CH₄" mitigation option at -1.4 GtC (median of
ensemble) is not visible in these figures. There has however been a net increase in the land carbon store in the "CH₄" scenario
when compared to the land carbon store in the control scenario (-70.8 GtC, median of ensemble). This then explains the positive
changes shown for the land carbon stores in the coupled "BECCS+ CH₄" and coupled "Natural+ CH₄" scenarios.
Figure 9(b) shows the mitigation potential of each strategy, calculated as the change in the AFFEB from the corresponding
control simulation, for the two temperature pathways (Sect. 2.4.1, Eq. (9) and (10)). Methane mitigation is a highly effective
strategy; the AFFEBs are increased by 188-206 GtC and 193-212 GtC for the 1.5°C and 2°C scenarios, respectively, where
the range represents the interquartile range from the 136-member ensemble (34 GCMs x 2 Q₁₀ x 2 ozone sensitivities). This
AFFEB increase equates to roughly 20-24 years of emissions at current rates for the 1.5°C target. Land-based mitigation
strategies also provide significant increases of 51-57 GtC and 56-62 GtC for the 1.5°C and 2°C AFFEB estimates, respectively.
This is equivalent to 6-7 years of emissions at current rates. For our BECCS assumptions (see also below), we find that the
BECCS contribution is small for the optimised land-based mitigation pathway and that AR are more effective land-based
mitigation strategies (Fig. 9(b)). Although the primary challenge remains mitigation of fossil fuel emissions, these results
highlight the potential of these mitigation options to make the Paris climate targets more achievable.
Furthermore, the $CH_4$ and land-based mitigation strategies show little interaction and their potential can be summed to
give a comparable result to the coupled simulation (coupled vs linear in Fig. 9(a) and (b)). This decoupling is despite the $CH_4$
emissions from the agricultural sector being influenced by land use choices. We can effectively treat the two mitigation
strategies as independent, and their sum approximates the combined potential. Such linearity enables simpler and more direct
comparisons.
Despite the substantial differences in the absolute AFFEBs for the 1.5° and 2°C targets, the mitigation potential of the
$CH_4$ and land-based strategies is similar for the two temperature pathways considered. This similarity suggests that the
mitigation strategies are robust to the target temperature; whether the international community aims for the 1.5° or 2°C target,
afforestation, reforestation, reduced deforestation and $CH_4$ mitigation are beneficial mitigation approaches.
For both temperature pathways (i.e., 1.5°C or 2°C of warming), we investigate the contribution to the uncertainty range
from 'climate' as represented by the 34 GCMs emulated and from the land processes investigated (Sect. 2.1). A GCM with
higher climate sensitivity will have a lower AFFEB for a specific warming target (and vice versa). In our post-processing steps,
we derive a number of statistical parameters from the complete 136-member or the 34-member GCM ensemble for the
individual factorial runs (low $Q_{10}$/low $O_3$, low $Q_{10}$/high $O_3$, high $Q_{10}$/low $O_3$ and high $Q_{10}$/high $O_3$), such as mean, standard
deviation, median, and various percentiles. Our focus is on the contribution different factors make to the overall standard
deviation of the 136-member ensemble ($\sigma_{All}$). By factoring out the climate variation (via their means), we calculate the standard
deviation for the land processes investigated ($\sigma_{land}$). With a knowledge of the overall standard deviation and that for land-only
processes, we derive the contribution from 'climate' ($\sigma_{climate}$) assuming that the variance are independent and can be summed
(Eq. (17)). The contributions of uncertainty are by comparing ratios of $\sigma_{land}$ to $\sigma_{climate}$.
$$\sigma_{all}^2 = \sigma_{climate}^2 + \sigma_{land}^2 \qquad\qquad\qquad (17)$$
We present the results of this analysis in Table 3 for the Anthropogenic Fossil Field $CO_2$ Emission Budgets and the
Mitigation Potential (= scenario – "CTL") for the 1.5°C temperature profile (Supplementary Information, Table SI.2 is
equivalent table for the 2°C temperature profile). Our overall finding is that the climate uncertainty dominates the uncertainty
of the AFFEBs. However, when considering different trade-offs between land uncertainty and mitigation options, the impact
of climate uncertainty is much weaker. Within the land uncertainty, the $O_3$ vegetation damage appears to make the greater
contribution (from the changes in the mean). Although there is some variation in the ratio ($\sigma_{climate}$:$\sigma_{land}$) between the scenarios
(0.32±0.13, mean ± standard deviation), this gives us confidence in the robustness of the uncertainty estimates derived, across
the scenarios and the 2 temperature profiles.
**3.2 Sensitivity to BECCS Efficiency**
The BECCS parameterisation used here makes BECCS less effective compared to those in other studies (van Vuuren et
al., 2018). Globally across the two temperature targets, our simulations imply a removal of 27-30 GtC from the active carbon
cycle via BECCS in the original "BECCS" scenario run, which is reduced to ~7-12 GtC after we optimise the land-use scenario.
These removal rates are significantly lower than other estimates based on the same land-use scenarios: 73 GtC in a similar

dynamic global vegetation model (LPJ-GUESS) and 130 GtC in IMAGE (Harper et al., 2018). We find that doubling the carbon captured with BECCS in our simulations (Sect. 2.4.3, $\kappa=2$) has a relatively small impact on the total mitigation potential in the optimised scenario (Fig. 10(a)). This low sensitivity is because the increased carbon removed by BECCS often accompanies a comparable decrease in the carbon uptake from the "natural" vegetation that it replaces. It is only when setting the BECCS carbon sequestration at 3-5 times its original value that there is a notable increase of the global AFFEB. Further, as shown in Fig. 10(b), there is reduction in soil carbon in specific regions (e.g. Northern temperate and boreal regions), which makes BECCS less effective for carbon sequestration than natural land management options (or there is a long payback time as discussed in Harper et al. (2018)).

Increased carbon removal with BECCS could be realised through either (1) minimizing the loss of carbon from farm to final storage ($\varepsilon$ in Sect. 2.4.3), or (2) maximizing the productivity of the bioenergy crop. Our IMOGEN-JULES simulations assume a 40% carbon loss from farm to final storage, although other studies have assumed this to be as low as 13% (Harper et al., 2018). The bioenergy crop yields in JULES (Fig. 10(c)) are lower than the median yield of Miscanthus (11.5 tons of dry matter (ton DM) ha$^{-1}$ yr$^{-1}$), measured from 990 mostly European plots (Li et al., 2018), and are about half the productivity of those in the IMAGE simulations. We calculate for each IMOGEN grid cell the increase in carbon removed via BECCS and the associated increase in bioenergy crop yields ($H^*$ in Sect. 2.4.3) required for BECCS to be the preferred mitigation option (Fig. 10(d)), rather than natural land carbon uptake, and assuming minimal amounts of carbon are lost during the BECCS lifecycle (13% carbon loss). In many places, we find that the required yield increases from <10 to 10-20 ton DM ha$^{-1}$ yr$^{-1}$ are achievable, but required yields of > 30 ton DM ha$^{-1}$ yr$^{-1}$ would be more difficult to realise, given the range of yields observed (Li et al., 2018). We provide additional information in the Supplementary Information, Tables SI.4a-SI.4d on the modelled bioenergy yields and the yields required for bioenergy crops to be the preferred land-based mitigation option by IMAGE region. The tables also show that area of bioenergy crops and carbon sequestered by BECCS increases, as expected, with the BECCS scale factor ($\kappa$),

We conclude that our uncorrected simulations are a lower estimate for the potential of carbon removal via BECCS. We provide a more optimistic estimate of the BECCS potential using $\kappa = 3$, which results from doubling the JULES yields and increasing the efficiency $\varepsilon$ from 0.6 to 0.87 (i.e., $\kappa \sim 2 \times 0.87 / 0.6$). We now find the global land-based mitigation potential to be 88-100 GtC across the two temperature targets, as shown in Fig. 9(c) and (d). Supporting Information, Figure SI.3 shows the corresponding plots for the 2°C warming target. We use $\kappa = 3$ in the subsequent analysis of regional mitigation options and of BECCS water requirements.

## 3.3 Regional Analysis

We consider the sub-continental implications of CH$_4$ and land-based mitigation options, using the 26 regions of the IMAGE model (Stehfest et al., 2014). Figure 11 shows the contributions of the three mitigation options - CH$_4$, carbon uptake through AR and BECCS - to the AFFEBs for each IMAGE region and for the temperature pathway stabilising at 1.5°C.

We estimate the regional land-based mitigation as the change in the land-carbon stores plus the carbon removal via BECCS for each IMAGE region in the IMOGEN-JULES model output. In this accounting, the region where the bioenergy crops are grown is credited with the carbon removal via BECCS. We assume a three-fold increase in carbon removal via BECCS compared to our default simulations ($\kappa=3$) to highlight regions where BECCS is potentially viable. Figure 12 shows the sensitivity of the global AFFEBs and Mitigation Potential for $\kappa = 1$, 2 and 3 for 1.5°C of warming (Supplementary Information, Figure SI.3 is the corresponding figure for 2°C of warming). For CH$_4$, we use regional scale factors to allocate changes in the global atmospheric CH$_4$ concentration, and therefore the CH$_4$ mitigation potential, to each region, as shown in Supplementary

Information, Table SI.3. To derive the regional scale factors, we separately sum the projected anthropogenic $CH_4$ emissions between 2020 and 2100 between the IMAGE SSP2-Baseline and SSP2-1.9 scenarios (van Vuuren et al., 2017). We calculate the scale factor as the regional fraction of the global difference in the summed emissions (Supplementary Information, Table SI.3). These two $CH_4$ scenarios are consistent with the $CH_4$ concentration pathways considered in the $CH_4$ scenario simulations (Sect. 2.3). We use the scale factors to produce Fig. 11 and 12 (and Supplementary Information, Figures SI.3 and SI.4).

$CH_4$ mitigation is an effective mitigation strategy for all regions, and especially the major methane emitting regions: India, S. Africa, USA, China and Australasia. Figure 4 presented time series of the anthropogenic $CH_4$ emissions for selected IMAGE region from 2000 to 2100 (and Supplementary Information, Figure SI.1 presents emission time series for all IMAGE regions). The mitigation of $CH_4$ emissions from fossil-fuel production, distribution and use for energy is the largest contributor for India, S. Africa, USA, China and Australasia. The emissions from agriculture-cattle (for India, USA and China) and rice production (China and other Asian regions) make smaller contributions.

The impact of the land-based mitigation options links strongly to the managed land-use and land-use change (LULUC). As discussed in Sect. 2.3.2, we list in Table 2 the maximum area of BECCS deployed in each IMAGE region and the main differences in land use between the BECCS and Natural scenarios. Figure 5 presents time series of the land areas calculated for trees and prescribed for agriculture (including bioenergy crops) and bioenergy crops for the BECCS and Natural scenarios for the Russia and Brazil IMAGE regions, each as a difference to the baseline scenario (IM-BL) (see Supplementary Information, Figure SI.2 for all the IMAGE regions). The West Africa region shows the largest natural land carbon uptake (WAF in Fig. 12). Here, there is conversion of crop and pasture to forest, with little land used for bioenergy crops for BECCS. For Brazil (Fig. 5(a)) and the rest of South America, both bioenergy crops and forest expand at the expense of agricultural land. For many other regions, notably Canada, Russia, W. & C. Europe, China, Oceania, there is less carbon uptake from the 'land' in the optimised mitigation scenario, even though the overall carbon uptake has increased. For Canada and Russia, this results from the loss of forest in the BECCS land use scenario (see Fig. 5(b) and Supplementary Information, Figure SI.3). The carbon uptake by BECCS increases as κ increases from 1 to 3 because there are more grid cells where 'BECCS' is the preferred mitigation option in the optimisation process, as evidenced by the increase in area of bioenergy crops (Supplementary Information, Tables SI.4a and SI.4c). As κ only affects the 'BECCS' term (Sect. 2.4.3, Eq. (13)), the increased carbon removed by BECCS is often accompanied by a decrease in the carbon uptake from the "natural" vegetation that it replaces. This can be seen more clearly in Fig. 12 (and Supplementary Information, Figure SI.3 for 2°C warming) and the Supplementary Information, Tables SI.4b and SI.4d . The version of JULES used in this study currently lacks a fire regime. There will be risks to long-term storage of carbon stored in vegetation in regions with significant areas of fire-dominated vegetation cover (e.g. savannah in Brazil and Africa). Further, this version of JULES does not include a nitrogen cycle, which has been implemented in more recent versions of the model. This will enable the impact of changes in land use and agriculture on $N_2O$ emissions to be integrated into the assessments.

There is relatively little difference in the additional allowable carbon emission budgets introduced by $CH_4$ and/or the land-based mitigation between 2015 and 2100 for the two temperature pathways considered (Supplementary Information, Figure SI.4 for the contributions at 2°C of warming).

**3.4 Water Resources**

Smith et al. (2016) estimate the global water requirements for different negative emission technologies, including BECCS. We also derive the water requirements from the carbon uptake by BECCS for our optimised land-based mitigation scenarios. The IM-1.9 land use scenario (Sect. 2.3.2) assumes that bioenergy crops are grown sustainably and are rain-fed (Daioglou et

al., 2019; Hoogwijk et al., 2005). Our land surface modelling system explicitly accounts for this. We derive the additional water requirements for BECCS, using $\kappa = 3$ and assuming (a) a marginal increase in water use of 80 m$^3$ (tC eq)$^{-1}$ yr$^{-1}$ when replacing the average short vegetation (i.e., C3/C4 grasses in JULES) by a biomass energy crop (Smith et al., 2016); and (b) 450 m$^3$ (tC eq)$^{-1}$ yr$^{-1}$ for the CCS component (Smith et al., 2016).

Following Postel et al. (1996), we derive the accessible runoff, using their assumptions that only 5% of the total runoff is geographically and/or temporally accessible for the Brazil, Russia and Canada IMAGE regions, and 40% elsewhere. Our present-day estimates of the global annual runoff (43,000-44,200 km$^3$ yr$^{-1}$) and the accessible runoff for human use (11,400-11,720 km$^3$ yr$^{-1}$) (see Fig. 13) are both in agreement with the values given in Postel et al. (1996), i.e., total and accessible runoffs of 40,700 and 12,500 km$^3$ yr$^{-1}$, respectively.

We use the water withdrawals for each IMAGE region given in the IMAGE-SSP2-RCP2.6 scenario for the water demand for agricultural irrigation (Rost et al., 2008) and for other human activities, such as energy generation, industry and domestic usage (Bijl et al., 2016), between 2015 and 2100 (Table 4a and 4b). We assume the same water demands from these sectors for both the 1.5°and 2°C warming targets.

Figure 14 compares the accessible water with the water demand for BECCS and other human activities for the regions that produce a substantial amount of BECCS: Canada, USA, Brazil, Europe, Russia, China, Southern Africa and Oceania for the optimised land-based mitigation. Table 4a and b show the additional water requirements of BECCS calculated for 2060 and 2100, respectively, for the 2°C warming target. We find that the additional demand for BECCS would lead to an exceedence (or use >90%) of the available water for the Oceania and Rest of Southern Africa regions. We also find that the additional demand for BECCS is greater than the total water withdrawals from anthropogenic activities for the Canada and Brazil IMAGE regions. Our estimates represent a maximum possible water usage for BECCS as (i) the SSP2 scenario used already accounts for the lower power generation efficiencies and hence higher water requirements in switching from fossil fuels to bioenergy crops (which could be up to 20-25%) and (ii) the figure used for the CCS component does not allow for future technological improvements in water use. For example, Fajardy and Mac Dowell (2017) indicate a 30-fold reduction in water use when changing from a once-through to a recirculating cooling tower. Our results are less severe than other studies considering BECCS water requirements (Séférian et al., 2018; Yamagata et al., 2018), because the carbon removed by BECCS in this study (30 GtC) is already limited to regions where it is more beneficial to the AFFEB than forest-based mitigation options. We also note from Bijl et al. (2016) that the water demand for irrigation, derived using the coupled IMAGE-LPJmL models, is low compared to other estimates in the literature. Higher water demand for irrigation existing agriculture would be an additional constraint on the water available for BECCS. Nevertheless, our results indicate that the additional water demand for BECCS would have large impacts in half of the regions substantially invested in BECCS: Oceania, Rest of South Africa, Brazil and Canada.

## 4    Conclusions

Our paper brings together previous studies that looked separately into the potential of methane mitigation (Collins et al., 2018) and land-management options (especially forest conservation and BECCS) (Harper et al., 2018), into a single unified framework. Uniquely, this allows us to compare these options at local and regional scales. We utilise the detailed JULES land-surface model, which includes methane production from wetlands and permafrost thaw (Comyn-Platt et al., 2018a) and the effect of CH$_4$ emissions on land carbon storage via ozone impacts on vegetation (Sitch et al., 2007), and also span the range of climate model projections using the IMOGEN ESM-emulator. For each temperature pathway and each of the three

mitigation options, the set of scenario runs comprises a 136-member ensemble (34 GCMs x 2 ozone damage sensitivities x 2
methanogenesis $Q_{10}$ temperature sensitivities).

This analysis quantifies the regional differences in potential $CH_4$ and/or land-based strategies to aid mitigation of climate

change. We present our findings within a full probabilistic framework, capturing uncertainty in climate projections across the
CMIP5 ensemble, as well as process uncertainties associated with the strength of natural $CH_4$ climate feedbacks from wetlands
and ozone-induced vegetation damage. Globally, mitigation of anthropogenic $CH_4$ emissions and the optimised land-based
mitigation can potentially offset (i.e. allow extra) fossil fuel carbon dioxide emissions of 188-212 GtC and 51-100 GtC,
respectively. These bounds are almost independent of the eventual global-warming target, or the climate sensitivity of the
climate models emulated. As shown in Sect. 3.1, the $CH_4$ and land-based mitigation strategies show little interaction and their
potential can be summed to give a comparable result to the corresponding coupled simulation. This decoupling is despite the
$CH_4$ emissions from the agricultural sector being influenced by land use choices. We can therefore treat the two mitigation
strategies as independent, and sum their individual potentials. Such linearity enables simpler and more direct comparisons
between the carbon budgets of methane and land-based mitigation strategies. Some caveats remain however. Land surface
models still require refinement, alongside improved characterisation of the assumptions inherent in the socio-economic
pathways and IAM modelling. Further, we do not allow for the reduced emissions from fossil fuel combustion due to the
bioenergy crop being grown (or the converse when bioenergy crops are replaced in the Natural model run), as this would
require energy sector modelling that is beyond the scope of this study.

For the "Natural" land-based scenario (see Table 1), we find a mitigation potential of 50-55 GtC (183-201 $GtCO_2$). The

land-based mitigation estimates vary over wide ranges, partly related to different assumptions on land use and carbon pools.
Our results are within the wide range of the overall deployment of $CO_2$ removal by Agriculture, Forestry and Other Land Use
(including afforestation and reforestation) to 2100 of 200 [0-550] $GtCO_2$ (Page 2.40 in IPCC (2018)) and of estimates of the
cumulative potential to 2100 from 80 to 260 $GtCO_2$ (Table 2) in Minx et al. (2018). In the "BECCS" scenario, we obtain a
geological carbon storage via BECCS (27±1 GtC median, interquartile range) similar to that (30±1 GtC) derived by Harper et
al. (2018), for the same land use scenario (IM-1.9). Our result is lower as we include the natural methane feedbacks from
wetlands and permafrost thaw. Inclusion of this better process description leads to ~10% reduction in carbon budgets(Comyn-
Platt et al., 2018a). These estimates for the geological carbon storage via BECCS are much lower than the corresponding value
derived by the IMAGE IAM (130 GtC). Harper et al. (2018) discuss this difference, identifying a number of reasons for the
lower value: the use of initial above ground biomass harvested in boreal forests for BECCS, the replacement of fossil-fuel
based emissions in the energy system, as well as specific assumptions about crop yields, conversion efficiency, use of residues,
the proportion of bioenergy crops used with CCS. Estimates of the BECCS contribution in the literature vary over a wide range
(from 178 to >1000 $GtCO_2$, according to Minx et al. (2018)), but in recent studies these result are typically revised downwards
taking into account among others sustainability constraints (e.g. Fuss et al. (2018) suggests a potential of 0.5-5 $GtCO_2$ per year
in 2050).

We investigate the efficacy of our "BECCS" scenario by increasing the productivity of BECCS (using a scale factor κ).

From comparison with observed bioenergy crop yields, we argue that the scale factor could be between 1 and 3. We highlight
how using this range of κ provides characterisation of an additional source of uncertainty on the land-based mitigation
potential. In our optimised land-based mitigation scenario, which maximises the land carbon uptake (Sect 2.4.2, Eq. (13)), the
increased carbon removed by BECCS is often accompanied by a decrease in the carbon uptake from the "natural" vegetation
that it replaces (as discussed in Sect. 3.3 and shown in Figure 12). This concern is equivalent to the statement in Harper et
al. (2018) that the "use of BECCS in regions where bioenergy crops replace ecosystems with high carbon contents could easily

result in negative carbon balance". Hence the particularly novel feature of our paper is that our optimal approach accounts explicitly for that trade-off, only suggesting BECCS where there is a net gain. For boreal forest regions there is a preference for avoided deforestation, whereas in tropical forest regions both AR and avoided deforestation offer significant potential. From a carbon sequestration perspective, growing bioenergy crops for BECCS is only preferable where it replaces existing agricultural land. BECCS has particular potential if productivities and power production efficiencies are towards the upper limit of expected photosynthetic capability, whilst noting the strong water demand of such crops requires consideration in the context of a growing population.

Stabilising the climate primarily requires urgent action to mitigate $CO_2$ emissions. However, we estimate that $CH_4$ mitigation may offset up to 188-212 GtC of anthropogenic $CO_2$ emissions, while still meeting the same global-warming targets. This offset is a direct consequence of the reduced radiative forcing by methane and of carbon cycle gains. These balances and related flexibilities have the potential to make the Paris targets more achievable. Our range of additional $CO_2$ emissions broadly applies to both the 1.5° and 2°C warming targets, as the mitigation potential of the $CH_4$ scenario is similar for the two temperature pathways considered. Although there are differences in the precise methane emission scenarios used, our mitigation potential is similar to that given in Collins et al. (2018). That paper presents values of 155 or 235 GtC for offsetting $CH_4$ mitigation from a high to a medium or from a high to a low emission scenario, respectively. Our value, and those of Collins et al. (2018), can be compared to the increase of 130 GtC in the carbon budget between a no and a stringent $CH_4$ emission mitigation scenario estimated by Rogelj et al. (2015). More recently, Harmsen et al. (2020) have also investigated the mitigation potential of methane, although their results are expressed in terms of changes in radiative forcing and temperature, rather than carbon budgets. An advantage of our analysis remains the inclusion of climate response to altered radiative forcing, enabling understanding in terms of actual $CO_2$ emissions. We conclude that $CH_4$ mitigation would be effective globally as a contribution to constraining global warming, and especially so for the major $CH_4$-emitting regions of India, USA and China.

**Code and Data Availability**

The IMOGEN-JULES source code used in this work is available from the JULES code repository (https://code.metoffice.gov.uk/trac/jules/browser/main/branches/dev/annaharper/r7971_vn4.8_1P5_DEGREES_CCS, at JULES revision 14477, user account required). The rose suites used for the specific IMOGEN-JULES runs are: u-as624, u-at010, u-at011, u-at013, u-av005, u-av007, u-av008, u-av009, u-ax327, u-ax332, u-ax455, u-ax456, u-ax521, u-ax523, u-ax524, u-ax525, u-bh009, u-bh023, u-bh046, u-bh081, u-bh084, u-bh098, u-bh103 and u-bh105. These can be found at https://code.metoffice.gov.uk/trac/roses-u/ (user account required).

The IMOGEN-JULES source code is also available as a zipped tarball from http://doi.org/10.5281/zenodo.4620139, as are the python scripts used for post-processing. Data and output used with the scripts is available from https://doi.org/10.5281/zenodo.4625977. The pattern-scaling and energy balance parameters used to emulate the CMIP5 models are available at https://doi.org/10.5285/343885af-0f5e-4062-88e1-a9e612f77779. We will look to make other relevant outputs from the IMOGEN-JULES runs available through a publically-accessible data repository.

**Author Contributions**

G.H., C.H., E.C-P., A.H., P.C., T.P., J.H., W.C., J.L. and S.C. designed the IMOGEN runs. All authors contributed to the interpretation of the results and to the writing of or review of the paper. C.H. provided IMOGEN parameters calibrated against

the CMIP5 database, and E.C-P and C.H. led the development of the inverse IMOGEN model version. The following specific
contributions were also made: (a) E.B., S.C. and N.G.: code and expertise on permafrost, soil carbon and wetland methane
modelling, respectively; (b) A.H. and T.P.: land use change data; (c) W.C. and C.W.: ozone ancillary data; (d) D.P.vV. and
J.C.D.: IMAGE scenario data on land use, anthropogenic methane emissions and water consumption and withdrawals, and (e)
S.S.: expertise on the ozone damage effects.

**Competing interests**

The authors declare no competing interests.

**Acknowledgements**

The work was undertaken as part of the UK Natural Environment Research Council's programme "Understanding the
Pathways to and Impacts of a 1.5°C Rise in Global Temperature" through grants NE/P015050/1 CLIFFTOP (G.H., E.C-P,
S.C.), NE/P014909/1, MOC1.5 (W.C., C.W., J.L., C.H., P.C., S.S.) and NE/P014941/1 CLUES (P.C., A.H., T.P., J.H.). We
also acknowledge the support for: (a) G.H and E.C.P by NERC NE/N015746/1 The Global Methane Budget, MOYA; (b) A.H.
through her EPSRC Fellowship "Negative Emissions and the Food-Energy-Water Nexus" (EP/N030141/1); (c) A.H. by NERC
NE/P019951/1 FAB GGR, (d) W.C. from the Research Council of Norway, project no. 235548; (e) C.H. from CEH National
Capability Funding; (f) E.B.. from the Joint UK BEIS/Defra Met Office Hadley Centre Climate Programme (GA01101); (g)
E.B., D.P.vV. and J.C.D. from CRESCENDO (EU project 641816); and (h) NG from the Newton Fund through the Met Office
Climate Science for Service Partnership Brazil (CSSP Brazil). All authors acknowledge the CMIP5 database, and its outputs
from Earth System Models developed by climate research centres across the world. We also acknowledge Lars Kutzbach and
David Holl, who kindly provided the methane emission data for the Samoylov Island field site. We are grateful to the Editor
and the two anonymous reviewers, whose comments have helped to improve the clarity of the paper.

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

**Figures**

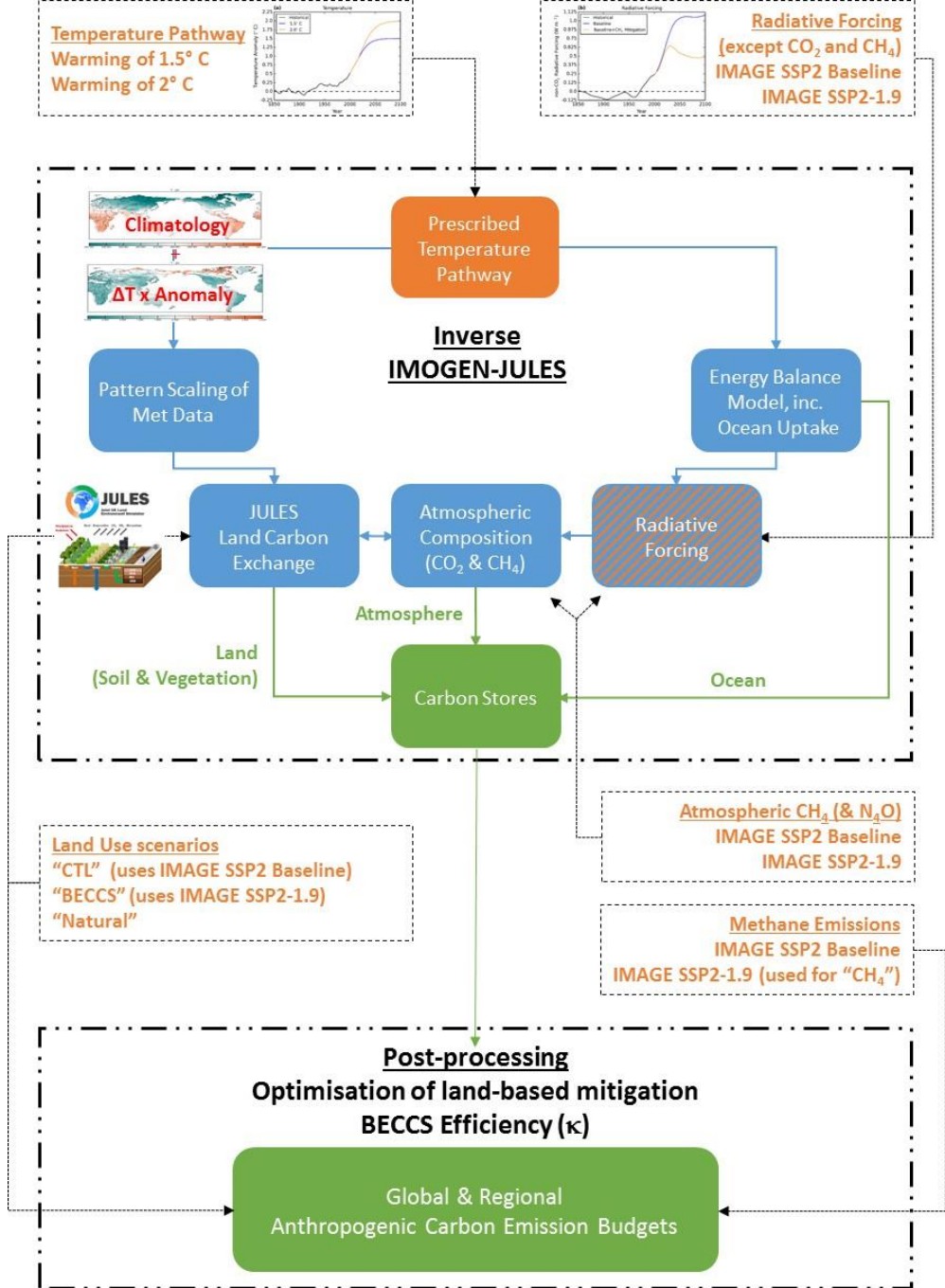

**Figure 1 | Schematic of the modelling approach and the workflow. The coloured boxes and text show (a) the key components of the**
**inverted IMOGEN-JULES model (blue), the prescribed and input data used in this study (orange) and the outputs (green).**

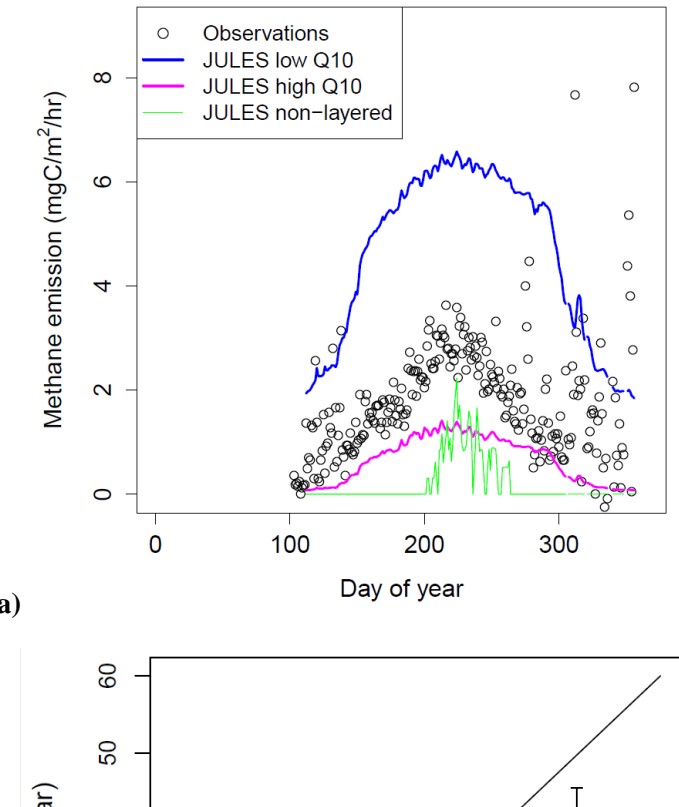

**(a)**

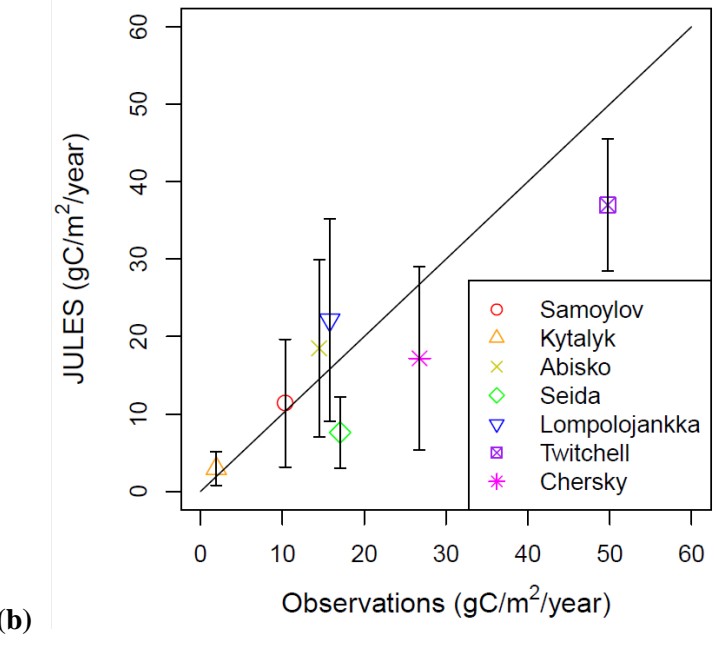

**(b)**

**Figure 2 | (a) Observed (circles) and modelled wetland methane emissions at the Samoylov Island field site. Modelled wetland methane emissions are shown for the standard JULES non-layered soil carbon configuration (green) and for the JULES layered soil carbon configurations with the low (blue line) and high (magenta line) $Q_{10}$ temperature sensitivities; the low $Q_{10}$ configuration gives higher methane emissions at high-latitude sites such as the Samoylov Island field site. The methane emission data is preliminary and was provided by Lars Kutzbach and David Holl. (b) Comparison of observed and modelled annual mean wetland $CH_4$ emission fluxes at a number of northern high-latitude and temperate sites. The error bars denote the lower and upper estimates from the low and high $Q_{10}$ simulations. The symbols represent the mean value between these estimates.**


(a) Time series of the prescribed temperature pathways

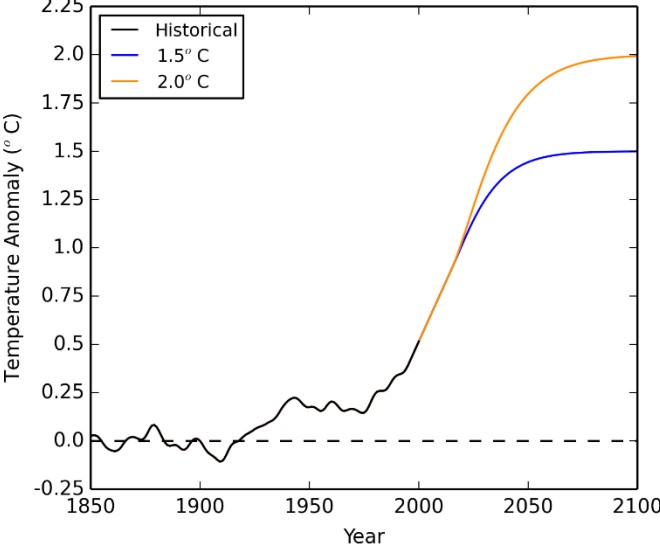


(b) Input time series of the input non-CO₂ radiative forcing

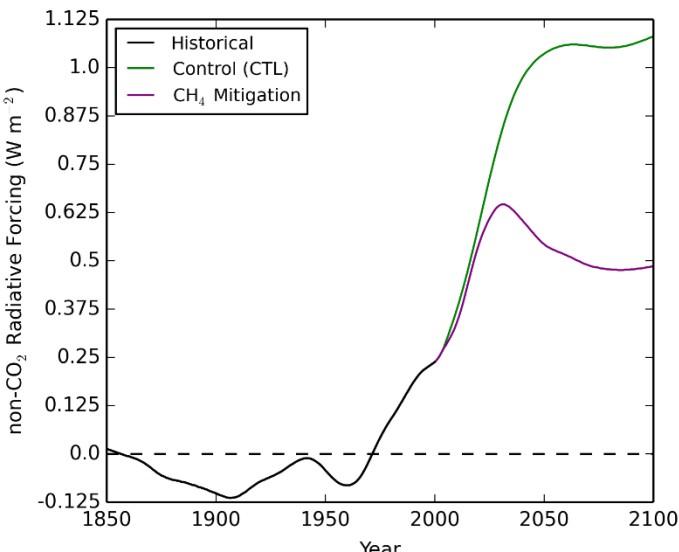



**Figure 3 | Time series of key datasets used in the study: (a) the historic temperature record (black) and the prescribed temperature**
**profiles used to represent warming of 1.5°C (blue) and 2°C (orange); (b) the historic (black) and the projected non-CO₂ greenhouse**
**gas radiative forcing (W m⁻²) for the control (greem) and methane mitigation (purple) scenarios.**

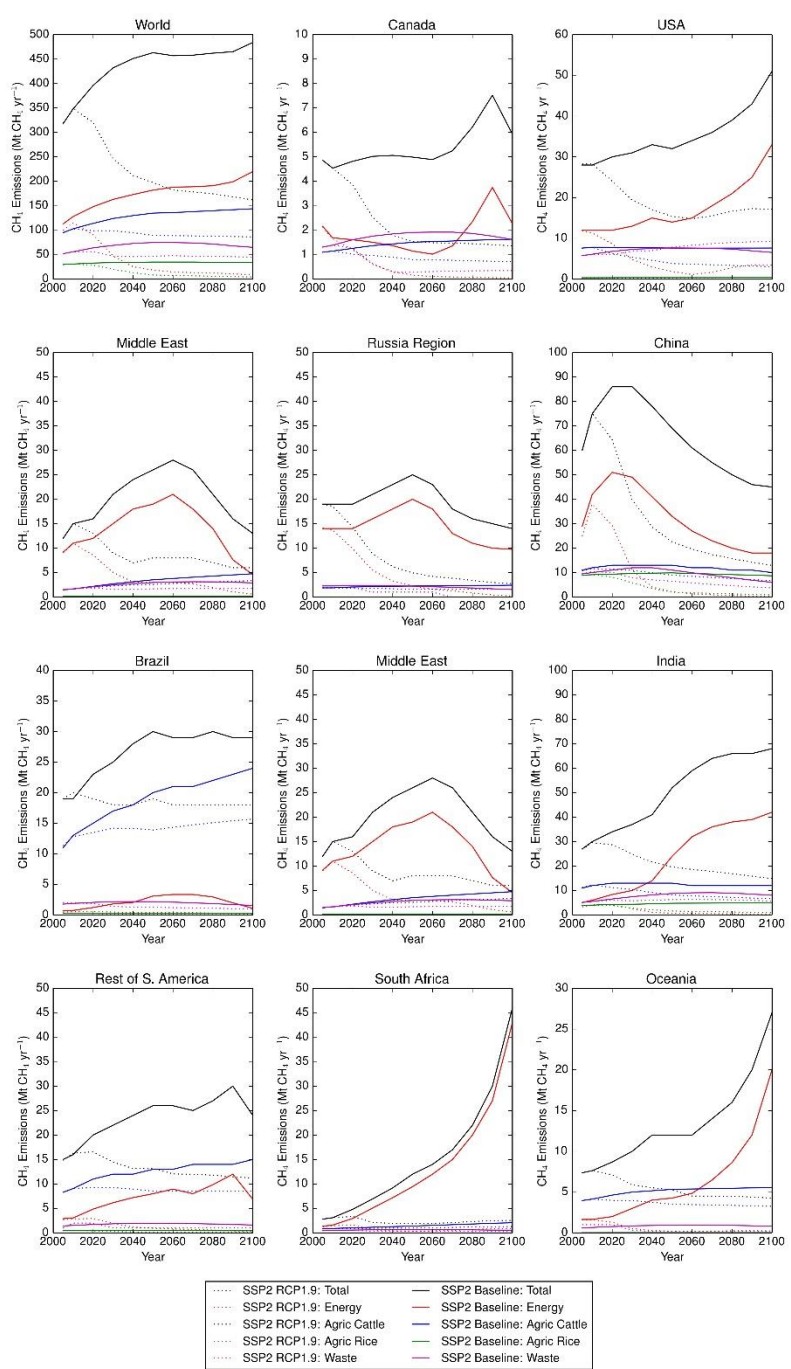


**Figure 4 | Time series of annual methane emissions between 2005 and 2100 from all and selected anthropogenic sources according to the IMAGE SSP2 Baseline (solid lines) and SSP2-RCP1.9 (dotted lines) scenarios, globally and for selected IMAGE regions, with total emissions in black, energy sector in red, agriculture-cattle in blue, agriculture-rice in green and waste in magenta. Note the y-axes have different scales for clarity.**




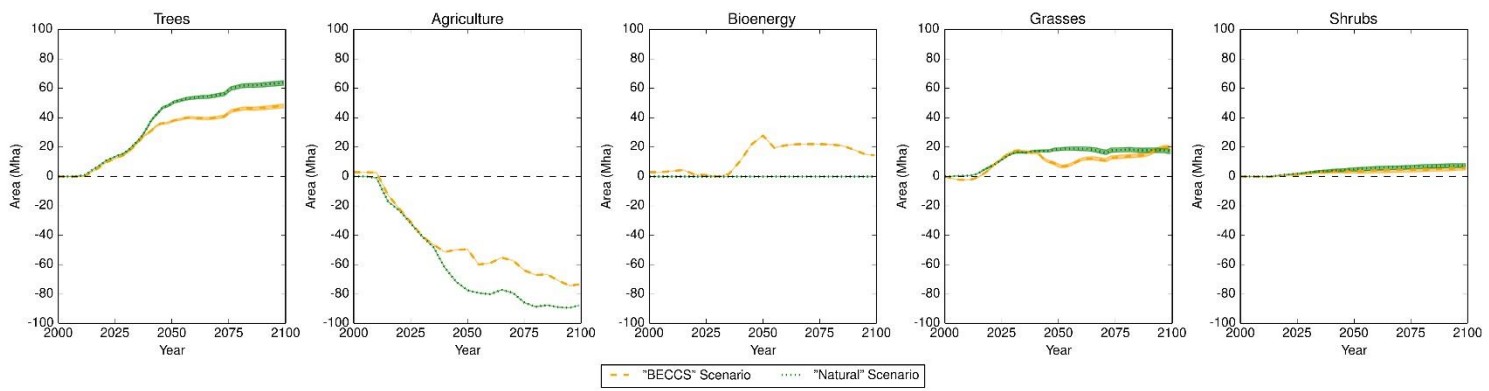



(b) IMAGE Russia Region

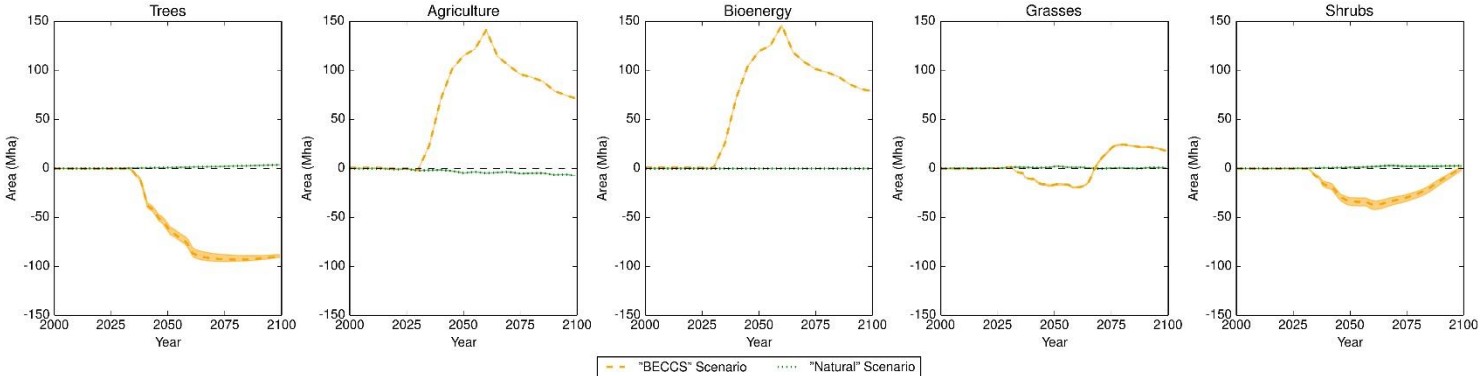



**Figure 5 | Time series of the land areas (in Mha) calculated for trees and prescribed for agriculture (including bioenergy crops) and bioenergy crops for the 'BECCS' (orange) and 'Natural, (green), as a difference to the baseline scenario (IM-BL), for Brazil (panel a) and the Russia (panel b) IMAGE regions between 2000 and 2100. The dotted lines are the median and the spread the interquartile range for the 34 GCMs emulated and 4 factorial sensitivity simulations.**


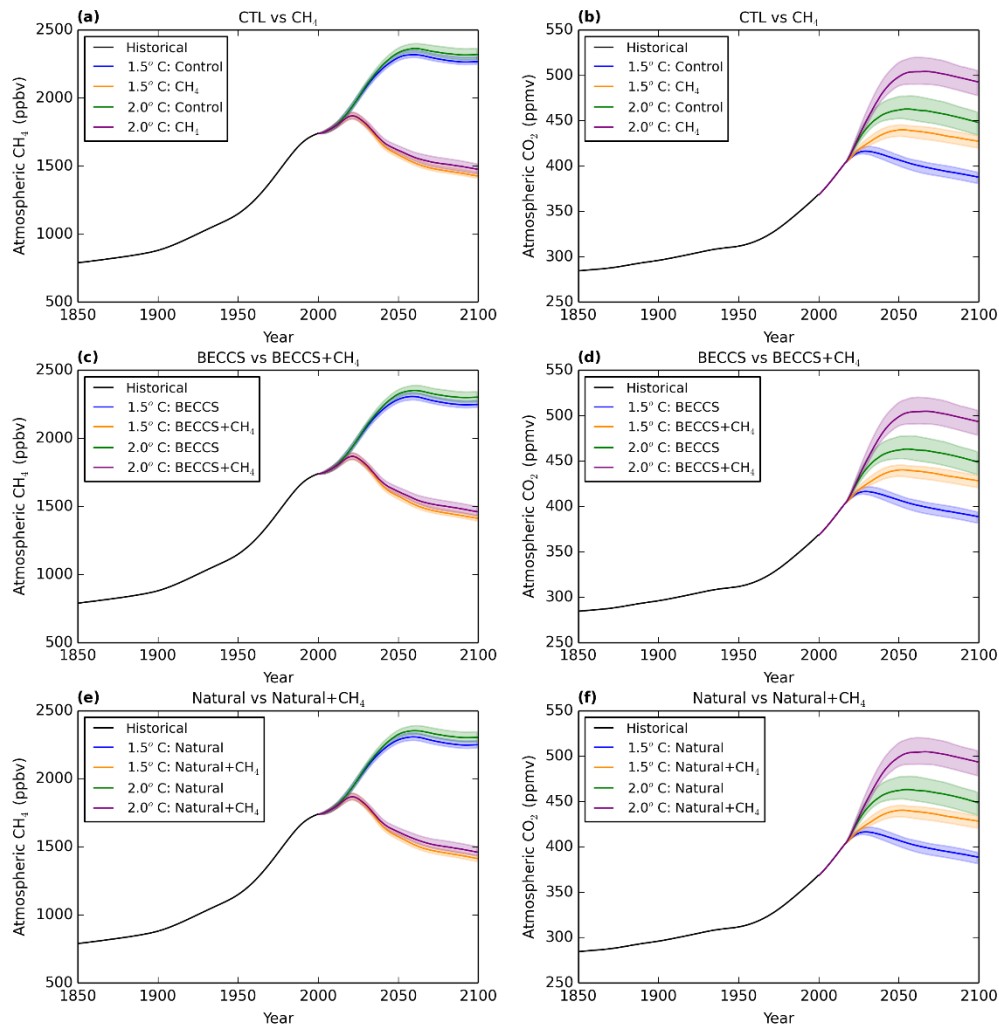


**Figure 6 | (a, c, e) Time series of the ensemble median atmospheric CH₄ concentrations (with interquartile range as spread) derived for each temperature profile for the scenarios: (a) "CTL" and "CH₄", (c) "BECCS" and "BECCS+CH₄", (e) "Natural" and "Natural+ CH₄". (d, f, h) show the corresponding time series for the atmospheric CO₂ concentrations.**


(a)

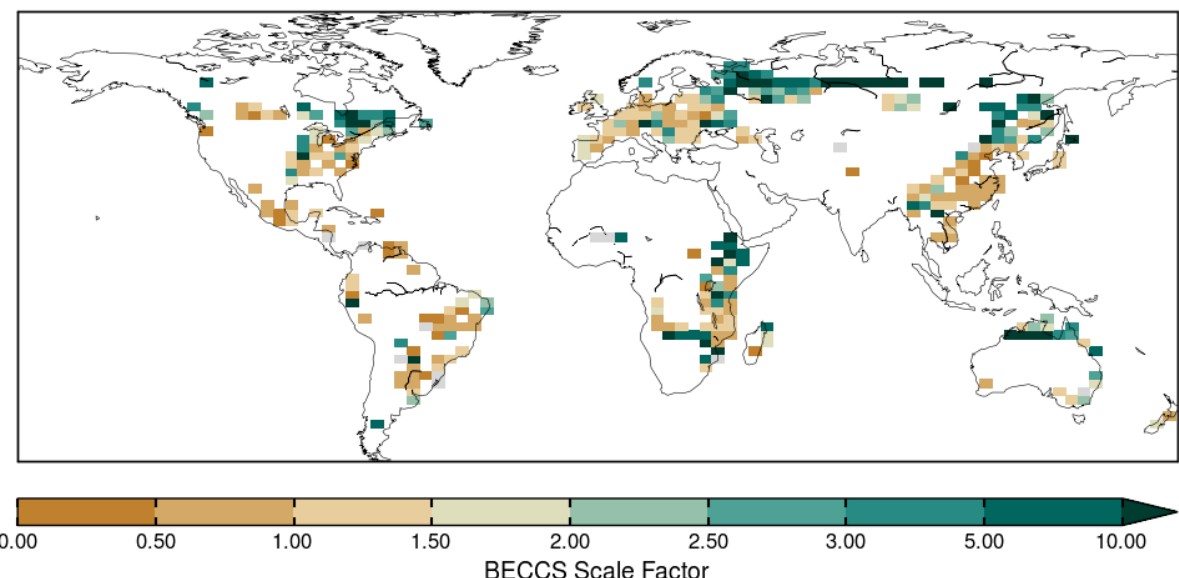


(b)

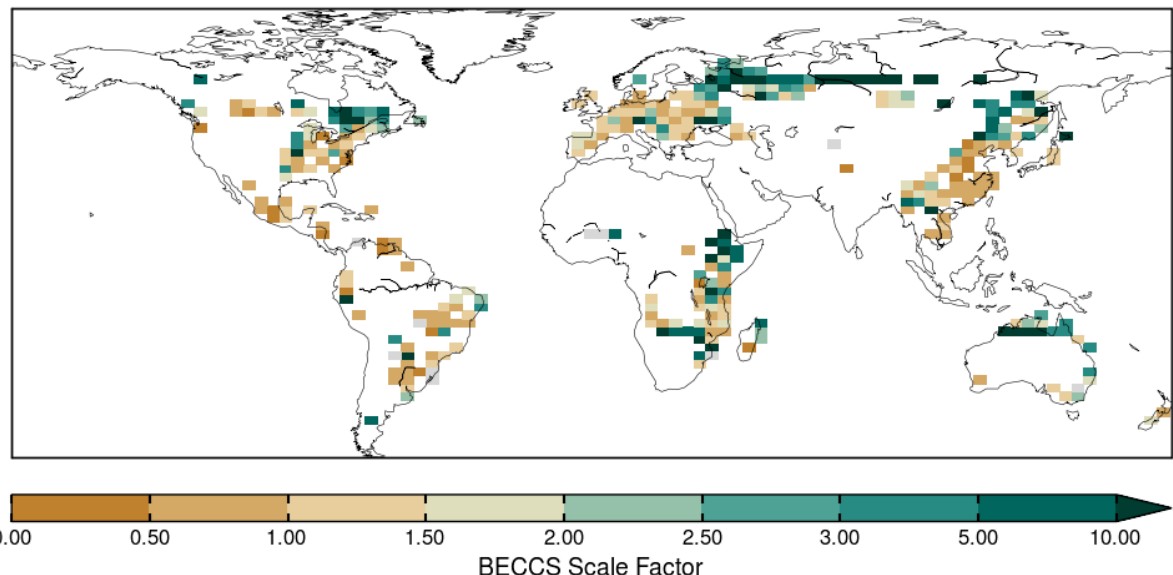


**Figure 7 | Scale factor required for BECCS to be the preferable mitigation option, as opposed to natural land carbon uptake. The**
**data represents the median of the 136 member ensemble for the optimised land-based mitigation simulation. Panel (a) is for**
**stabilisation at 1.5°C and panel (b) is for stabilisation at 2°C.**

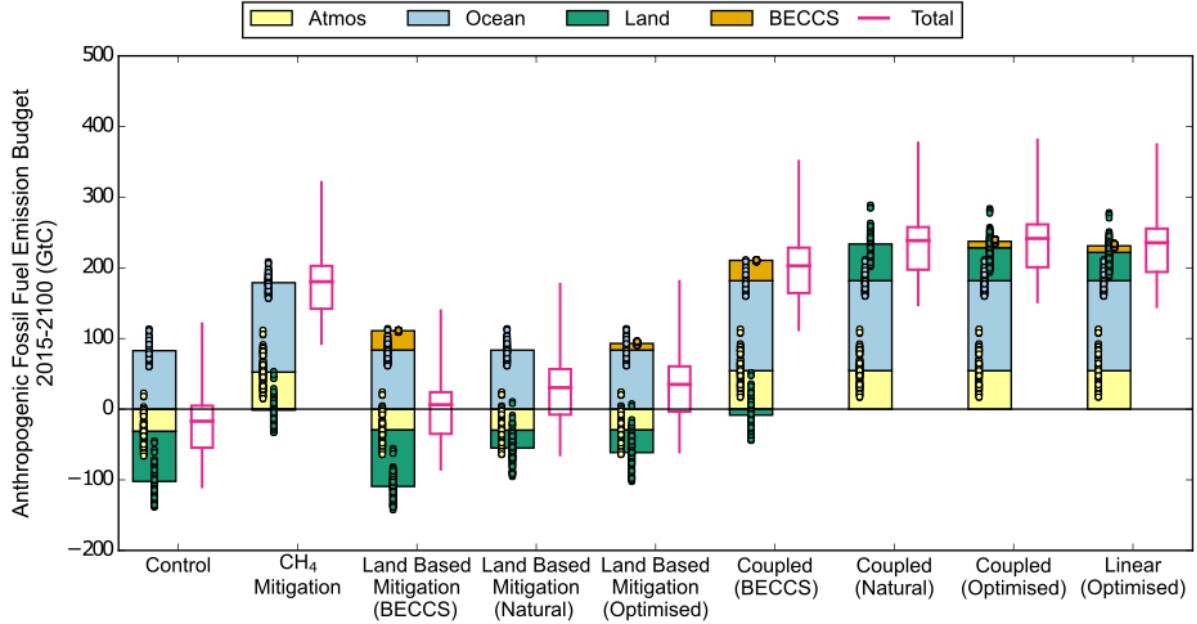



**Figure 8 | The contribution to the allowable anthropogenic fossil fuel emission budget (AFFEBs, GtC) from the changes in the**
**different carbon stores (atmosphere, ocean, land and BECCS) for the various control and mitigationscenarios, illustrated using the**
**temperature pathways for 1.5°C of warming. The bars are the median of the component 136-member ensembles, with the individual**
**members shown as points. The accompanying pink box and whiskers plots to the right of each set of bars are for the AFFEBs (as**
**the sum of the changes in the component carbon stores). The box and whisker plots show the median, interquartile range, minimum**
**and maximum derived of the resulting AFFEB ensemble. The optimised land based and coupled mitigation options selects the land**
**use option, which maximises the AFFEB for each model grid cell. Note that the land carbon store for the CH₄ scenario is at -1.4 GtC**
**(median of ensemble) is not visible, although the individual ensemble members can be seen as the green points.**

BECCS Scale Factor (κ) = 1

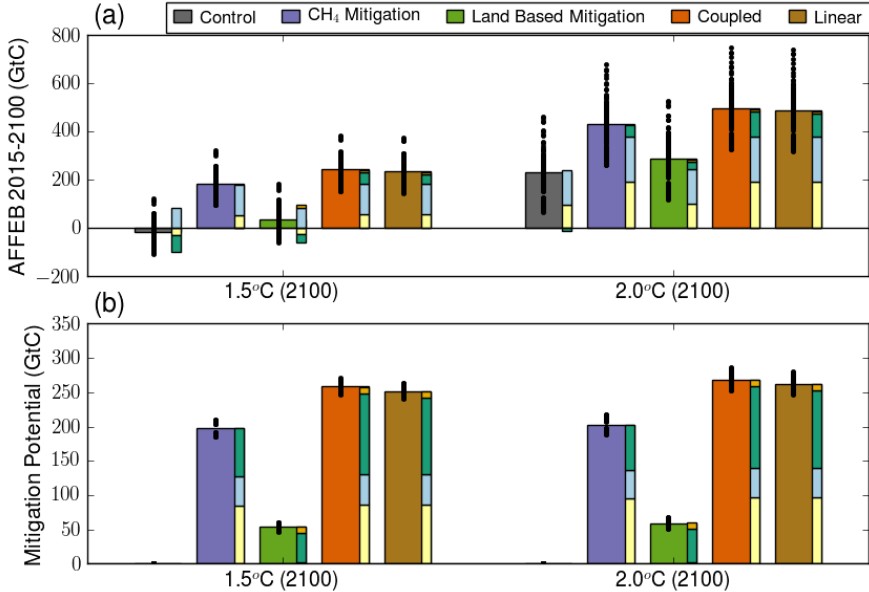


BECCS Scale Factor (κ) = 3

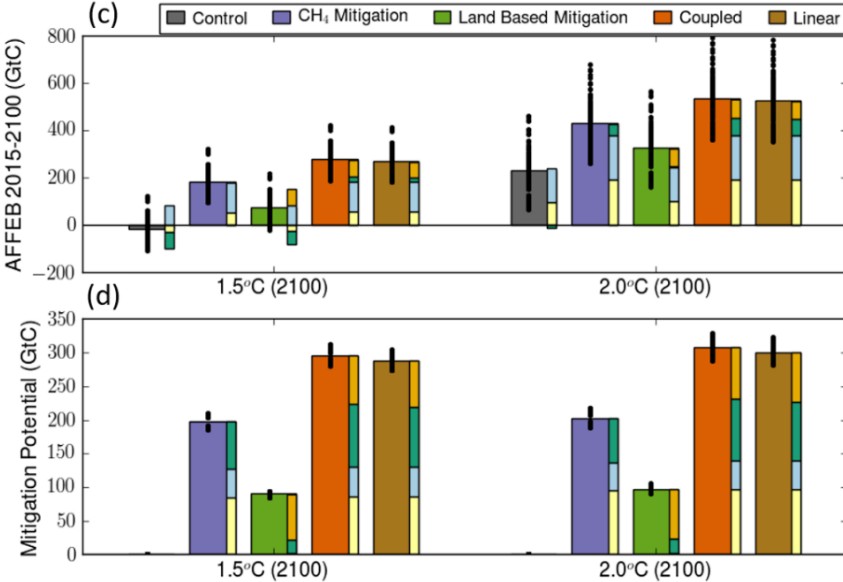


**Figure 9 | Panels (a & c): The allowable anthropogenic fossil fuel emission budgets (AFFEBs; GtC) for the control (grey), CH₄**
**mitigation (purple), land-based mitigation (green), coupled methane and land-based mitigation (orange) and the linearly summed**
**methane and land-based mitigation (brown), for 2 temperature pathways asymptoting at 1.5°C (left) and 2.0°C (right). (b & d) The**
**mitigation potential (GtC) as the increase in AFFEB from the corresponding control run. The breakdown of each AFFEB and**
**mitigation potential by the changes in the carbon stores is also shown: atmosphere (pale yellow), ocean (light blue), land (dark green)**
**and BECCS (gold) is included alongside each bar. Note that the land carbon store for the "CH₄" scenario at -1.4 GtC (median of**
ensemble) is not visible. There has however been a net increase in the land carbon store in this scenario when compared to the land
carbon store in the control run ( -70.8 GtC, median of ensemble).

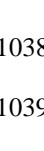

**Figure 10| (a) The total and component mitigation potential (GtC) for different mitigation options, involving methane and land use, as a function of the BECCS efficiency factor (κ, Sect. 2.4.3) for the temperature pathway reaching 1.5°C. The width of the lines represent the interquartile range of the 136-member ensembles. Maps of (b) the change of the modelled soil carbon (kg-C m$^{-2}$) between 2015 and 2099, as the difference between the scenario with BECCS and the natural land-management scenario; (c) the modelled mean bioenergy crop yield in the JULES simulations (κ = 1) and (d) the required bioenergy crop yield for BECCS to provide a larger carbon uptake than forest regrowth/afforestation (assuming κ = κ* and 87% efficiency of BECCS). Grid cells which do not exceed 1% BECCS cover for any year in the simulation are masked grey.**

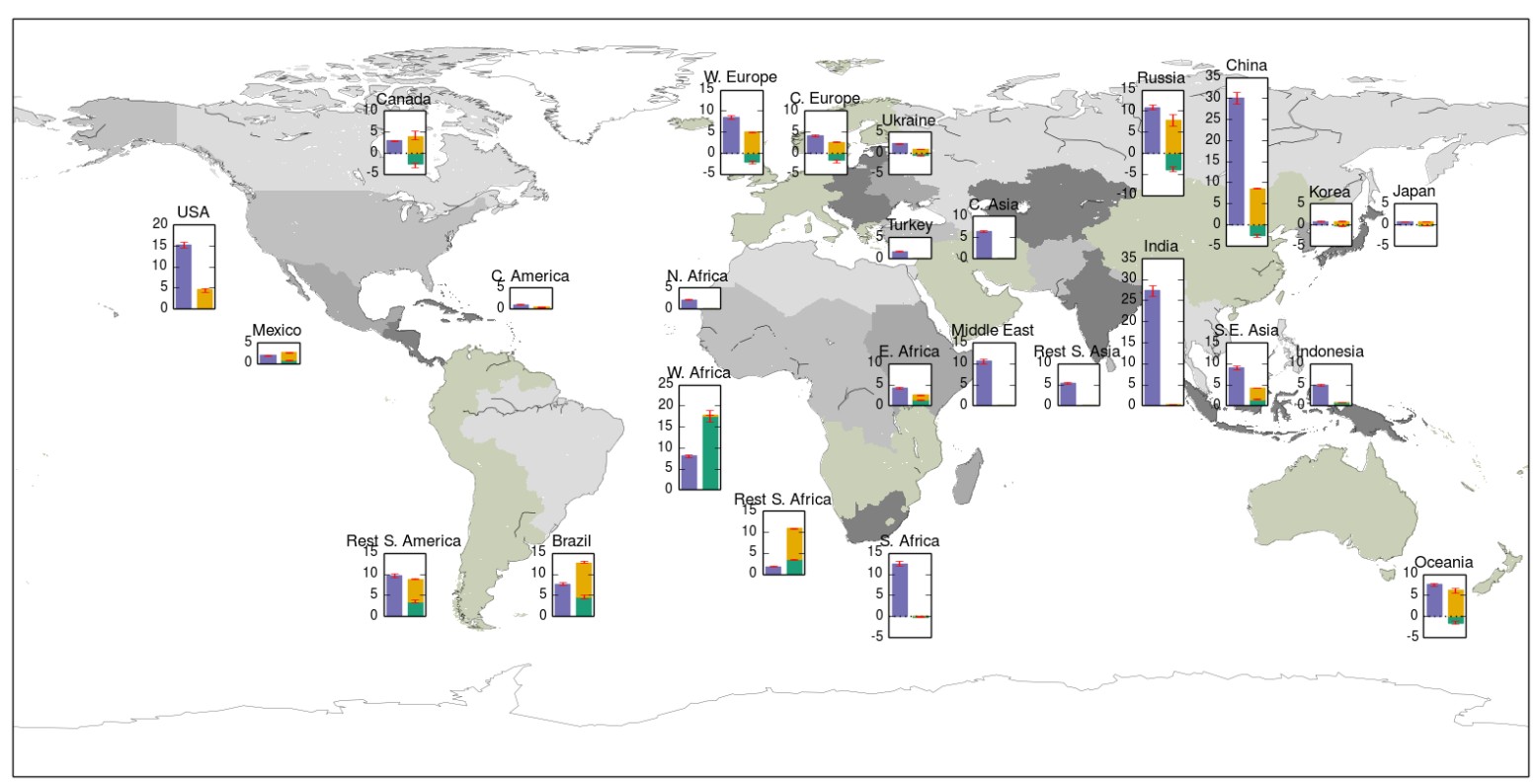

Figure 11 | The contribution to the allowable carbon emission budgets (GtC) between 2015 and 2100 for each of the 26 IMAGE IAM regions from methane mitigation (purple bars) and land-based mitigation options (green: natural land uptake; yellow: BECCS with κ = 3), for the temperature pathway stabilising at 1.5° warming without overshoot. The bars and error bars respectively show the median and the interquartile range, from the 136-member ensembles.

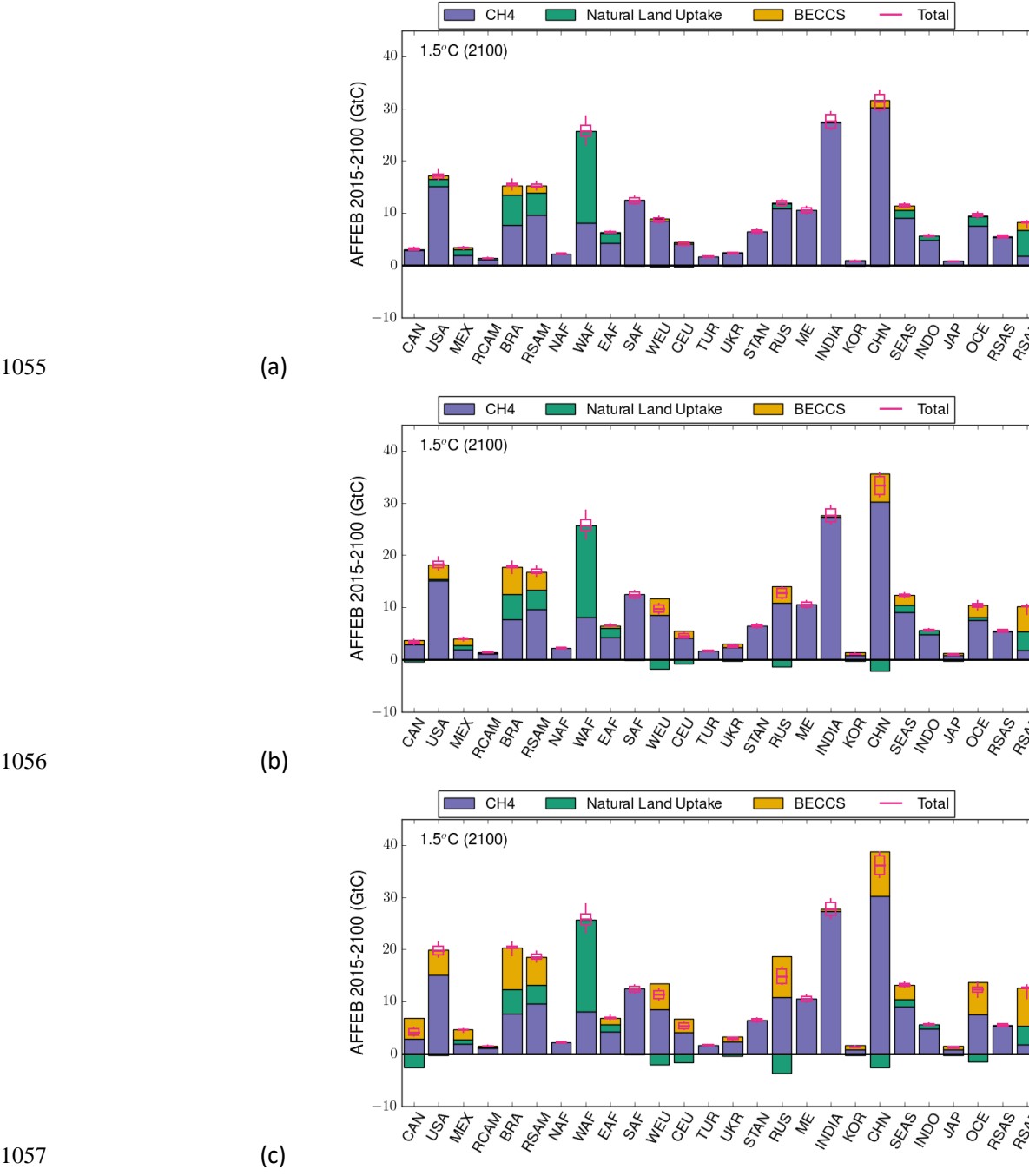

(a)

(b)

(c)


**Figure 12 | Contribution of different mitigation options to the increase in allowable anthropogenic fossil fuel emission budgets by**
**IMAGE region to meet the 1.5°C target. The stacked bars represent the median methane mitigation potential (purple bars) and**
**median land-based mitigation potential (natural land uptake, green; BECCS, brown). Panel (a) is based on a BECCS scaling factor**
**of unity, (b) a BECCS scaling factor of 2 and (c) a BECCS scaling factor of 3. The total (pink) shows the median and interquartile**
**range of the 136-member ensembles.**


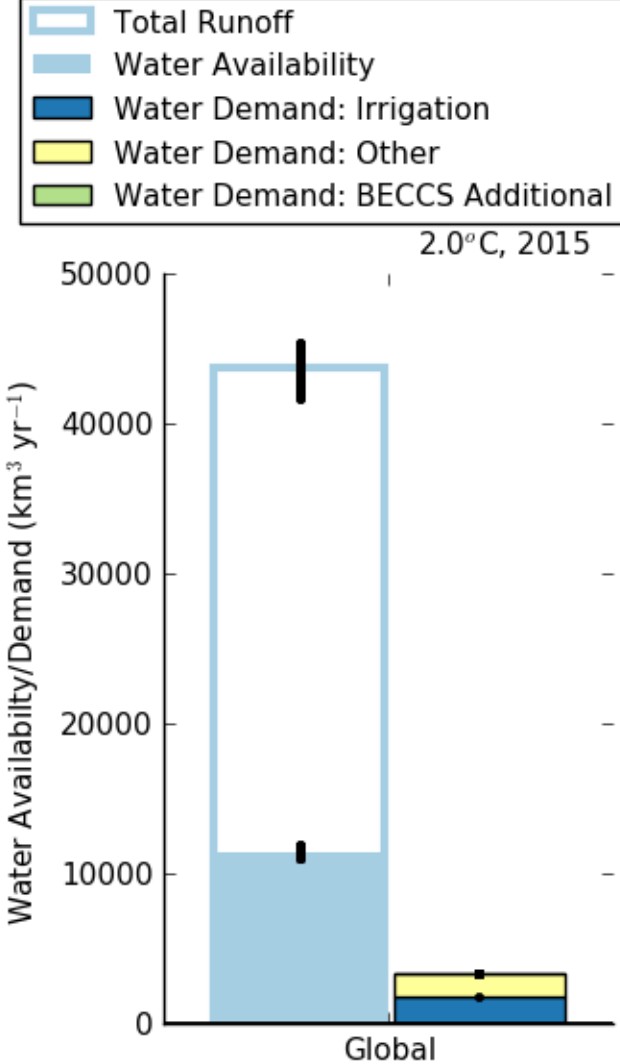


**Figure 13 | Global water availability (filled light blue bar) as a regionally dependent fraction of runoff (hollow light blue bar) for the**
**year 2015. The water demand for irrigation (dark blue) and for other uses (i.e., energy generation, industry and domestic; yellow),**
**are taken from the SSP2-RCP2.6-IMAGE database. Note there is very little BECCS additional water demand (green) in 2015.**



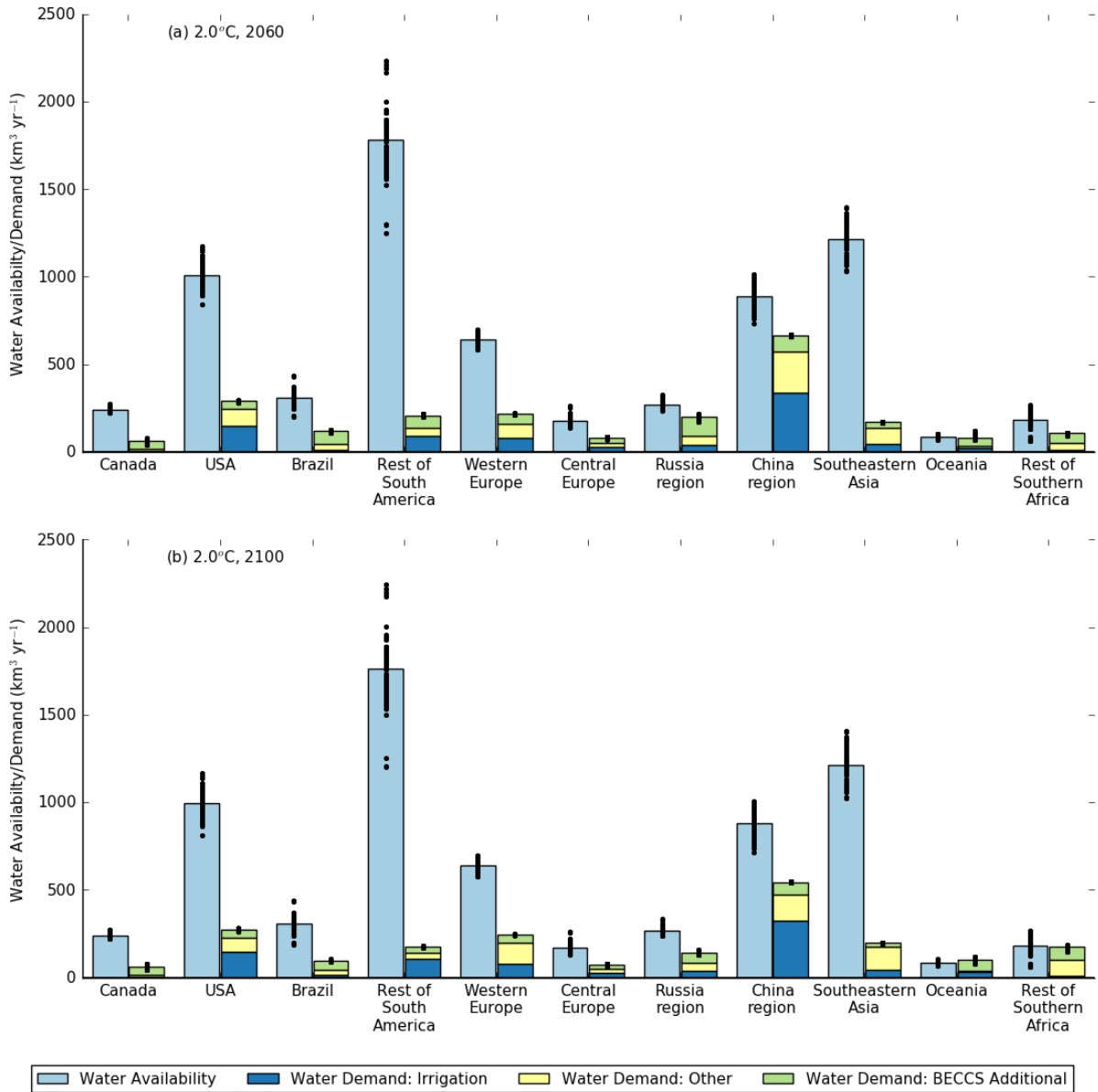


**Figure 14 | Water availability (light blue), SSP2-IMAGE water demand estimates for irrigation (dark blue), other uses (i.e., energy**
**generation, industry and domestic; yellow) and the additional water demand from BECCS (green) for the years 2059-2060 and 2099-**
**2100 for the 2.0°C warming target, with a BECCS κ factor of 3. The points are the individual results from the 136-member ensembles,**
**while the bars are the corresponding median values of the ensembles.**

**Tables**

**Table 1 | The IMOGEN-JULES and post processing scenario runs, key features and the input and prescribed datasets used in the scenarios.**
(a) IMOGEN-JULES modelling scenarios (Note 1)

| | **Scenario (Abbreviation)** **Key features of the Scenario** | **Scenario-specific input and prescribed datasets (Notes 2, 3)** |
|---|---|---|
| 1. | **Control ("CTL")** <br> • Agricultural land accrued to feed growing populations associated with the SSP2 pathway <br> • No deployment of BECCS <br> • Anthropogenic $CH_4$ emissions rise from 318 Tg yr$^{-1}$ in 2005 to 484 Tg yr$^{-1}$ in 2100 <br> • Effects of the methane and carbon-climate feedbacks from wetlands and permafrost thaw included | Scenario-specific input data <br> • Time series of radiative forcing by non-$CO_2$ GHG and other non-$CO_2$ climate forcers, for the IMAGE SSP2 baseline scenario <br> • Time series of annual global atmospheric concentrations of $CH_4$ and $N_2O$ for the IMAGE SSP2 baseline scenario <br> Scenario-specific prescribed data <br> • Gridded annual time series of areas assigned to agriculture (crops & pasture), for the IMAGE SSP2 baseline scenario, converted into fractions of the IMOGEN-JULES grid cell |
| 2. | **Methane mitigation ("CH₄")** <br> • Agricultural land-use as in Control ("CTL") scenario <br> • Anthropogenic $CH_4$ emissions decline from 318 Tg yr$^{-1}$ in 2005 to 162 Tg yr$^{-1}$ in 2100, from the IMAGE SSP2 RCP1.9 scenario <br> • Effects of the methane and carbon-climate feedbacks from wetlands and permafrost thaw included | Scenario-specific input data <br> • Time series of radiative forcing by non-$CO_2$ GHG and other non-$CO_2$ climate forcers, for the IMAGE SSP2 RCP1.9 scenario <br> • Time series of annual global atmospheric concentrations of $CH_4$ and $N_2O$ for the IMAGE SSP2 RCP1.9 scenario <br> Scenario-specific prescribed data <br> • As 1, gridded annual time series of area assigned to agriculture (crops & pasture). Converted into fractions of the IMOGEN-JULES grid cell |
| 3. | **Land-based mitigation, including BECCS ("BECCS")** <br> • Land use change based on the IMAGE SSP2 RCP1.9 scenario <br> • High levels of REDD and full reforestation <br> • Food-first policy so that bioenergy crops (BE) are only implemented on land not required for food production <br> • Anthropogenic $CH_4$ emissions as in Control ("CTL") scenario <br> • Effects of the methane and carbon-climate feedbacks from wetlands and permafrost thaw included | Scenario-specific input data <br> • Time series of radiative forcing by non-$CO_2$ GHG and other non-$CO_2$ climate forcers, for the IMAGE SSP2 baseline scenario <br> • Time series of annual global atmospheric concentrations of $CH_4$ and $N_2O$ for the IMAGE SSP2 baseline scenario (as used in "CTL") <br> Scenario-specific prescribed data <br> • Gridded annual time series of areas assigned to agriculture (crops & pasture) and within that the area for bioenergy crops, for the IMAGE SSP2 RCP1.9 scenario. Converted into a fraction of the IMOGEN-JULES grid cell |
| 4. | **Land-based mitigation with no BECCS ("Natural")** <br> • Land use as 3, except any land area allocated to bioenergy crops is set to zero, allowing expansion of natural vegetation <br> • Anthropogenic $CH_4$ emissions as in Control ("CTL") scenario <br> • Effects of the methane and carbon-climate feedbacks from wetlands and permafrost thaw included | Scenario-specific input data <br> • Time series of radiative forcing by non-$CO_2$ GHG and other non-$CO_2$ climate forcers, for the IMAGE SSP2 baseline scenario <br> • Time series of annual global atmospheric concentrations of $CH_4$ and $N_2O$ for the IMAGE SSP2 baseline scenario (as used in "CTL") <br> Scenario-specific prescribed data <br> • Gridded annual time series of areas assigned to agriculture (crops & pasture). As 3, except any land allocated to bioenergy crops is set to zero. Converted into a fraction of the IMOGEN-JULES grid cell |

| | | |
|---|---|---|
| 5. | **Combined methane & land-based mitigation** <br> **"Coupled(CH$_4$+BECCS)"** <br> • Combines CH$_4$ mitigation of 2 with land-based mitigation scenario of 3 | Scenario-specific input data <br> • As 2, time series of radiative forcing by non-CO$_2$ GHG and other non-CO$_2$ climate forcers, for the IMAGE SSP2 RCP1.9 scenario <br> • As 2, time series of annual global atmospheric concentrations of CH$_4$ and N$_2$O for the IMAGE SSP2 RCP1.9 scenario <br> Scenario-specific prescribed data <br> • As 3, gridded annual time series of areas assigned to agriculture (crops & pasture) and within that the area for bioenergy crops, for the IMAGE SSP2 RCP1.9 scenario. Converted into prescribed fractions of the IMOGEN-JULES grid cell |
| 6. | **Combined methane & land-based mitigation with no BECCS** <br> **"Coupled (CH$_4$+Natural)"** <br> • Combines CH$_4$ mitigation of 2 with land-based mitigation scenario of 4 | Scenario-specific input data <br> • As 2, time series of radiative forcing by non-CO$_2$ GHG and other non-CO$_2$ climate forcers, for the IMAGE SSP2 RCP1.9 scenario <br> • As 2, time series of annual global atmospheric concentrations of CH$_4$ and N$_2$O for the IMAGE SSP2 RCP1.9 scenario <br> Scenario-specific prescribed data <br> • As 4, gridded annual time series of areas assigned to agriculture (crops & pasture). Converted into a fraction of the IMOGEN-JULES grid cell |

(b) Post-processing scenarios (Note 1)

| | **Scenario** <br> **"Abbreviation"** | **Description of the Scenario** |
|---|---|---|
| **7.** | **Optimisation of land-based mitigation** <br> **"Land-based mitigation: Optimised"** | • Optimisation of scenarios 3 and 4 by selecting the scenario which has the larger carbon uptake, on a grid cell by grid cell basis |
| 8. | **Optimisation of the combined methane & land-based mitigation** <br> **"Coupled Optimised"** | • Optimisation of scenarios 5 and 6 by selecting the scenario which has the larger carbon uptake, on a grid cell by grid cell basis |

**Notes**
1. Each scenario comprises two 136-member ensembles (34 GCMs x 2 ozone damage sensitivities x 2 methanogenesis Q$_{10}$ temperature sensitivities), one for the 1.5°C warming target
and the second for the 2°C warming target.
2. All of the above scenarios also use time series of (1) observed temperature changes between 1850 and 2015; (2) profiles of temperature change between 2015 and 2100 to achieve the
1.5°C and the 2°C warming targets; and (3) the radiative forcing changes of non-CO$_2$ radiative forcing between 1850 and 2015.
3. We define (a) a "prescribed" dataset as one that is used unchanged in the IMOGEN-JULES modelling; (b) an "input" dataset as one that provides the initial values that are subsequently
changed.

**Table 2 | IMAGE regions, the maximum area of BECCS deployed (Mha) and the main differences in land use between the BECCS**
**and Natural scenarios.**

| Region | Abbreviation | Max. area of bioenergy crops (Mha) | Main land-use difference between BECCS and Natural scenarios |
|---|---|---|---|
| Canada | CAN | 65.9 | Forest to BECCS in BECCS scenario |
| USA | USA | 39.0 | Agricultural land and forest to BECCS (BECCS). Agricultural land to forest (Natural) |
| Mexico | MEX | 7.1 | Agricultural land to BECCS and forest (BECCS). Agricultural land to forest (Natural) |
| Central America | RCAM | 0.5 | Little BECCS. Agricultural land to forests in both scenarios. |
| Brazil | BRA | 27.8 | Agricultural land to BECCS and forest (BECCS). Agricultural land to forest (Natural) |
| Rest of South America | RSAM | 20.3 | Agricultural land to BECCS and forest (BECCS). Agricultural land to forest (Natural) |
| Northern Africa | NAF | 0.0 | No BECCS. No real differences between scenarios |
| Western Africa | WAF | 3.1 | Little BECCS. Agricultural land to forests in both scenarios. |
| Eastern Africa | EAF | 33.9 | Agricultural land to BECCS and forest (BECCS). Agricultural land to forest (Natural) |
| South Africa | SAF | 1.0 | Little BECCS. Agricultural land to forests in both scenarios. |
| Rest of Southern Africa | RSAF | 63.7 | Agricultural land to BECCS and forest (BECCS). Agricultural land to forest (Natural) |
| Western Europe | WEU | 23.6 | Forest to BECCS in BECCS scenario |
| Central Europe | CEU | 19.3 | Forest to BECCS in BECCS scenario |
| Turkey | TUR | 0.0 | No BECCS. No real differences between scenarios |
| Ukraine Region | UKR | 11.4 | Forest to BECCS in BECCS scenario |
| Central Asia | STAN | 0.7 | Little BECCS. No real differences between scenarios |
| Russia Region | RUS | 146.1 | Forest to BECCS in BECCS scenario |
| Middle East | ME | 0.0 | No BECCS. No real differences between scenarios |
| India | INDIA | 6.0 | Forest to BECCS in BECCS scenario |
| Korea Region | KOR | 4.3 | Forest to BECCS in BECCS scenario |
| China | CHN | 58.1 | Forest to BECCS in BECCS scenario |
| South East Asia | SEAS | 24.5 | Forest to BECCS in BECCS scenario. Agricultural land to forest (Natural) |
| Indonesia | INDO | 0.0 | No BECCS. Agricultural land to forests in both scenarios. |
| Japan | JAP | 2.7 | Forest to BECCS in BECCS scenario |
| Rest of South Asia | RSAS | 0.0 | No BECCS. No real differences between scenarios |
| Oceania | OCE | 78.7 | Forest to BECCS in BECCS scenario |



**Table 3 | For the 1.5°C temperature profile, the mean of the 34-GCM member ensembles for the "CTL" and mitigation scenarios for the different factorial runs (low $Q_{10}$/low $O_3$, low**
**$Q_{10}$/high $O_3$, high $Q_{10}$/low $O_3$ and high $Q_{10}$/high $O_3$), the standard deviation of the full 136-member ensemble (GtC), the derived standard deviations for land processes ($\sigma_{land}$) and**
**climate ($\sigma_{climate}$, as represented by the 34 GCMs) and the ratio of $\sigma_{climate}/\sigma_{land}$ for (a) the Anthropogenic Fossil Field $CO_2$ Emission Budgets and (b) the Mitigation Potential (= scenario**
**– CTL).**
(1) AFFEB

| Scenario | Mean of 34-member Factorial Run (GtC) | | | | Standard Deviation (GtC) | | | Ratio $\sigma_{climate}:\sigma_{land}$ |
|---|---|---|---|---|---|---|---|---|
| | Low $Q_{10}$ Low $O_3$ | Low $Q_{10}$ High $O_3$ | High $Q_{10}$ Low $O_3$ | High $Q_{10}$ High $O_3$ | 136-member Ensemble | Land $\sigma_{land}$ | Climate $\sigma_{climate}$ | |
| CTL | -9.66 | -20.58 | -18.91 | -31.06 | 47.12 | 7.60 | 46.50 | 6.12 |
| CH₄ | 179.44 | 186.79 | 168.73 | 174.90 | 47.54 | 6.59 | 47.08 | 7.14 |
| BECCS | 6.49 | 3.42 | -2.09 | -5.80 | 47.45 | 4.76 | 47.21 | 9.91 |
| Natural | 42.57 | 24.60 | 35.00 | 16.05 | 48.95 | 10.07 | 47.90 | 4.75 |
| Optimised Land-based | 46.42 | 29.18 | 37.89 | 20.00 | 48.85 | 9.84 | 47.85 | 4.86 |
| Linear BECCS+CH₄ | 195.58 | 210.79 | 185.55 | 200.15 | 48.64 | 9.07 | 47.79 | 5.27 |
| Linear_Natural+CH₄ | 231.67 | 231.97 | 222.64 | 222.00 | 48.70 | 4.76 | 48.47 | 10.19 |
| Linear optimised | 235.51 | 236.55 | 225.53 | 225.96 | 48.69 | 5.16 | 48.42 | 9.39 |
| Coupled BECCS+CH₄ | 199.69 | 214.62 | 189.50 | 203.94 | 48.48 | 9.01 | 47.64 | 5.29 |
| Coupled Natural+CH₄ | 237.83 | 238.95 | 228.72 | 228.91 | 48.60 | 4.80 | 48.36 | 10.07 |
| Coupled optimised | 241.50 | 243.29 | 231.35 | 232.60 | 48.60 | 5.27 | 48.31 | 9.17 |

(2) Mitigation Potential

| Scenario | Mean of 34-member Factorial Run (GtC) | | | | Standard Deviation (GtC) | | | Ratio $\sigma_{climate}:\sigma_{land}$ |
|---|---|---|---|---|---|---|---|---|
| | Low $Q_{10}$ Low $O_3$ | Low $Q_{10}$ High $O_3$ | High $Q_{10}$ Low $O_3$ | High $Q_{10}$ High $O_3$ | 136-member Ensemble | Land $\sigma_{land}$ | Climate $\sigma_{climate}$ | |
| CTL | - | - | - | - | - | - | - | - |
| CH₄ | 189.10 | 207.37 | 187.64 | 205.96 | 9.28 | 9.18 | 1.39 | 0.15 |
| BECCS | 16.14 | 24.01 | 16.82 | 25.26 | 4.24 | 4.11 | 1.05 | 0.26 |
| Natural | 52.23 | 45.18 | 53.91 | 47.11 | 3.93 | 3.58 | 1.62 | 0.45 |
| Optimised Land-based | 56.07 | 49.76 | 56.80 | 51.06 | 3.44 | 3.06 | 1.57 | 0.51 |
| Linear BECCS+CH₄ | 205.24 | 231.38 | 204.46 | 231.21 | 13.39 | 13.23 | 2.09 | 0.16 |
| Linear_Natural+CH₄ | 241.33 | 252.55 | 241.55 | 253.06 | 6.14 | 5.69 | 2.32 | 0.41 |
| Linear optimised | 245.17 | 257.13 | 244.44 | 257.02 | 6.55 | 6.14 | 2.28 | 0.37 |
| Coupled BECCS+CH₄ | 209.34 | 235.20 | 208.41 | 235.00 | 13.27 | 13.12 | 2.01 | 0.15 |
| Coupled Natural+CH₄ | 247.48 | 259.54 | 247.63 | 259.97 | 6.49 | 6.10 | 2.21 | 0.36 |
| Coupled optimised | 251.15 | 263.87 | 250.26 | 263.66 | 6.89 | 6.54 | 2.17 | 0.33 |


**Table 4a | Comparison by IMAGE region of the modelled available water (km³ yr⁻¹), the projected water withdrawals (km³ yr⁻¹) for irrigation and for other anthropogenic activities**
**(energy generation, industry, domestic) from the IMAGE SSP2-RCP2.6 scenario, and the additional water required for BECCS (km³ yr⁻¹ and as percentages of the net available water**
**and of the water withdrawals for irrigation and other), for the year 2060. The percentage of runoff available for human use by IMAGE region is also included.**

| Region | Abbreviation | % of Regional Runoff Available | Available Water (km³ yr⁻¹) | Water Demand | | | Total Demand as % of Available Water | BECCS Demand as % of Total Demand |
|---|---|---|---|---|---|---|---|---|
| | | | | Irrigation (km³ yr⁻¹) | Other (km³ yr⁻¹) | BECCS (km³ yr⁻¹) | | |
| Canada | CAN | 40% | 243.19 | 3.39 | 14.21 | 44.45 | 25.5% | 71.6% |
| USA | USA | 5% | 1,010.82 | 149.55 | 96.07 | 44.55 | 28.7% | 15.4% |
| Mexico | MEX | 5% | 75.89 | 76.58 | 25.56 | 24.48 | 166.8% | 19.3% |
| Central America | RCAM | 5% | 185.92 | 8.16 | 15.49 | 2.28 | 13.9% | 8.8% |
| Brazil | BRA | 40% | 310.65 | 12.24 | 34.44 | 73.12 | 38.6% | 61.0% |
| Rest of South America | RSAM | 5% | 1,779.42 | 93.50 | 46.49 | 67.66 | 11.7% | 32.6% |
| Northern Africa | NAF | 5% | 0.11 | 61.60 | 54.63 | 0.00 | - | - |
| Western Africa | WAF | 5% | 1,962.47 | 28.29 | 118.83 | 0.39 | 7.5% | 0.3% |
| Eastern Africa | EAF | 5% | 485.18 | 53.92 | 63.10 | 2.45 | 24.6% | 2.1% |
| South Africa | SAF | 5% | 0.60 | 13.45 | 9.28 | 0.48 | 3868.3% | 2.1% |
| Rest of Southern Africa | RSAF | 5% | 182.48 | 10.03 | 41.36 | 56.02 | 58.9% | 52.2% |
| Western Europe | WEU | 5% | 642.34 | 78.72 | 82.01 | 56.22 | 33.8% | 25.9% |
| Central Europe | CEU | 5% | 176.27 | 27.46 | 22.32 | 29.68 | 45.1% | 37.4% |
| Turkey | TUR | 5% | 29.98 | 60.35 | 15.86 | 0.00 | - | - |
| Ukraine Region | UKR | 5% | 67.47 | 11.73 | 25.90 | 12.28 | 74.0% | 24.6% |
| Central Asia | STAN | 5% | 20.57 | 88.26 | 32.62 | 0.00 | - | - |
| Russia Region | RUS | 40% | 270.32 | 42.30 | 51.60 | 103.87 | 73.2% | 52.5% |
| Middle East | ME | 5% | 8.65 | 149.55 | 40.97 | 0.00 | - | - |
| India | INDIA | 5% | 319.36 | 374.18 | 501.06 | 0.00 | - | - |
| Korea Region | KOR | 5% | 42.85 | 6.20 | 9.75 | 12.64 | 66.7% | 44.2% |
| China | CHN | 5% | 887.26 | 338.81 | 236.89 | 87.73 | 74.8% | 13.2% |
| South East Asia | SEAS | 5% | 1,212.00 | 46.52 | 92.99 | 31.56 | 14.1% | 18.4% |
| Indonesia | INDO | 5% | 1,293.05 | 8.18 | 113.87 | 0.00 | - | - |
| Japan | JAP | 5% | 209.49 | 2.79 | 18.99 | 7.69 | 14.1% | 26.1% |
| Rest of South Asia | RSAS | 5% | 74.57 | 259.95 | 154.42 | 0.00 | - | - |
| Oceania | OCE | 5% | 85.46 | 24.99 | 8.91 | 48.06 | 95.9% | 58.6% |



**Table 4b | As Table 4a for 2100**

| Region | Abbreviation | % of Regional Runoff Available | Available Water (km³ yr⁻¹) | Water Demand | | | Total Demand as % of Available Water | BECCS Demand as % of Total Demand |
|---|---|---|---|---|---|---|---|---|
| | | | | Irrigation (km³ yr⁻¹) | Other (km³ yr⁻¹) | BECCS (km³ yr⁻¹) | | |
| Canada | CAN | 40% | 240.14 | 4.31 | 11.72 | 45.21 | 25.5% | 73.8% |
| USA | USA | 5% | 993.09 | 148.57 | 81.35 | 45.45 | 27.7% | 16.5% |
| Mexico | MEX | 5% | 72.79 | 77.27 | 23.78 | 11.14 | 154.1% | 9.9% |
| Central America | RCAM | 5% | 182.12 | 8.74 | 13.96 | 0.66 | 12.8% | 2.8% |
| Brazil | BRA | 40% | 307.53 | 12.31 | 30.80 | 54.89 | 31.9% | 56.0% |
| Rest of South America | RSAM | 5% | 1,765.14 | 103.97 | 38.34 | 32.65 | 9.9% | 18.7% |
| Northern Africa | NAF | 5% | 0.11 | 57.89 | 56.98 | 0.00 | - | - |
| Western Africa | WAF | 5% | 1,953.10 | 37.23 | 262.07 | 0.62 | 15.4% | 0.2% |
| Eastern Africa | EAF | 5% | 485.02 | 58.96 | 128.33 | 20.54 | 42.8% | 9.9% |
| South Africa | SAF | 5% | 0.60 | 13.43 | 7.50 | 0.45 | 3563.3% | 2.1% |
| Rest of Southern Africa | RSAF | 5% | 179.63 | 11.20 | 89.87 | 74.85 | 97.9% | 42.5% |
| Western Europe | WEU | 5% | 637.68 | 80.39 | 118.64 | 45.25 | 38.3% | 18.5% |
| Central Europe | CEU | 5% | 171.05 | 26.90 | 20.63 | 23.19 | 41.3% | 32.8% |
| Turkey | TUR | 5% | 29.52 | 60.49 | 12.87 | 0.00 | - | - |
| Ukraine Region | UKR | 5% | 66.45 | 10.40 | 19.58 | 8.62 | 58.1% | 22.3% |
| Central Asia | STAN | 5% | 19.67 | 82.08 | 37.90 | 0.00 | - | - |
| Russia Region | RUS | 40% | 266.36 | 40.25 | 43.82 | 58.40 | 53.5% | 41.0% |
| Middle East | ME | 5% | 8.60 | 136.63 | 39.30 | 0.00 | - | - |
| India | INDIA | 5% | 320.08 | 388.69 | 585.48 | 0.00 | - | - |
| Korea Region | KOR | 5% | 42.73 | 7.41 | 5.47 | 0.00 | - | - |
| China | CHN | 5% | 881.00 | 326.62 | 144.80 | 72.75 | 61.8% | 13.4% |
| South East Asia | SEAS | 5% | 1,213.01 | 45.46 | 131.95 | 19.49 | 16.2% | 9.9% |
| Indonesia | INDO | 5% | 1,291.53 | 15.08 | 114.33 | 0.00 | - | - |
| Japan | JAP | 5% | 208.43 | 2.12 | 13.29 | 6.94 | 10.7% | 31.1% |
| Rest of South Asia | RSAS | 5% | 74.19 | 245.78 | 227.85 | 0.00 | 0.0% | 0.0% |
| Oceania | OCE | 5% | 85.46 | 30.57 | 8.77 | 62.96 | 136.5% | 160.0% |
