# Peer review of "Regional variation in the effectiveness of methane-based and landbased climate mitigation options"

_Earth System Dynamics, 2020_

## Referee Comment (RC1) · Anonymous Referee #1 · 27 Jul 2020

Review of esd-2020-24: Regional variation in the effectiveness of methane and land-based climate mitigation options

Summary

The authors present and apply a method for examining the contributions of methane and land-based mitigation to meeting 1.5 and 2 degree warming targets. They give a thorough description of their modeling framework and present the results of an ensemble analysis of individual and combined contributions of methane vs land mitigation to emissions reductions and how these reductions can allow for complementary fossil fuel emissions. They conclude that methane mitigation contributes 2-4 times more reduc-

tion potential than land-based mitigation, depending on the BECCS assumptions. They also show that there are regional differences in how effective BECCS is compared to afforestation/reforestation, and estimate that bioenergy crop productivity must be fairly high in some places (with low transport losses ) for it to be an effective strategy. They also show that water usage for BECCS may impose limits to BECCS deployment in some regions.

Overall Response

This is an interesting and well-thought-out paper that examines key uncertainties in how to reach 1.5 and 2 degree targets. While I am not an expert in methane dynamics, the description of the framework is detailed enough to convey that the approach is reasonable for this analysis (assuming the methane references this is based on also have adequate methods). My main concern is the framing, and there is also some clarification of the experimental design that is needed. I elaborate on these two things here, with more detail below.

1) While I can appreciate the goal of presenting alternatives that allow for some fossil fuel emissions in these strict scenarios, this goal is not clearly articulated, and I am not sure that it is the most reasonable framing for this issue. Given that these are idealized scenarios and that there is considerable uncertainty on the adoption of mitigation policies, the actual extent of implementation of mitigation strategies, the assumptions and efficacy of mitigation strategies, and the modeling method, the estimated reduction levels here indicate that these approaches are more at the level of additional measures that would help ensure meeting particular targets under certain fossil fuel emission scenarios, rather than allow for more emissions to occur. Stating that doing these other mitigation actions allows for more fossil fuel emissions simply shifts responsibility away from the primary cause and increases the risk that these targets would not be met (the probability of exceedance is not particularly low to begin with). I suggest re-framing the study as additional mitigation potential or "insurance" mitigation potential. Barring a complete rework of the framing of the study away from allowing more

fossil fuel emissions and toward additional mitigation potential, there at least needs to be more discussion regarding the magnitude of these results in relation to the large uncertainties inherent in mitigation approaches, idealized scenarios, and modeling.

2) The description and figures and tables associated with the experimental design and its corresponding conditions are inconsistent and confusing.

Specific comments and suggestions

Abstract

lines 29-31: You should include that BECCS assumptions in general contribute to most of this range.

Introduction

You should include a description and examples of the expected emissions for 1.5 and 2 degree targets, which generally indicate that total (and fossil fuel) emissions need to drop to zero or negative to reach these goals. This provides a better context for why you are looking at how methane and land mitigation can alleviate the pressure to eliminate fossil fuel emissions completely.

Approach and Methodology

line 149: What is the first variable k?

line 172: It isn't clear how the global annual CH4 concentration is used to linearly interpolate monthly ozone values.

lines 203-204? Why only the non-CO2 components? If the models use different radiation schemes the CO2 component could also contribute to this uncertainty. Unless the CO2 radiative forcing is calculated the same way across the GCMs and in IMOGEN?

line 208: They "use" or "define" a framework?

line 207: "emissions"

lines 219-265: (section 2.3) This is very confusing and doesn't align with figure 3 or table 2. Table 2 is the most clear expression of the experimental design and should be used to organize this section and should be referenced up front. I suggest starting this section with a clear explanation of how many scenarios there are and each of the distinct components used to build them. Also, the nomenclature across the text, table 2, and figure 3 is inconsistent, adding to the confusion. It is also unclear how you reach different temperatures for control simulation, which appears to have a prescribed radiative forcing. I presume that the total radiative forcing is not prescribed, and that $CO_2$ conc. and associated $CH_4$ conc. feedbacks adjust to meet the prescribed temperature.

line 231: specify "...reduction in $CH_4$ emissions..."

line 240: figures 4 and 3 should be switched

lines 267-281: Are you assuming that carbon stored in the atmosphere is just $CO_2$?

line 296: Is there a better word than "productivity" here? Maybe "efficacy" or "mitigation potential"?

lines 328-353: this section 2.5 should be moved up and merged with section 2.3 (see previous comment) in order to clarify the experimental design.

Results and Discussion

line 367: and saturation effects?

lines 370-378: I would also like to see this put into the context of overall scenario uncertainty, as this is highly dependent upon human action. For example, the methane-related-mitigation AFFEB (over 85 years) is on the order of only about 6 years of late-century SSP5-8.5 emissions, which is a reference scenario. In the greater context, the potential methane mitigation effects represent more of a cushion or insurance approach to meeting idealized targets.

lines 379-383: This appears to be true, but this may be a coincidence as the dynamics

appear to be quite different between separate and coupled mitigation. the figures do not show the correct breakdowns for the linear sum.

lines 389-415: section 3.2 Is this for the 1.5 degree scenario only?

lines 413-415: Clarify that this is the BECCS only amount (for only 1.5 degrees?), which is double the original amount. Also add the numbers for land mitigation potential shown in figure 7c-d, as these are apparently in the abstract (100 GtC) and are comparable to the numbers in the previous section. This also makes a better case for "strong sensitivity" to BECCS assumptions, although tripling productivity and reducing transport losses by 2/3 to get a doubling in reduction is hardly a "strong sensitivity."

line 430: larger than what? the land mitigation? You should also note the exceptions here, which are abundant: canada, mexico, south america, brazil, west africa, south africa, korea, japan

line 473: "...regions that produce..."

lines 481-482: This needs more explanation. It isn't clear from the figure that these three regions have water issues under this case, especially china. While two of them would use all water availability, one does not, and none appear to exceed availability.

line 469: Table 4 should include BECCS demand and percent of available used in the example cases. Then you would have a basis for the statement in lines 48-482.

Conclusion

line 499: This "strong sensitivity" is not clear from the paper. The results can more clearly explain how BECCS mitigation can double, although based on tripling of productivity and a 2/3 reduction in transport losses, which nearly doubles the land mitigation potential. This is tremendous increase in BECCS efficacy to get this result, so I am not sure that it is a "strong sensitivity." And you don't show what figure 11 looks like with the original beccs values, to see how much difference the beccs efficacy makes on land cover. Also note that this has a much smaller relative effect on the total AFFEB.

Tables and Figures

Figures 3 and 4 are out of order.

Figure 3 The legends and the caption and table 2 and the references in the text don't match, which makes the experimental design unclear. The CH4 plots are c, e, g

Figure 6 Is the linear optimized the sum of ch4 and land based mitigation? Is so, then the bar breakdown is incorrect, as the coupling changes the land response.

Figure 7 again the linear sum does not look correct

Figures 12 and 13 I am confused about water withdrawal vs. irrigation demand. Isn't water withdrawn for irrigation? is this irrigation demand assumed to be additional water withdrawal?

Table 4 You should include BECCS demand and percent of available that would be used.

---

## Referee Comment (RC2) · Anonymous Referee #2 · 20 Aug 2020

Overall, I found the paper "Regional variation in the effectiveness of methane-based and land-based climate mitigation options" interesting and relevant. I have several comments that should be addressed prior to its publication.

Lines 21-24: Why only land-based mitigation and CH4?

Line 40-43: Add reference

Line 44-45: Add reference

Line 51-54: This sentence as written is confusing. Why are the requirements greater if the literature says they are similar? Are you saying that within the same model and

socioeconomic background land for BECCS is larger in 1.5 than 2C, but across the full literature range 2C scenarios have higher land requirements?

Line 59-60: This sentence should be made more elaborated on or removed.

Lines 255-259: Do you also adjust the energy system or its emissions to account for the reduction in bioenergy?

Line 284: does "preferred mitigation pathway" mean lowest terrestrial emissions or lowest total emissions (including CCS)?

Lines 286-292: Can you determine how much bioenergy (in EJ or Mt per year) you produce from this calculation?

Lines 347-354: This seems repetitive with previous text.

Lines 384-387: This paragraph needs some editing for clarity. The analysis you are doing is focused on the climate sensitivity of mitigation options, not an analysis of their economics or how that would change under different temperature targets. I don't think you can say that these are "worthwhile mitigation approaches" given your analysis. But, you can say that across the range of temperatures you analyzed there is no noticeable difference in the potential or performance of these mitigation strategies.

Lines 464-465: Why are those regions different?

Lines 468-471: What does "take the water requirements" mean? Do you use the water per unit of output from those studies and apply it to the IMAGE outputs? Or do you use the total water from those studies? If the latter, is it consistent? Also, does this mean you use the RCP2.6 water for the baseline and 1.9 simulations here? Is that water from the IMAGE-LPJmL model (which you note is low) or are you overwriting the IMAGE-LPJmL with the values from those papers?

Lines 472-482: It would be nice to have one sentence in this paragraph reporting the quantitative results before you go through the caveats.

Figure 3: Should the titles of panels d, f, and h say "Carbon Dioxide" instead of "Methane"? In general, I find the naming in this figure difficult difficult since you have 1.5C and 2C on a baseline panel and 2C on 1.5C panels.

Figure 7: Some of the detail in the caption would be good to include in the figure. In particular, the difference between panels a & c OR b & d.

Figure 9: This figure is pretty busy. Do you need the map? Or if you want the map, do you need the colors on the map? It is hard to see the bars and axes.

---

## Author Response (AR1)

"**Regional variation in the effectiveness of methane-based
and land-based climate mitigation options**"
Hayman et al. (ESD-2020-24)

**Author Response**

**General**

We are grateful to the two reviewers for their comments, which have helped us to improve the paper and make it clearer.

The reviewer comments about "reframing the study" and "reorganising" Section 2 suggest significant issues with the paper. Although we were advised that major revisions are needed, the changes made have nearly all been to improve the clarity of the paper or to provide further information. Our model results, analyses and interpretation remain unchanged.

- Reframing the study: It was certainly not our intention to suggest that methane mitigation is an alternative to mitigation of fossil fuel emissions. To avoid this impression, we make changes to the abstract, Section 3 and the conclusions (as given in Responses 1.1 and 1.16).

- Restructuring of Sections 2.3-2.5. We rework the original sections 2.3 and 2.5. Section 2.4 is unaffected by this. We move and integrate the first two paragraphs of the original Section 2.5 into Section 2.3. We integrate the third paragraph of the original Section 2.5 into Section 2.3.3 (Model Runs). We delete the fourth and final paragraph, as this repeats material in the original Section 2.3 and is therefore no longer required (see Responses 1.9 and 2.9).

We accept that there was some inconsistency in the abbreviations used for the land-based mitigation scenarios (bioenergy: CCS/IM-1.9; natural: Natural Land/IM-1.9N). We now define and use a consistent set of model scenario descriptors and abbreviations:

- Control ("CTL")
- "CH4" for the methane mitigation scenario
- "BECCS" for the land-based mitigation using bioenergy with carbon capture & storage
- "Natural" for the variant land-based mitigation scenario
- Coupled ("BECCS+CH$_4$") and coupled (Natural+CH$_4$") for the corresponding combined land-based and methane mitigation scenarios.

We make changes to the text, figures and tables to ensure a consistent use of these scenario descriptors and abbreviations.

We give our Response and the Change(s) made to the paper and to the Supplementary Information, after each reviewer comment. The reviewer comments are in normal font, with our "Response" and "Change(s) to Paper" in ***bold italics and indented***. The line numbers for the reviewer comments refer to the originally submitted manuscript. The line numbers for the "Change(s) to Paper" refer to the track change versions of the paper and Supporting Information that are included in this document (after the Author Response).

**"Regional variation in the effectiveness of methane-based
and land-based climate mitigation options"**
Hayman et al. (ESD-2020-24)

**Author Response to Reviewer 1 Comments**

**Reviewer 1**

Summary

The authors present and apply a method for examining the contributions of methane and land-based mitigation to meeting 1.5 and 2 degree warming targets. They give a thorough description of their modeling framework and present the results of an ensemble analysis of individual and combined contributions of methane vs land mitigation to emissions reductions and how these reductions can allow for complementary fossil fuel emissions. They conclude that methane mitigation contributes 2-4 times more reduction potential than land-based mitigation, depending on the BECCS assumptions. They also show that there are regional differences in how effective BECCS is compared to afforestation/reforestation, and estimate that bioenergy crop productivity must be fairly high in some places (with low transport losses ) for it to be an effective strategy. They also show that water usage for BECCS may impose limits to BECCS deployment in some regions.

Overall Response

This is an interesting and well-thought-out paper that examines key uncertainties in how to reach 1.5 and 2 degree targets. While I am not an expert in methane dynamics, the description of the framework is detailed enough to convey that the approach is reasonable for this analysis (assuming the methane references this is based on also have adequate methods). My main concern is the framing, and there is also some clarification of the experimental design that is needed. I elaborate on these two things here, with more detail below.

1) While I can appreciate the goal of presenting alternatives that allow for some fossil fuel emissions in these strict scenarios, this goal is not clearly articulated, and I am not sure that it is the most reasonable framing for this issue. Given that these are idealized scenarios and that there is considerable uncertainty on the adoption of mitigation policies, the actual extent of implementation of mitigation strategies, the assumptions and efficacy of mitigation strategies, and the modeling method, the estimated reduction levels here indicate that these approaches are more at the level of additional measures that would help ensure meeting particular targets under certain fossil fuel emission scenarios, rather than allow for more emissions to occur. Stating that doing these other mitigation actions allows for more fossil fuel emissions simply shifts responsibility away from the primary cause and increases the risk that these targets would not be met (the probability of exceedance is not particularly low to begin with). I suggest re-framing the study as additional mitigation potential or "insurance" mitigation potential. Barring a complete rework of the framing of the study away from allowing more fossil fuel emissions and toward additional mitigation potential, there at least needs to be more discussion regarding the magnitude of these results in relation to the large uncertainties inherent in mitigation approaches, idealized scenarios, and modeling.

> ***Response 1.1: It was certainly not our intention to suggest that methane mitigation is an alternative to $CO_2$ mitigation. To avoid this impression, we will make the following changes:***
>
> ***a) We replace the last line of the abstract with: "Although the primary problem remains mitigation of fossil fuel emissions, our results highlight the unrealised potential for the mitigation of $CH_4$ emissions to make the Paris climate targets more achievable".***

b) *By rewriting the last three sentences of the paper as: "Stabilising the climate primarily requires urgent action to mitigate $CO_2$ emissions. However, $CH_4$ mitigation has the potential to make the Paris targets more achievable by offsetting up to 188-212 GtC of anthropogenic $CO_2$ emissions. We conclude that $CH_4$ mitigation would be effective globally and especially so for the major $CH_4$-emitting regions of India, USA and China".*

*Change(s) to Paper: For (a). We amend lines 32-36 (page 1), showing the added and deleted text:*

*" Although the primary requirement remains mitigation of fossil fuel emissions, our results highlight the unrealised potential for the mitigation of $CH_4$ emissions to make the Paris climate targets more achievable".*

*For (b): We amend lines 595-600 (page 19), showing the added and deleted text:*

*" Stabilising the climate primarily requires urgent action to mitigate $CO_2$ emissions. However, $CH_4$ mitigation has the potential to make the Paris targets more achievable by offsetting up to 188-212 GtC of anthropogenic $CO_2$ emissions. We conclude that $CH_4$ mitigation would be effective globally and especially so for the major $CH_4$-emitting regions of India, USA and China".*

*See also Response 1.16, where we add the following sentence (lines 450-451) "Although the primary challenge remains mitigation of fossil fuel emissions, these results highlight the unrealised potential of these mitigation options to make the Paris climate targets more achievable".*

2) The description and figures and tables associated with the experimental design and its corresponding conditions are inconsistent and confusing.

*Response: We respond below to the specific reviewer comments on these points.*

Specific comments and suggestions

Abstract

Lines 29-31: You should include that BECCS assumptions in general contribute to most of this range.

*Response 1.2: We will amend the abstract to include this point about the BECCS assumptions.*

*Change(s) to Paper: At line 30, we add "the large range reflecting assumptions and uncertainties associated with BECCS".*

Introduction

You should include a description and examples of the expected emissions for 1.5 and 2 degree targets, which generally indicate that total (and fossil fuel) emissions need to drop to zero or negative to reach these goals. This provides a better context for why you are looking at how methane and land mitigation can alleviate the pressure to eliminate fossil fuel emissions completely.

*Response 1.3:* *The idealised temperature pathways used in this work imply the need to drop to zero emissions (lines 209-210). In our previous work (cited paper by Comyn-Platt et al., 2018), we derive the parameters for the temperature pathways from comparison with CMIP5 simulations for the RCP2.6 scenario (Supplementary Information, Figure 2 of Comyn-Platt et al., 2018).*

*We add remaining carbon budget for the 1.5°C and 2°C warming targets, equivalent to the discussion we gave in our earlier paper (cited paper by Comyn-Platt et al., 2018) and also from the published literature and IPCC reports. As indicated above, we amend the text to make explicit the need for complete removal of fossil carbon emissions and the likely need for negative emission technologies.*

*Change(s) to Paper:* *At line 42, we add:*

*"The IPCC Special Report on Global Warming of 1.5°C (IPCC, 2018) gives the median remaining carbon budgets between 2018 and 2100 as 770 GtCO$_2$ (210 GtC) and 1690 GtCO$_2$ (~461 GtC) to limit global warming to 1.5°C and 2°C, respectively. These budgets represent ~20 and ~41 years at present-day emission rates. The actual budgets could however be smaller, as they exclude Earth system feedbacks such as CO$_2$ released by permafrost thaw or CH$_4$ released by wetlands".*

*We also include the underlined text in lines 46-49:*

*"Meeting the Paris Agreement goals will, therefore, require sustained reductions in sources of fossil carbon emissions, other long-lived anthropogenic greenhouse gases (GHGs) and some short-lived climate forcers (SLCFs) such as methane (CH$_4$), alongside increasingly extensive implementations of carbon dioxide removal (CDR) technologies (IPCC, 2018)".*

Approach and Methodology

line 149: What is the first variable k?

*Response 1.4:* *k is a dimensionless scaling constant such that the global annual wetland methane emissions are 180 Tg CH$_4$ in 2000. We will add this sentence.*

*Change(s) to Paper:* *At line 160, we add: "k is a dimensionless scaling constant such that the global annual wetland methane emissions are 180 Tg CH$_4$ in 2000 (as described in Comyn-Platt et al. (2018))"*

line 172: It isn't clear how the global annual CH4 concentration is used to linearly interpolate monthly ozone values.

*Response 1.5:* *We do not model tropospheric ozone production from methane explicitly in IMOGEN. Instead, we use two sets of monthly near-surface O$_3$ concentration fields (January-December) from HADGEM3-A GA4.0 model runs, the sets corresponding to low (1285 ppbv) and high (2062 ppbv) global mean atmospheric CH$_4$ concentrations. We assume that the atmospheric O$_3$ concentration responds linearly to the atmospheric CH$_4$ concentration in each grid cell. We derive separate linear relationships for each month and grid cell, and use these to calculate the surface O$_3$ concentration from the corresponding atmospheric CH$_4$ concentration as it evolves during the IMOGEN run.*

*We will amend the text using the above.*

*Change(s) to Paper:* *We amend the text from line 179, including , to:*

*"…, we do not model tropospheric ozone production from CH₄ explicitly in IMOGEN. Instead, we use two sets of monthly near-surface O₃ concentration fields (January-December) from HADGEM3-A GA4.0 model runs, with the sets corresponding to*  *low (1285 ppbv) and high (2062 ppbv) global mean atmospheric CH₄ concentrations (Stohl et al., 2015). We assume that the atmospheric O₃ concentration in each grid cell responds linearly to the atmospheric CH₄ concentration. We derive separate linear relationships for each month and grid cell, and use these to calculate the surface O₃ concentration from the corresponding global atmospheric CH₄ concentration as it evolves during the IMOGEN run.* .

lines 203-204? Why only the non-CO₂ components? If the models use different radiation schemes the CO2 component could also contribute to this uncertainty. Unless the CO₂ radiative forcing is calculated the same way across the GCMs and in IMOGEN?

*__Response 1.6:__ From the cited paper by Huntingford et al. (2010), IMOGEN uses four parameters for the energy balance model: these are climate feedback parameters over land and ocean, $\lambda_l$ and $\lambda_o$ (W m⁻² K⁻¹) respectively, oceanic "effective thermal diffusivity", $\kappa$ (W m⁻¹ K⁻¹) representing the ocean thermal inertia and a land-sea temperature contrast parameter, $\nu$, linearly relating warming over land, $\Delta T_l$ (K) to warming over ocean, $\Delta T_o$ (K), as $\Delta T_l = \nu \Delta T_o$. The climate feedback parameters ($\lambda_l$ and $\lambda_o$) are calibrated using GCM data for top of the atmosphere radiative fluxes, mean land and ocean surface temperatures, along with an estimate of the radiative forcing modelled by the GCM for the CO₂ changes. Thus, IMOGEN emulates the radiative forcing of CO₂ within the individual GCMs.*

*For a given prescribed trajectory in temperature, and pathway in atmospheric non-CO₂ greenhouse gas emissions, we calculate compatible CO₂ emissions. These emissions trajectories are different, dependent on which climate or Earth system (ES) model is emulated (via IMOGEN).*

*__Change(s) to Paper:__ We add the following paragraph (lines 204-210)*

*"The EBM includes a simple representation of the ocean uptake of heat and CO₂ and uses a separate set of four parameters for each climate and Earth system model emulated (Huntingford et al., 2010): the climate feedback parameters over land and ocean, $\lambda_l$ and $\lambda_o$ (W m⁻² K⁻¹) respectively, the oceanic "effective thermal diffusivity", $\kappa$ (W m⁻¹ K⁻¹) representing the ocean thermal inertia and a land-sea temperature contrast parameter, $\nu$, linearly relating warming over land, $\Delta T_l$ (K) to warming over ocean, $\Delta T_o$ (K), as $\Delta T_l = \nu \Delta T_o$. The climate feedback parameters ($\lambda_l$ and $\lambda_o$) are calibrated using model-specific data for the top of the atmosphere radiative fluxes, the mean land and ocean surface temperatures, along with an estimate of the radiative forcing modelled for the CO₂ changes."*

*We amend lines 227-229, showing __added__ and  text "__As we have a model-specific estimate of the radiative forcing modelled for the CO₂ changes (see above),__ ~~We__we__we__,__ therefore, attribute the spread in ΔQ to the uncertainty in the non-CO₂ radiative forcing component".*

line 208: They "use" or "define" a framework?

*__Response 1.7:__ Accepted, missing word in "Huntingford et al. (2017) a framework"*

*__Change(s) to Paper:__ Line 233 now reads: "Huntingford et al. (2017) __define__ a framework"*

line 207: "emissions"

*Response 1.8: We will add "of greenhouse gases and short-lived climate forcers" after emissions in "model the efforts of humanity to limit emissions and, if necessary, capture atmospheric carbon"*

*Change(s) to Paper: Line 234 now reads: "model the efforts of humanity to limit emissions of greenhouse gases and short-lived climate forcers, and, if necessary, capture atmospheric carbon".*

lines 219-265: (section 2.3) This is very confusing and doesn't align with figure 3 or table 2. Table 2 is the most clear expression of the experimental design and should be used to organize this section and should be referenced up front. I suggest starting this section with a clear explanation of how many scenarios there are and each of the distinct components used to build them. Also, the nomenclature across the text, table 2, and figure 3 is inconsistent, adding to the confusion. It is also unclear how you reach different temperatures for control simulation, which appears to have a prescribed radiative forcing. I presume that the total radiative forcing is not prescribed, and that $CO_2$ conc. and associated $CH_4$ conc. feedbacks adjust to meet the prescribed temperature.

*Response 1.9: With this and the comments from Reviewer 2, we restructure Section 2.3-2.5 to make the scenarios used clearer, to remove any inconsistencies and to avoid any repetition. See also the response below to the comment on lines 328-353.*

*The reviewer is correct that "the total radiative forcing is not prescribed, … adjust to meet the prescribed temperature" .We will expand the text on the inverse version of IMOGEN to make this clear (lines 197-206).*

*Change(s) to Paper: We add the following text on the radiative forcing to Section 2.2.1 (lines 220-221) "In this study, we use the inverse version of IMOGEN, which follows prescribed temperature pathways (Fig. 3(a)), to derive the total radiative forcing ($\Delta Q$ [total]) and then the $CO_2$ radiative forcing ($\Delta Q$ [$CO_2$]), using Eq. 2".*

*We rework the original sections 2.3 and 2.5. Section 2.4 is unaffected by this. We move and integrate the first two paragraphs of the original Section 2.5 into Section 2.3. We delete the fourth and final paragraph, as this repeats material in Section 2.3 and is therefore no longer required (see also Response 2.9).*

*As indicated in the general comments, we define and use a consistent set of model scenario descriptors and abbreviations: Control ("CTL"), "$CH_4$" for the methane mitigation scenario, "BECCS" for the land-based mitigation using bioenergy with carbon capture & storage, "Natural" for the variant land-based mitigation scenario, coupled ("BECCS+$CH_4$") and coupled ("Natural+$CH_4$"), for the corresponding combined land-based and methane mitigation scenarios.*

*We make the following changes to Section 2.3 (lines 243-322), showing added and  text:*

[revised manuscript text omitted]

line 231: specify "… reduction in CH4 emissions …"

*Response 1.10: We will amend the text as suggested.*

*Change(s) to Paper: Line 265 now reads "the reduction in $CH_4$ emissions from specific source sectors"*

line 240: figures 4 and 3 should be switched

*Response 1.11:* **This may follow from the restructuring of Sections 2.3-2.5.**

*Change(s) to Paper:* **Following the re-working of Sections 2.3-2.5, we split Figure 3. Panels (a) and (b) of the original Figure 3 form Figure 3 in the revised paper (page 31). Figure 4 remains unchanged (page 32) and Panels (c)-(h) of the original Figure 3 form a new Figure 6 (page 34). In consequence, we have had to renumber the subsequent figures.**

lines 267-281: Are you assuming that carbon stored in the atmosphere is just CO2?

*Response 1.12:* **The reviewer is correct that the calculation of the atmospheric carbon store in the post-processing does not take account of $CH_4$. This is a reasonable approximation, however, given the relative magnitude of the atmospheric concentrations of methane (~2 ppmv at the surface) and carbon dioxide (400 ppmv).**

**For the contemporary period, IMOGEN retains ~50% of carbon dioxide emissions in the atmosphere, after land and ocean draw down is accounted. In that sense, there is closure of direct carbon units. For methane, a slightly different approach is used. We calculate the atmospheric $CH_4$ concentrations from the $CH_4$ emissions (from both anthropogenic and natural sources) and an atmospheric loss term, parameterised via a methane turnover lifetime. We take account of the radiative forcing of atmospheric methane and its effect on the terrestrial carbon cycle (through tropospheric $O_3$ production and vegetation $O_3$ damage).**

*Change(s) to Paper:* **We add the following text at line 330: " The atmospheric carbon store does not include $CH_4$. This is a reasonable approximation, however, given the relative magnitudes of the atmospheric concentrations of $CH_4$ (~2 ppmv at the surface) and $CO_2$ (400 ppmv)".**

line 296: Is there a better word than "productivity" here? Maybe "efficacy" or "mitigation potential"?

*Response 1.13:* **We will amend the text along the lines suggested.**

*Change(s) to Paper:* **Line 358 now reads "The efficacy of the BECCS scheme implemented in JULES"**

lines 328-353: this section 2.5 should be moved up and merged with section 2.3 (see previous comment) in order to clarify the experimental design.

*Response 1.14:* **As per the responses to the comment on lines 219-265 and from Reviewer 2, we will restructure sections 2.3-2.5. Some of the existing Section 2.5 will be moved, but the material on optimisation and mitigation potential needs to come after Section 2.4.**

*Change(s) to Paper:* **This is covered by the "Change(s) to Paper" for Response 1.9.**

Results and Discussion

line 367: and saturation effects?

*Response 1.15:* **We reconfirm this sentence is correct, but there would also be eventual saturation. The oceanic draw down of $CO_2$ is based on the work of Joos et al (1996), with the equations specific to IMOGEN given in the Appendix of Huntingford et al (2004). The $CO_2$ draw down flux is based on the difference between atmospheric $CO_2$ and $CO_2$ concentrations near the ocean surface. The oceanic $CO_2$ concentration is calculated as a weighted integration in time of this flux, and where such weighting accounts for oceanic diffusive mixing. If atmospheric $CO_2$ rises quickly, there is a co-benefit as the oceanic draw down will rise due to the gradient between the two $CO_2$ concentrations. However, the reviewer is correct, that under climate stabilisation, saturation will occur and this flux will decrease to zero.**

*Change(s) to Paper: We add the underlined text to lines 434-435 "the oceanic drawdown of  $CO_2$ rises (although it eventually falls to zero under climate stabilisation and there would also be implications for ocean acidification)".*

lines 370-378: I would also like to see this put into the context of overall scenario uncertainty, as this is highly dependent upon human action. For example, the methane-related-mitigation AFFEB (over 85 years) is on the order of only about 6 years of late century SSP5-8.5 emissions, which is a reference scenario. In the greater context, the potential methane mitigation effects represent more of a cushion or insurance approach to meeting idealized targets.

*Response 1.16: We use the SSP2 reference scenario as the control scenario. The baseline forcing in this scenario is slightly above 6 W $m^{-2}$. The SSP5 8.5 scenario is a long-way from the aspirations of the Paris agreement. This paper focusses on the more policy-relevant question of how $CH_4$ mitigation can contribute to meeting the Paris targets. In response to this and related comments, we make it clearer that $CH_4$ mitigation is in no way an alternative to $CO_2$ mitigation.*

*Change(s) to Paper: We add the following sentence (lines 450-451) "Although the primary challenge remains mitigation of fossil fuel emissions, these results highlight the unrealised potential of these mitigation options to make the Paris climate targets more achievable".*

lines 379-383: This appears to be true, but this may be a coincidence as the dynamics appear to be quite different between separate and coupled mitigation. The figures do not show the correct breakdowns for the linear sum.

*Response 1.17: Although Figures 6 and 7 are correct and as the reviewer notes for Figure 6 (see below), adding the $CH_4$ mitigation (second bar) to any of the land-based mitigation options (bars 3-5) does not appear to give the corresponding coupled option (bars 6-8). This is because the gain in the land carbon store for the methane mitigation option is shown as a reduction from -70.8 GtC in the control run to -1.4 GtC in the methane mitigation option (median of ensemble). This then explains the positive changes shown for the land carbon stores in the coupled runs. Comparing the final two bars, there is very good agreement in the breakdown of the carbon stores for the coupled and linear (i.e., the sum of the individual) mitigation options. Thus, the dynamics and in the coupled and linear cases are almost identical.*

*Although we state that "there is increased uptake of carbon by the land, directly because of the increased atmospheric $CO_2$ concentration and indirectly through the reduction in $O_3$ damage, which is greater than the land carbon lost through land-use changes" (lines 363-365), this did not make clear of the size of the change. We will add a sentence to the text and figure caption to make the point about the change in the land carbon store for the methane mitigation option.*

*Figure 6*

[Figure]

*Figure 7*

[Figure]

*Change(s) to Paper: We add the following text (lines 438-441) "In both Figs. 8 and 9, it should be noted that the gain in the land carbon store for the "CH₄" mitigation option is shown as a reduction from -70.8 GtC loss of land carbon in the control run to -1.4 GtC loss in the methane mitigation option (median of ensemble). This then explains the positive changes shown for the land carbon stores in the coupled "BECCS+ CH₄" and coupled "Natural+ CH₄" scenarios ".*

*We also add the following text to the figure captions for the new Figures 8 (page 36) and 9 (page 37) [original Figures 6 and 7]: "Note that the gain in the land carbon store for the CH₄ scenario is shown as a reduction from -70.8 GtC in the control run to -1.4 GtC in the "CH₄" mitigation option (median of ensemble)".*

*See also Response 1.26 and 1.27, which make the same comment on the figures.*

lines 389-415: section 3.2 Is this for the 1.5 degree scenario only?

*Response 1.18: We will amend the text and add a reference to Supplementary information, Figure 2, which shows the equivalent plot for 2°C of warming to Figure 10.*

*Change(s) to Paper: We add the underline text to line 464 "Globally across the two temperature targets, our simulations imply a removal of 27-30 GtC from the active carbon-cycle". We also add the following text (lines 487-488) "Supporting Information, Figure SI.3 shows the corresponding plots for the 2°C warming target".*

lines 413-415: Clarify that this is the BECCS only amount (for only 1.5 degrees?), which is double the original amount. Also add the numbers for land mitigation potential shown in figure 7c-d, as these are apparently in the abstract (100 GtC) and are comparable to the numbers in the previous section. This also makes a better case for "strong sensitivity" to BECCS assumptions, although tripling productivity and reducing transport losses by 2/3 to get a doubling in reduction is hardly a "strong sensitivity."

*Response 1.19: We apologise as we inadvertently gave the incorrect numbers for the carbon budgets. The correct text should read "We now find the global land-based mitigation potential to be  88-100 GtC, as shown in Fig. 7(c) and (d). We use κ = 3 in the subsequent analysis of regional mitigation options and of BECCS water requirements." This is now consistent with the land-based bars in Figure 7d (second bar) and the upper range of 100 GtC given in the abstract. The range is based on the results from both the 1.5°C and 2°C runs for the optimised land-based mitigation.*

*Change(s) to Paper: We amend the text at lines 487-488: "We now find the global land-based mitigation potential to be 88-100 GtC across the two temperature targets".*

line 430: larger than what? the land mitigation? You should also note the exceptions here, which are abundant: canada, mexico, south america, brazil, west africa, south africa, korea, japan

*Response 1.20: We were referring to the global impact of methane mitigation. As presented in Figures 9 and 10, the reviewer is correct that the land-based mitigation is larger than methane mitigation for a number of the IMAGE regions, especially when using a BECCS scaling factor κ = 3. As we are discussing the mitigation options at a regional level, we will amend lines 430-431 as shown "CH₄ mitigation  is an effective mitigation strategy for all regions, and especially the major methane emitting region."*

*Change(s) to Paper: We delete the underlined text in line 505: "CH₄ mitigation  is an effective mitigation strategy for all regions".*

line 473: "… regions that produce …"

> **_Response 1.21:_** *Accept text change*

> **_Change(s) to Paper:_** *We delete the underlined text in lines 554-555: "for the  regions that produce a substantial amount of BECCS".*

lines 481-482: This needs more explanation. It isn't clear from the figure that these three regions have water issues under this case, especially china. While two of them would use all water availability, one does not, and none appear to exceed availability.

> **_Response 1.22:_** *In the next comment, the reviewer has suggested adding the BECCS water demand and percent of available used to Table 4. We will use this information to clarify whether these IMAGE regions have issues with water availability.*

> **_Change(s) to Paper:_** *We add the following text (lines 555-559) "Tables 4a and 4b show the additional water requirements of BECCS calculated for 2060 and 2100, respectively, for the 2°C warming target. We find that the additional demand for BECCS would lead to an exceedence (or is >90%) of the available water for the Oceania and Rest of Southern Africa regions. We also find that the additional demand for BECCS is greater than the total water withdrawals from anthropogenic activities for the Canada and Brazil IMAGE regions".*

> *We amend the text (lines 568-570), showing added and  text: "Nevertheless, our results indicate that the additional water demand for BECCS have large impacts in half of the regions substantially invested in BECCS: Oceania, Rest of South Africa, Brazil and Canada ".*

line 469: Table 4 should include BECCS demand and percent of available used in the example cases. Then you would have a basis for the statement in lines 48-482.

> **_Response 1.23:_** *We will add these to Table4 or add as a new Table.*

> **_Change(s) to Paper:_** *We have added three additional columns to Table 4: (a) the additional water demand from BECCS; (b) the total water demand including that for BECCS, as a percentage of the available water; and (c) the water demand for BECCS as a percentage of total water demand including that for BECCS. We now have separate tables for 2060 (Table 4a, page 55) and 2100 (Table 4b, page 56).*

Conclusion

line 499: This "strong sensitivity" is not clear from the paper. The results can more clearly explain how BECCS mitigation can double, although based on tripling of productivity and a 2/3 reduction in transport losses, which nearly doubles the land mitigation potential. This is tremendous increase in BECCS efficacy to get this result, so I am not sure that it is a "strong sensitivity." And you don't show what figure 11 looks like with the original beccs values, to see how much difference the beccs efficacy makes on land cover. Also note that this has a much smaller relative effect on the total AFFEB.

> **_Response 1.24:_** *We accept the reviewer's viewpoint that our perturbations are relatively large so the changes do not necessarily imply a "strong sensitivity". We have therefore removed "strong" from line 499.*

> **_Change(s) to Paper:_** *We amend the text in line588 to "quantify  the sensitivity", deleting the text underlined.*

Tables and Figures

Figures 3 and 4 are out of order.

*Response 1.25: This will be resolved with the restructuring of Sections 2.3-2.5. This is linked to Response 1.11.*

*Change(s) to Paper: Following the re-working of Sections 2.3-2.5, we split Figure 3. Panels (a) and (b) of the original Figure 3 form Figure 3 in the revised paper (page 31). These show the temperature (panel a) and radiative forcing (panel b) pathways. Figure 4 remains unchanged (page 32). Panels (c)-(h) of the original Figure 3 form a new figure (Figure 6, page 34).*

Figure 3 The legends and the caption and table 2 and the references in the text don't match, which makes the experimental design unclear. The CH4 plots are c, e, g

*Response 1.26: The figure is intended to show key data inputs for or differences between the model runs to help inform reader's understanding of the paper. The titles of the panels are correct but we accept the figure and panels need careful reading. The other reviewer also commented that the figure is unclear. We will amend the figure to make it clearer (and potentially split the figure into two as part of the restructuring of Section 2.3-2.5).*

*Change(s) to Paper:*

*We define a consistent set of model scenarios at the beginning of the revised Section 2.3 (lines 245-250) "We undertake a control run and other simulations with anthropogenic $CH_4$ mitigation or land-based mitigation, stabilising at either 1.5°C or 2.0°C warming without a temperature overshoot. We denote the control run as "CTL", the anthropogenic $CH_4$ mitigation scenario as "$CH_4$", a land-based mitigation scenario using BECCS as "BECCS" and a variant land-based scenario focussing on AR as "Natural". We also undertake runs combining the $CH_4$ and land-based mitigation scenarios (Coupled "BECCS+$CH_4$" and Coupled "Natural+$CH_4$") to determine if there are any non-linearities when we combine these mitigation scenarios".*

*We amend the following abbreviations of the model scenarios in Table 2 (page 52): (a) "CCS" is replaced with "BECCS"; (b) "Natural Land" becomes "Natural", (c) "Coupled ($CH_4$+CCS)" becomes "Coupled (BECCS+$CH_4$)" and (d) "Coupled ($CH_4$+Natural Land)" becomes "Coupled (Natural+$CH_4$)".*

*Figure 6 (part of the original Figure 3) shows the modelled evolution of the atmospheric concentrations of $CH_4$ (the left-hand panels: a, c and e) and of $CO_2$ (right-hand panels: b, d and f) for the pairs of model runs: "CTL" vs "$CH_4$" (upper row), "BECSS" vs "BECCS+$CH_4$" (middle row) and "Natural" vs "Natural+$CH_4$". The title of each panel is the pair of model runs. The subtitle and legend use the same set of model run abbreviations (i.e., "CTL", "$CH_4$", "BECSS", "Natural", "BECCS+$CH_4$" and "Natural").*

Figure 6 Is the linear optimized the sum of ch4 and land based mitigation? Is so, then the bar breakdown is incorrect, as the coupling changes the land response.

*Response 1.27: Please see response to comment on lines 379-383 above (Response 1.17), as this is a repeat of that comment.*

*Change(s) to Paper: We add the following text to the figure caption for the new Figures 8 (page 36): "Note that the gain in the land carbon store for the $CH_4$ scenario is shown as a reduction from -70.8 GtC in the control run to -1.4 GtC in the "$CH_4$" mitigation option (median of ensemble)".*

Figure 7 again the linear sum does not look correct

> *Response 1.28:* *Please see response to comment on lines 379-383 above (Response 1.17), as this is a repeat of that comment.*

> *Change(s) to Paper:* *We add the following text to the figure caption for the new Figures 9 (page 37): "Note that the gain in the land carbon store for the $CH_4$ scenario is shown as a reduction from -70.8 GtC in the control run to -1.4 GtC in the "$CH_4$" mitigation option (median of ensemble)".*

Figures 12 and 13 I am confused about water withdrawal vs. irrigation demand. Isn't water withdrawn for irrigation? is this irrigation demand assumed to be additional water withdrawal?

> *Response 1.29:* *We will amend this. The reviewer is correct that irrigation is water withdrawal. The separation arose as we had separate datasets for (a) agricultural irrigation and (b) water withdrawals for energy production, use in industry and in cities. We will adjust the figure to remove any confusion.*

> *Change(s) to Paper:* *We amend the legends to the new Figures 13 and 14, replacing "SSP2 Irrigation Demand" and "SSP2 Water Withdrawal" with "Water Demand: Irrigation" and "Water Demand: Other" (i.e., for energy generation, industry and domestic uses), respectively.*

> *We also amend the captions to the new Figure 13, replacing "The water withdrawal (dark blue) and irrigation demand are taken from the SSP2-RCP2.6-IMAGE database" with "The water demand for irrigation (yellow) and for other uses (i.e., energy generation, industry and domestic; dark blue) are taken from the SSP2-RCP2.6-IMAGE database. Note there is very little BECCS additional water demand (green) in 2015".*

> *We also amend the captions to the new Figure 14, adding the underlined text "SSP2-IMAGE water demand estimates for irrigation (yellow) and other (dark blue)".*

Table 4 You should include BECCS demand and percent of available that would be used.

> *Response 1.30:* *Please see the response to the comment on line 469 above (Response 1.23), as this is a repeat of that comment.*

"**Regional variation in the effectiveness of methane-based
and land-based climate mitigation options**"
Hayman et al. (ESD-2020-24)

**Author Response to Reviewer 2 Comments**

**Reviewer 2**

Overall, I found the paper "Regional variation in the effectiveness of methane-based and land-based climate mitigation options" interesting and relevant. I have several comments that should be addressed prior to its publication.

> *Thank you for the positive comments about the paper and its relevance.*

Lines 21-24: Why only land-based mitigation and CH4?

> *Response 2.1: We only considered methane and land-based mitigation options as we could investigate the climate/land carbon-cycle interactions and feedbacks of these mitigation options within our modelling framework. The paper also builds on our earlier studies, as described in the Introduction (lines 83-86).*
>
> *Change(s) to Paper: We add the underlined text to lines 21-22: "Specifically, within this IMOGEN-JULES framework, we focus on and characterise the global and regional effectiveness of land-based".*

Line 40-43: Add reference

> *Response 2.2: We add reference(s). Refers to "Meeting the Paris Agreement goals will, therefore, require sustained reductions in sources of long-lived anthropogenic greenhouse gases (GHGs) and some short-lived climate forcers (SLCFs) such as methane, alongside increasingly extensive implementations of carbon dioxide removal (CDR) technologies. Accurate information is needed on the range and efficacy of options available to achieve this."*
>
> *Change(s) to Paper: We add the underlined text to lines 46-49 "Meeting the Paris Agreement goals will…… alongside increasingly extensive implementations of carbon dioxide removal (CDR) technologies (IPCC 1998)".*

Line 44-45: Add reference

> *Response 2.3: We add reference(s). Refers to "Biomass energy with carbon capture and storage (BECCS) and afforestation/reforestation (AR) are the most widely considered CDR technologies in the climate and energy literature".*
>
> *Change(s) to Paper: We add the reference "(Minx et al., 2018)" to the end of the sentence above (line 52).*

Line 51-54: This sentence as written is confusing. Why are the requirements greater if the literature says they are similar? Are you saying that within the same model and socioeconomic background land for BECCS is larger in 1.5 than 2C, but across the full literature range 2C scenarios have higher land requirements?

> *Response 2.4: We cite Smith et al. (2016) for the land needed for large-scale bioenergy crops to achieve the 2°C target. "BECCS delivering 3.3 Gt Ceq yr$^{-1}$ of negative emissions would require a land area of approximately 380–700 Mha in 2100 (Table 2)". From the cited paper of van Vuuren*

*et al. (2018), "In the default mitigation scenario (DEF_1.9 which is compatible with the 1.5°C target), more than 600 Mha is required for bioenergy".*

*Our earlier paper (Harper et al., 2018) clearly shows that less land is required for bioenergy crops to achieve the 2°C warming target. This is the case within a given shared socio-economic pathway (SSP2 as used here) but there are larger differences across different SSPs. The land requirements for bio-energy productions strongly differs in the literature. Key elements include the contribution of residues and the assumed yields and yield improvement. In IMAGE, the total land use requirement in 2100 is 360 Mha for the SSP2-2.6 and similar numbers for the SSP2-1.9. Interestingly, area used for bio-energy is higher in the SSP1-2.6 scenario given the much lower land claim for food production. We will use this to amend the text.*

*Change(s) to Paper: We amend the text from lines 58-64, showing additions and .*

*"The land requirements for BECCS will be greater for the 1.5°C target within a given shared socio-economic pathway (e.g., SSP2), although published estimates are similar for the two warming targets, with between 380-700 Mha required for the 2°C target (Smith et al., 2016) and greater than 600 Mha for the 1.5°C target (van Vuuren et al., 2018). This is because the land requirements for bioenergy production differ strongly across the different SSPs, depending on assumptions about the contribution of residues, assumed yields and yield improvement, start dates of implementation and the rates of deployment".*

Line 59-60: This sentence should be made more elaborated on or removed.

*Response 2.5: We delete the sentence.*

*Change(s) to Paper: We delete the sentence "The IPCC Special Report "Climate Change and Land Use"(IPCC, 2019) provides a further synthesis and perspective on BECCS" (lines 67-68).*

Lines 255-259: Do you also adjust the energy system or its emissions to account for the reduction in bioenergy?

*Response 2.6: The reviewer is correct, as there would be an adjustment to the energy system. We do not account for this. We however do acknowledge this limitation for the converse case when bioenergy crops are grown (lines 416-418) "Further, we do not allow for the reduced emissions from fossil fuel combustion due to the bioenergy crop being grown, as this would require energy sector modelling that is beyond the scope of this study". We will add this as a caveat to lines 255-259 and amend lines 416-418 to cover both cases.*

*Change(s) to Paper: We add the following sentence "We make no allowance for any changes in the energy generation system, as this would require energy sector modelling that is beyond the scope of this study" (lines 295-296).*

*We add the following text (lines 584-587) "Further, we do not allow for the reduced emissions from fossil fuel combustion due to the bioenergy crop being grown (or the converse when bioenergy crops are replaced in the Natural scenario run), as this would require energy sector modelling that is beyond the scope of this study".*

Line 284: does "preferred mitigation pathway" mean lowest terrestrial emissions or lowest total emissions (including CCS)?

*Response 2.7: This refers to our earlier work. "Harper et al. (2018) find that the land-use pathways do not provide a clear choice for the preferred mitigation pathway." We will amend the text.*

*Change(s) to Paper:* *We add the sentence "In such circumstances, Harper et al. (2018) find that
the loss of soil carbon in regions with high carbon density makes it difficult for BECCS to deliver
a net negative emission of $CO_2$" (lines 346-347).*

Lines 286-292: Can you determine how much bioenergy (in EJ or Mt per year) you produce from this
calculation?

*Response 2.8:* *We will add this information.*

*Change(s) to Paper:* *We amend lines 364-369, showing* underlined added and  *text:*

*"{JULES* in this study and *in Harper et al. (2018) simulated* median *average yields of ~4.8 and
~4.6 tDM ha$^{-1}$ yr$^{-1}$, respectively, compared to measured median of 11.5 tDM ha$^{-1}$ yr$^{-1}$ and
simulated average of 15.8 tDM ha$^{-1}$ yr$^{-1}$ in IMAGE}.* The JULES yield of ~4.8 tDM ha$^{-1}$ yr$^{-1}$
corresponds to ~59 EJ yr-1 of primary energy, using the maximum area for BECCS from Table 2
of 637.7 Mha and an energy yield of 19.5 GJ t DM$^{-1}$ (Daioglou et al., 2017). Bioenergy supplied
55.6 EJ yr$^{-1}$ or ~10% of primary energy requirement worldwide in 2017 (WBA, 2019). According
to Smith et al. (2016), this would increase to ~170 EJ yr$^{-1}$ of primary energy in 2100, for negative
emissions of 3.3 Gt $C_{eq}$ yr$^{-1}$ from BECCS (as required for a 2°C warming target)".*

Lines 347-354: This seems repetitive with previous text.

*Response 2.9:* *It was intended as a summary but we accept that it repeat texts in previous
sections. In reply to the comments from Reviewer 1, we restructure Sections 2.3-2.5 to remove
any duplication.*

*Change(s) to Paper:* *We delete the paragraph (lines 414-421) "The difference in anthropogenic
fossil fuel emission budget (AFFEB) between the mitigated pathway and the control simulation
gives an estimate of the Mitigation Potential (MP) of the mitigation strategy. The IM-1.9
scenario relies on a combination of BECCS, reduced emissions from deforestation and
degradation (REDD), and reforestation of degraded forest areas to achieve a 1.5°C climate
target, while IM-BL has limited land-based mitigation via moderate levels of REDD and AR. We
develop an additional land-based mitigation scenario (IM-1.9N) where any bioenergy cropland
area in IM-1.9 is replaced by natural vegetation (Sect. 2.3). We then derive an optimal land-
based mitigation pathway in a post-processing step by selecting the option, (a) BECCS or (b)
natural vegetation, which has the more positive impact on the AFFEB in each grid cell (Sect.
2.4.1)"*

Lines 384-387: This paragraph needs some editing for clarity. The analysis you are doing is focused on
the climate sensitivity of mitigation options, not an analysis of their economics or how that would
change under different temperature targets. I don't think you can say that these are "worthwhile
mitigation approaches" given your analysis. But, you can say that across the range of temperatures
you analyzed there is no noticeable difference in the potential or performance of these mitigation
strategies.

*Response 2.10:* *We accept that worthwhile has a value judgement. We amend the text*

*"Despite the substantial differences in the absolute AFFEBs for the 1.5° and 2°C targets, the
mitigation potential of the $CH_4$ and land-based strategies is similar for the two temperature
scenarios considered. This similarity suggests that the*  mitigation
strategies *are robust to the target temperature; whether the international community aims for
the 1.5° or 2°C target, afforestation, reforestation, reduced deforestation and $CH_4$ mitigation are
all*  beneficial *mitigation approaches".*

*Change(s) to Paper:* *We make the above changes at lines 457-461.*

Lines 464-465: Why are those regions different?

> *Response 2.11: We use information in the cited paper by Postel et al. (2016) to assume that "only 5% of the total runoff is accessible for the Brazil, Russia and Canada IMAGE regions and 40% elsewhere". Postel et al. adjusted the total runoff for geographic and temporal inaccessibility. Specifically, the Amazon River "accounts for 15% of global runoff (11). It is currently accessible, however, to -25 million people (12) - 0.4% of world population-and no massive expansion of irrigation is likely that would warrant major diversions from it. We thus consider 95% of its flow inaccessible". For rivers in the boreal zone, "The final subtraction is for the remote rivers of North America and Eurasia, 55 of which have no dams on their main channels (13). Most of this river flow is in tundra and taiga biomes that are remote from population centers. The combined average annual flow of these northern untapped rivers is 1815 km³/year, and we subtract 95% of it". We will add a sentence about the adjustments made to the total runoff for geographic and temporal inaccessibility.*

> *Change(s) to Paper: We amend lines 543-544: "Following Postel et al. (1996), we derive the accessible runoff,  using their assumptions that only 5% of the total runoff is geographically and/or temporally accessible for the Brazil, Russia and Canada IMAGE regions, and 40% elsewhere*

Lines 468-471: What does "take the water requirements" mean? Do you use the water per unit of output from those studies and apply it to the IMAGE outputs? Or do you use the total water from those studies? If the latter, is it consistent? Also, does this mean you use the RCP2.6 water for the baseline and 1.9 simulations here? Is that water from the IMAGE-LPJmL model (which you note is low) or are you overwriting the IMAGE-LPJmL with the values from those papers?

> *Response 2.12: We take the water requirements for agricultural irrigation (Rost et al., 2008) and for other human activities (Bijl et al., 2016) (Table 4), as the total water withdrawal for each IMAGE region from the IMAGE-SSP2-RCP2.6 scenario. We use this for all our scenarios and add to this the additional water requirements for BECCS in the relevant scenarios. We acknowledge that this introduces new caveats and will add these to those already listed (lines 474-478).*

> *Additional comment: We do not need the baseline (or control, "CTL") scenario for this evaluation, We only derive the water demand for the optimised land-based mitigation scenario. The IMAGE-SSP2-RCP2.6 scenario used for the water demand for agricultural irrigation, energy generation, industry and domestic usage is compatible with the 2°C warming target. We assume that it can also be used the 1.5°C warming target.*

> *Change(s) to Paper: We amend lines 548-552, showing added and deleted text: "We use  the water withdrawals for each IMAGE region  given in the IMAGE-SSP2-RCP2.6 scenarios for the water  demand for agricultural irrigation (Rost et al., 2008) and for other human activities, such as energy generation, industry and domestic usage (Bijl et al., 2016), between 2015 and 2100 (Tab. 4a and 4b), We assume the same water demands from these sectors for both the 1.5°and 2°C warming targets".*  `Commented [HGD3]: Moved to next paragraph, as caveat`

> *We add lines 565-567 "We also note from Bijl et al. (2016) that the water demand for irrigation, derived using the coupled IMAGE-LPJmL models, is low compared to other estimates in the literature. Higher water demand for irrigation existing agriculture would be an additional constraint on the water available for BECCS".*

Lines 472-482: It would be nice to have one sentence in this paragraph reporting the quantitative results before you go through the caveats.

*Response 2.13: Reviewer 1 has suggested adding the BECCS water demand and percent of available used to Table 4. We will use this information to add a sentence or two with quantitative results.*

*Change(s) to Paper: We add the following text (lines 555-559) "… for the optimised land-based mitigation. Tables 4a and b show the additional water requirements of BECCS calculated for 2060 and 2100, respectively, for the 2°C warming target. We find that the additional demand for BECCS would lead to an exceedence (or use >90%) of the available water for the Oceania and Rest of Southern Africa regions. We also find that the additional demand for BECCS is greater than the total water withdrawals from anthropogenic activities for the Canada and Brazil IMAGE regions"*

Figure 3: Should the titles of panels d, f, and h say "Carbon Dioxide" instead of "Methane"? In general, I find the naming in this figure difficult since you have 1.5C and 2C on a baseline panel and 2C on 1.5C panels.

*Response 2.14: The figure is intended to show key data inputs for or differences between the model runs to help inform reader's understanding of the paper. The titles of the panels are correct but we accept the figure and panels need careful reading. We will amend the figure to make it clearer (and potentially split the figure into two as part of the restructuring of Section 2.3-2.5).*

*Change(s) to Paper:*

*We define a consistent set of model scenarios at the beginning of the revised Section 2.3 (lines 245-250) "We undertake a control run and other simulations with anthropogenic $CH_4$ mitigation or land-based mitigation, stabilising at either 1.5°C or 2.0°C warming without a temperature overshoot. We denote the control run as "CTL", the anthropogenic $CH_4$ mitigation scenario as "$CH_4$", a land-based mitigation scenario using BECCS as "BECCS" and a variant land-based scenario focussing on AR as "Natural". We also undertake runs combining the $CH_4$ and land-based mitigation scenarios ("BECCS+$CH_4$" and "Natural+$CH_4$") to determine if there are any non-linearities when we combine these mitigation scenarios".*

*Figure 6 (part of the original Figure 3) shows the modelled evolution of the atmospheric concentrations of $CH_4$ (the left-hand panels: a, c and e) and of $CO_2$ (right-hand panels: b, d and f) for the pairs of model runs: "CTL" vs "$CH_4$" (upper row), "BECSS" vs "BECCS+$CH_4$" (middle row) and "Natural" vs "Natural+$CH_4$". The title of each panel is the pair of model runs. The subtitle and legend use the same set of model run abbreviations (i.e., "CTL", "$CH_4$", "BECSS", "Natural", "BECCS+$CH_4$" and "Natural").*

*See also Response 1.26, which address the same point.*

Figure 7: Some of the detail in the caption would be good to include in the figure. In particular, the difference between panels a & c OR b & d.

*Response 2.15: These are a pair of plots for different BECCS efficiency scale factors (Panels a & b are for $\kappa$ =1 and Panels c & d for $\kappa$ = 3). The caption can be shortened by deleting "a & b are for the standard JULES BECCS productivity and efficiency (κ=1, Sect. 2.4.3), in c & d the BECCS productivity and efficiency uses κ=3" and adding BECCS $\kappa$ =1 and $\kappa$ =3 to the figure.*

*Change(s) to Paper: We add (1) BECCS Scale Factor ($\kappa$) = 1 above panels (a) and (b); (2) BECCS Scale Factor ($\kappa$) = 3 above panels (c) and (d). We shorten the figure caption, showing the additions and :*

*"Figure 9 |  Panels (a & c): The allowable anthropogenic fossil fuel*

*emission budgets (AFFEBs; GtC) for the control (grey), CH₄ mitigation (purple), land-based mitigation (green), coupled methane and land-based mitigation (orange) and the linearly summed methane and land-based mitigation (brown), for 2 temperature pathways asymptoting at 1.5°C (left) and 2.0°C (right). (b & d) The mitigation potential (GtC) as the increase in AFFEB from the corresponding control run.  The breakdown of the each AFFEB and mitigation potential by the changes in the carbon stores is also shown: atmosphere (pale yellow), ocean (light blue), land (dark green) and BECCS (gold) is included alongside each bar".*

*From Response 1.28, we add to the end of the caption "Note that the gain in the land carbon store for the CH₄ scenario is shown as a reduction from -70.8 GtC in the control run to -1.4 GtC in the "CH₄" mitigation option (median of ensemble)".*

Figure 9: This figure is pretty busy. Do you need the map? Or if you want the map, do you need the colors on the map? It is hard to see the bars and axes.

*Response 2.16: Figure 9 presents the regional mitigation options superimposed on a map of the IMAGE regions. The colours identify the different IMAGE regions. We accept that the colours used for specific regions make it hard to distinguish the bar charts for that region (the colours for the mitigation options are consistent with the colour scheme used throughout the paper for the methane and land-based mitigation). We will amend the figure by either using grey shading for the IMAGE regions or alternatively placing the bar charts around the edge of the plot.*

*Change(s) to Paper: We use a grey scale for the IMAGE regions in the new Figure 11 (page 41) and also for Supporting Information, Figure SI.4.*

[revised manuscript text omitted]

**Commented [HGD15]:** Reviewer comment 1.6: DONE, added text and paragraph above about the CO2 radiative forcing/

**Commented [HGD16]:** Reviewer comment 1.7: DONE

**Commented [HGD17]:** Reviewer comment 1.8: DONE

**Commented [HGD18]:** Reviewer comment 1.9: DONE, restructure moved first 2 paragraphs of Section 2.5 to Section 2.3.

the century. The intensity of resource and energy use declines. We define the upper and lower limits of anthropogenic mitigation as the lowest (RCP1.9, denoted "IM-1.9") and highest ("baseline", denoted "IM-BL") total radiative forcing pathways, respectively, within the IMAGE SSP2 ensemble (Riahi et al., 2017). We denote the RCP1.9 pathway as IM-1.9 and the "baseline" pathway as IM-BL.

**2.3.1 Methane: baseline and mitigation scenario**

The anthropogenic $CH_4$ emission increase from 318 Tg per annum$yr^{-1}$ in 2005 to 484 Tg $yr^{-1}$per annum in 2100 in the IMAGE SSP2 baseline scenario, but fall to 162 Tg $yr^{-1}$per annum in 2100 in the IMAGE SSP2 RCP1.9 scenario. The sectoral $CH_4$methane emissions in 2005 (Energy Supply & Demand: 113; Agriculture: 136; Other Land Use (primarily burning): 18; Waste 52, all in Tg $yr^{-1}$per annum) are in agreement with the latest estimates of the global methane cycle (Saunois et al., 2019). As summarised in Supplementary Information, Table SI.1, the reduction in $CH_4$ emissions from specific source sectors is achieved as follows: (a) coal production by maximising $CH_4$methane recovery from underground mining of hard coal; (b) oil/gas production & distribution, through control of fugitive emissions from equipment and pipeline leaks, and from venting during maintenance and repair; (c) enteric fermentation, through change in animal diet and the use of more productive animal types; (d) animal waste by capture and use of the $CH_4$methane emissions in anaerobic digesters; (e) wetland rice production, through changes to the water management regime and to the soils to reduce methanogenesis; (f) landfills by reducing the amount of organic material deposited and by capture of any $CH_4$methane released; (g) sewage and wastewater, through using more wastewater treatment plants and also recovery of the $CH_4$methane from such plants, and through more aerobic wastewater treatment. The levels of reduction vary between sectors, from 50% (agriculture) to 90% (fossil-fuel extraction and delivery). The abatement costs are between US$ 300-1000 (1995 US$) (Supplementary Information, Table SI.1). Figure 4 presents the IMAGE baseline and RCP1.9 $CH_4$ emission pathways globally and for selected IMAGE regions, including the major-emitting regions of India, USA and China (Supplementary Information, Figure SI.1 shows the emission pathways for all 26 IMAGE regions). These two methane emission pathways define our "CTL" and "$CH_4$" scenarios, respectively.

**2.3.2 Land-based mitigation: baseline, BECCS and Natural scenarios**

The IM-BL LULUC scenario assumes (a) moderate land-use change regulation; (b) moderately effective land-based mitigation; (c) the current preference for animal products; (d) moderate improvement in livestock efficiencies; and (e) moderate improvement in crop yields (Table 1 in (Doelman et al., 2018)). It represents a control scenario within which agricultural land is accrued to feed growing populations associated with the SSP2 pathway and with no deployment of BECCS. Three types of land-based climate change mitigation are implemented in the IMAGE land use mitigation scenarios (Doelman et al., 2018): (1) bioenergy; (2) reducing emissions from deforestation and degradation (REDD or avoided deforestation); and (3) reforestation of degraded forest areas. For the IM-1.9 scenario, there are high levels of REDD and full reforestation. The scenario assumes a food-first policy (Daioglou et al., 2019) so that bioenergy crops are only implemented on land not required for food production (e.g., abandoned agricultural crop land, most notably, in central Europe, southern China and eastern USA,

**Commented [HGD19]:** Reviewer comment 1.10: DONE

**Commented [HGD20]:** Reviewer comment 1.11: DONE, re-ordered Figures and updated figure references.

[revised manuscript text omitted]

**Commented [HGD24]:** Reviewer comment 2.8: DONE in next section. This comment is on the energy production from BECCS. It sits better but in the next section than here.

**Commented [HGD25]:** Reviewer comment 1.13: DONE

**Commented [HGD26]:** Reviewer comment 2.8: DONE, bioenergy (in EJ or Mt per year) produced.

Figure 7 presents maps of the scaling factor required for BECCS to be the preferable mitigation option, as opposed
to natural land carbon uptake, for each grid cell for warming of 1.5°C or 2°C. There are large factors in the northern temperate
and boreal regions, parts of Africa and Australia. As discussed in Harper et al. (2018), this follows from the loss of soil carbon
in the tropics and at high northern latitude leading to long recovery or payback times (10-100+ years and >100 years,
respectively, Fig. 6(c) in their paper). The payback time is however insignificant when bioenergy crops replace existing
agriculture, for example in Europe and eastern North America.

Additionally, we define a threshold efficiency factor, $\kappa^*$, which represents the required BECCS efficiency for BECCS to
be a preferable mitigation strategy for a given grid-cell, i.e.:

$$\kappa^* = \frac{\Delta C_{IM1.9N}^{land} - \Delta C_{IM1.9}^{land}}{BECCS_{IM1.9}} \tag{12}$$

This increased efficiency can be considered to be the additional bioenergy harvest (H) and/or the reduced carbon losses
from farm to storage needed to pay back the carbon debt accrued due to land-use change (since carbon removed via BECCS
= Hε, where ε is the assumed efficiency factor for farm to storage carbon conservation and H is the simulated biomass harvest).
In addition, $\kappa^*$ implies a new threshold (or break-even) level of BECCS:

$$BECCS^* = \kappa^* * BECCS_{IM1.9} \tag{13}$$

In other words, BECCS$^*$ is equivalent to the carbon loss due to the land use change to grow the bioenergy crops. To assess
the feasibility of meeting this break-even level of BECCS, we calculate the harvest (H$^*$) that would be needed if carbon losses
are to be minimised, i.e. by increasing ε from 0.6 to 0.87, and assuming in Eq. 13 that:

$$BECCS^* = 0.87 \, H^* \text{ and } BECCS_{IM1.9} = 0.60 \, H$$

So:

$$H^* = \kappa^* * \frac{0.6}{0.87} * H \tag{14}$$

We discuss this further in Sect. 3.2.

**2.4  Runs**

 (Riahi et al., 2017;Doelman et al., 2018)

**Commented [HGD27]:** Reviewer comment 1.14: DONE, first two paragraphs integrated into Section 2.3. Third paragraph deleted, as per comment 2.9.

[revised manuscript text omitted]
 as the total the water withdrawals for each IMAGE region from given in the IMAGE-SSP2-RCP2.6 scenarios for the water requirements demand for agricultural irrigation (Rost et al., 2008) and for other human activities, such as energy generation, industry and domestic usage (Bijl et al., 2016), between 2015 and 2100 (Table 4a and 4b). We note Bijl et al. (2016)that the irrigation water withdrawal, derived using the coupled IMAGE-LPJmL models, are low compared to other estimates in the literature. We assume the same water demands from these sectors for both the 1.5°and 2°C warming targets.

Figure 14Figure 13 compares the accessible water with the water demand for BECCS and other human activities for the eight regions, which that produce a substantial amount of BECCS: Canada, USA, Brazil, Europe, Russia, China, Southern Africa and Oceania for the optimised land-based mitigation. Table 4a and b show the additional water requirements of BECCS calculated for 2060 and 2100, respectively, for the 2°C warming target. We find that the additional demand for BECCS would lead to an exceedence (or use >90%) of the available water for the Oceania and Rest of Southern Africa regions. We also find that the additional demand for BECCS is greater than the total water withdrawals from anthropogenic activities for the Canada and Brazil IMAGE regions. Our estimates represent a maximum possible water usage for BECCS as (i) the SSP2 scenario used already accounts for the lower power generation efficiencies and hence higher water requirements in switching from fossil fuels to bioenergy crops (which could be up to 20-25%) and (ii) the figure used for the CCS component does not allow for future technological improvements in water use. For example, Fajardy and Mac Dowell (2017) indicate a 30-fold reduction in water use when changing from a once-through to a recirculating cooling tower. Our results are less severe than other studies considering BECCS water requirements (Séférian et al., 2018;Yamagata et al., 2018), because the carbon removed by BECCS

**Commented [HGD36]:** Reviewer comment 2.11. DONE, clarified accessible runoff

**Commented [HGD37]:** Reviewer comment 1.23: DONE, BECCS water demand added to Table.

**Commented [HGD38]:** Reviewer comment 2.12: DONE, clarified irrigation as part of water requirements. DONE, modify new Figures 13 and 14

**Commented [HGD39]:** Reviewer comment 1.21. DONE. Text changed.

**Commented [HGD40]:** Reviewer comment 2.13: DONE, added quantitative statements

565 in this study (30 GtC) is already limited to regions where it is more beneficial to the AFFEB than forest-based mitigation options. We also note from Bijl et al. (2016) that the water demand for irrigation, derived using the coupled IMAGE-LPJmL models, is low compared to other estimates in the literature. Higher water demand for irrigation existing agriculture would be an additional constraint on the water available for BECCS. Nevertheless, our results indicate that the additional water demand for BECCS would have large impacts in half of the regions substantially invested in BECCS: Oceania, Rest

570 of South Africa, Brazil and Canada .

**Commented [HGD41]:** Reviewer comment 2.12: DONE, clarified irrigation as part of water requirements. DONE, modify new Figures 13 and 14

**Commented [HGD42]:** Reviewer comment 1.22: DONE, added text earlier in paragraph

[revised manuscript text omitted]

**Commented [HGD46]:** Reviewer comment 1.25: DONE, split original Figures 3

Reviewer comment 1.26: DONE, checked consistency of scenarios.

Reviewer comment 2.14: DONE, checked figure titles & captions.

[Figure]

**Figure 4 | Time series of annual methane emissions between 2005 and 2100 from all and selected anthropogenic sources according to the IMAGE SSP2 Baseline (solid lines) and SSP2-RCP1.9 (dotted lines) scenarios, globally and for selected IMAGE regions, with total emissions in black, energy sector in red, agriculture-cattle in blue, agriculture-rice in green and waste in magenta. Note the y-axes have different scales for clarity.**

885

888

889 (a) IMAGE Brazil Region

890

891 (b) IMAGE Russia Region

892

893

894 **Figure 5 | Time series of the land areas (in Mha) calculated for trees and prescribed for agriculture (including bioenergy crops) and bioenergy crops for**
895 **the 'BECCS' (orange) and'Natural, (green), as a difference to the baseline scenario (IM-BL), for Brazil  (panel a) and the Russia (panel**
896 **b) IMAGE regions between 2000 and 2100. The dotted lines are the median and the spread the interquartile range for the 34 GCMs emulated and 4**
897 **factorial sensitivity simulations.**

898

[Figure]

899

Figure 6 | (a, c, e) Time series of the ensemble median atmospheric CH₄ concentrations (with interquartile range as spread) derived for each temperature profile for the scenarios: (a) "CTL" and "CH₄", (c) "BECCS" and "BECCS+CH₄", (g) "Natural" and "Natural+ CH₄". (d, f, h) show the corresponding time series for the atmospheric CO₂ concentrations.

903

**Commented [HGD47]:** Reviewer comment 1.25: DONE, split original Figures 3

Reviewer comment 1.26: DONE, amended to ensure consistency of scenario notation.

Reviewer comment 2.14: DONE, amended figure titles & captions.

904 (a)

[Figure]

905

906 (b)

[Figure]

907

**Figure 7 | Scale factor required for BECCS to be the preferable mitigation option, as opposed to natural land carbon uptake. The data represents the median of the 136 member ensemble for the optimised land-based mitigation simulation. Panel (a) is for stabilisation at 1.5°C and panel (b) is for stabilisation at 2°C.**

911

912

913

914

915 **Figure 8 | The contribution to the allowable anthropogenic fossil fuel emission budget (AFFEBs, GtC) from the changes in the**
916 **different carbon stores (atmosphere, ocean, land and BECCS) for the various control and mitigation runs, illustrated using the**
917 **temperature pathways reaching 1.5°C without overshoot. The optimised land based and coupled mitigation options selects the land**
918 **use option, which maximises the AFFEB for each model grid cell. Note that the gain in the land carbon store for the CH₄ scenario is**
919 **shown as a reduction from -70.8 GtC in the control run to -1.4 GtC in the "CH₄" mitigation option (median of ensemble).**

920

**Commented [HGD48]:** Reviewer comment 1.27: DONE, added note about land-carbon store for CH4 scenarios (See also next figure).

921 BECCS Scale Factor (κ) = 1

922

923 BECCS Scale Factor (κ) = 3

[Figure]

924

925 **Figure 9 |**
926 Panels **(a & c)**: **The allowable anthropogenic fossil fuel emission budgets (AFFEBs; GtC) for the control (grey),**
927 CH₄ **mitigation (purple), land-based mitigation (green), coupled methane and land-based mitigation (orange) and the linearly**
928 **summed methane and land-based mitigation (brown), for 2 temperature pathways asymptoting at 1.5°C (left) and 2.0°C (right). (b**
929 **& d) The mitigation potential (GtC) as the increase in AFFEB from the corresponding control run.**
930  **The**

**Commented [HGD49]:** Reviewer comment 1.28: As 1.27.
DONE, added note about land-carbon store for CH4 scenarios
(See previous figure).

Reviewer comment 2.15. DONE, moved some of figure
caption.

[revised manuscript text omitted]

**Commented [HGD54]:** Tables 1 and 2 switched, with merging of Sections 2.5 into 2.3.

| Factorial Run | Abbreviation |
|---|---|
| 1. **Control**:
• IMAGE SSP2 baseline scenario
• Agricultural land accrued to feed growing populations associated with the SSP2 pathway. No deployment of BECCS
• Anthropogenic $CH_4$ emissions rise from 318 Tg $yr^{-1}$ per annum in 2005 to 484 Tg $yr^{-1}$ per annum in 2100
• IMAGE SSP2 baseline scenario for atmospheric concentrations of $CH_4$ and non-$CO_2$ radiative forcing | CTL |
| 2. **Methane mitigation**:
• IMAGE SSP2 RCP1.9 scenario for $CH_4$
• Agricultural land-use as in Control
• Anthropogenic $CH_4$ emissions decline from 318 Tg $yr^{-1}$ per annum in 2005 to 162 Tg $yr^{-1}$ per annum in 2100

[revised manuscript text omitted]

u) China
v) Korea Region
w) Japan
x) South East Asia
y) Indonesia

z) Oceania

Fig. SI.2a: Canada

[Figure]

Fig. SI.2b: USA

[Figure]

Fig. SI.2c: Mexico

[Figure]

Fig. SI.2d: Central America

[Figure]

Fig. SI.2e: Brazil

[Figure]

Fig. SI.2f: Rest of South America

[Figure]

Fig. SI.2g: Northern Africa

[Figure]

Fig. SI.2h: Western Africa

[Figure]

Fig. SI.2i: Eastern Africa

[Figure]

Fig. SI.2j: South Africa

[Figure]

Fig. SI.2k: Rest of Southern Africa

[Figure]

Fig. SI.2l: Western Europe

[Figure]

Fig. SI.2m: Central Europe

[Figure]

Fig. SI.2n: Turkey

[Figure]

Fig. SI.2o: Ukraine Region

[Figure]

Fig. SI.2p: Central Asia

[Figure]

[Figure]

Fig. SI.2r: Middle East

[Figure]

Fig. SI.2s: India

[Figure]

Fig. SI.2t: Rest of South Asia

[Figure]

[Figure]

Fig. SI.2u: China

[Figure]

Fig. SI.2v: Korea Region

120

[Figure]

Fig.  SI.2x: South East Asia

[Figure]

125

Fig. SI2.y: Indonesia

[Figure]

Fig. SI.2

[Figure]

Fig. SI.2y: Rest of South Asia

[Figure]

Fig. SI.2z: Oceania

140           (a)

(b)

(c)

**Fig. SI. 3 | Contribution of different mitigation options to the increase in allowable**
145  **anthropogenic fossil fuel emission budgets by IMAGE region to meet the 2°C target**.
The stacked bars represent the median methane mitigation potential (purple bars) and median
land-based mitigation potential (natural land uptake, green; BECCS, brown). Panel (a) is
based on a BECCS scaling factor of unity, (b) a BECCS scaling factor of 2 and (c) a BECCS
scaling factor of 3. The total (pink) shows the median and interquartile range for the 34
150  GCMs emulated and 4 factorial sensitivity simulations.

[Figure]

[Figure]

**Fig. SI.4 | Contribution of different mitigation options to the allowable anthropogenic carbon emission budgets by region**. The contribution to the allowable carbon emission budgets (GtC) between 2015 and 2100 for each of the 26 IMAGE IAM regions from methane mitigation (purple bars) and land-based mitigation options (green: natural land uptake; yellow: BECCS with κ = 3), for the temperature pathway stabilising at 2°C warming without overshoot. The bars and error bars respectively show the median and the interquartile range, from the 34 GCMs emulated and 4 factorial runs.

**Table SI.1** | Mitigation options, estimated maximum reduction potential and the accompanying marginal price for mitigation of different anthropogenic methane source sectors for 2050 and 2100 [based on Lucas et al., 2007].

| Source Sector | Mitigation option(s) | Max. possible reduction relative to baseline (%) | Marginal price of max. reduction (1995 US$/tC$_{eq}$) |
|---|---|---|---|
| Coal production | Maximising methane recovery from underground mining of hard coal | 90 (2050) 90 (2100) | 500 (2050) 500 (2100) |
| Oil/gas production & distribution | Control of fugitive emissions from equipment and pipeline leaks, and from venting during maintenance and repair. | 75 (2050) 90 (2100) | 300 (2050) 500 (2100) |
| Enteric fermentation | Change of animal diet and use of more productive animal types. | 50 (2050) 60 (2100) | 1000 (2050) 1000 (2100) |
| Animal waste | Capture and use of methane emissions through anaerobic digesters. | 50 (2050) 60 (2100) | 1000 (2050) 1000 (2100) |
| Wetland rice production | Changes to (1) the water management regime to reduce the period of anaerobic conditions in flooded fields; (2) the soils to reduce methanogenesis. | 80 (2050) 90 (2100) | 1000 (2050) 1000 (2100) |
| Landfills | (1) Reduced amount of organic material deposited in landfills; (2) capture of methane | 90 (2050) 90 (2100) | 500 (2050) 500 (2100) |
| Sewage and wastewater | (1) More wastewater treatment plants and also recovery of the methane from the plants; (2) More aerobic wastewater treatment. | 80 (2050) 90 (2100) | 500 (2050) 500 (2100) |
| Other anthropogenic sources | Note 1 | - | - |

160 **Note:** (1) These sources are either difficult to abate (e.g., land clearing for agricultural extension, and the use of traditional biomass for energy production and cooking) or are too small (e.g., methane emissions from industry, iron and steel production and the chemical sector).

**Reference**: Lucas, P. L., van Vuuren, D. P., Olivier, J. G. J. & den Elzen, M. G. J., 2007: Long-term reduction potential of non-CO$_2$ greenhouse gases. *Environmental Science & Policy* **10**, 85-103, doi: https://doi.org/10.1016/j.envsci.2006.10.007.

---

## Author Response (AR2)

"**Regional variation in the effectiveness of methane-based
and land-based climate mitigation options**"
Hayman et al. (ESD-2020-24)

**Author Response**

We thank the Editor for his interest in and for his comments on the paper. From some of the Editor's comments, it is clear that the Editor has not fully understood parts of the study design, assumptions and hence interpretation of our results. We acknowledge that these areas of the paper need attention.

Before proceeding to the specific comments, we make some general points (GP) about the study:

GP 1: **Previous work**: This paper brings together and builds on 3 separate papers that we have had published in the scientific literature (cited papers by Comyn-Platt et al., 2018; Collins et al., 2018 and Harper et al., 2018). We look to provide sufficient detail from those studies, while minimising any repetition.

GP 2: **Scenarios**: We will use the word "scenario" to define the control (or baseline) scenario and the 5 mitigation options that we investigated with the inverse version of IMOGEN-JULES. The 6 scenarios, which are listed in Table 1, are:
   1. Control or baseline (denoted "CTL")
   2. "$CH_4$" for the methane mitigation scenario
   3. "BECCS" for the land-based mitigation using bioenergy crops with carbon capture & storage
   4. "Natural" for the variant land-based mitigation scenario
   5. Coupled ("BECCS+$CH_4$") for the combined land-based (BECCS) and methane mitigation scenario.
   6. Coupled ("Natural+$CH_4$") for the combined land-based (Natural) and methane mitigation scenario.

   As described in Section 2.4.2, we generate an additional 2 scenarios in the post-processing optimisation of the land-based mitigation scenarios: (1) land-based mitigation (optimised) and (2) coupled (optimised). These are the scenarios referred to in, for example, Figures 8 and 9. We concede that we did not explicitly define these scenarios as such in Section 2.4.2. We now include these scenarios in Table 1. See also Response 1.

GP 3: **Prescribed and input data:** We use prescribed temperature pathways for the 1.5°C warming and 2°C warming targets taken from our previous work (cited paper Huntingford et al., 2017). The scenarios described above use specific datasets developed for the IMAGE SSP2 baseline and SSP2 1.9 scenarios: (a) time series of annual atmospheric methane concentrations; (b) time series of the radiative forcing by non-$CO_2$ greenhouse gases and other climate forcers; and (c) time series of the gridded annual land area assigned to agriculture, and within that to bioenergy crops.

   We use the time series of anthropogenic methane emissions in two of our post-processing scripts to attribute the global atmospheric carbon stores to the different IMAGE regions in those scenarios involving methane mitigation. We use the time series to generate Figures 11 and 12 (and Supplementary Information, Figures SI.3 and SI.4). See also Response 21.

GP 4: **Uncertainty**: For each of the 6 scenarios investigated using the inverse version of IMOGEN-JULES, we make 136 separate model runs for the 1.5°C warming target and a second set of 136 runs for the 2°C warming target. Within each ensemble of 136 runs, we emulate 34 of the CMIP5 climate and Earth system models. For each CMIP5 model, we make 4 runs (high & low $Q_{10}$ and high & low vegetation $O_3$ sensitivity), which we now denote "factorial" runs. We use these ensembles to determine the range or "uncertainty" in the derived carbon budgets,

specifically from climate change (as given by the 34 CMIP5 models) and from key land-surface processes.

For BECCS, there is an additional source of uncertainty from the productivity of the bioenergy crops and the assumptions about the losses from harvest to final long-term storage. We introduced a BECCS scale factor and investigated the effect of the scale factor on the derived carbon budgets in the post-processing step. We include this parameter uncertainty in the ranges given for the land-based mitigation scenarios involving BECCS.

GP 5: **1.5°C warming vs. 2°C warming targets:** We use the same input datasets for the scenario runs for the 1.5°C warming and 2°C warming targets, e.g., the gridded annual time series of areas assigned to agriculture, and within that the area assigned to bioenergy crops. This is in contrast to Integrated Assessment Model scenarios where, for example, a greater area of bioenergy crops would be needed to meet the 1.5°C warming compared to a 2°C warming target (Lines 56-61, version submitted in December 2020).

Figure 8 shows that the allowable fossil fuel carbon dioxide budget is, as expected, higher for the 2°C than the 1.5°C warming target. However, when we compare the mitigation potential of the mitigation scenarios, there is little difference between the two warming targets. If we were to make different assumptions about the deployment of BECCS in the two warming targets, this would affect the carbon budgets derived.

We give our Response and the Change(s) made to the paper and to the Supplementary Information, after each reviewer comment. The reviewer comments are in normal font, with our "Response" and "Change(s) to Paper" in **bold italics and indented**. The line numbers for the Editor's comments refer to the revised manuscript submitted in December 2020. The line numbers for the "Change(s) to Paper" refer to the track change versions of the paper and Supporting Information that are in the separate track change version of the paper.

**Editor Decision:**
Reconsider after major revisions (04 Jan 2021) by Steven Smith

**Comments to the Author:**
I've reviewed the revised paper and responses to referee comments. The paper is improved, however is still difficult to understand. The methodology is complex, combining forward modeling, inverse modeling, and scale factors, and needs to be more clearly explained.

The first major point of confusion regarding the methodology is the overall structure of the calculation with respect to how emissions, concentration, and forcing are calculated. This should be clarified in one place for the reader. This statement, Line 105, " Using a combination of calculated and prescribed time series of annual radiative forcings, we derive the atmospheric CO2 radiative forcing and hence its concentration, taking account of any land and ocean feedbacks." is unclear, but is central to the calculations done in the paper. I've searched through the paper but cannot find a succinct description of the methodology on this point. This should be added either in this section or soon after to clarify this for the reader (some specific comments below relate to this as well).

For example I infer from various parts of the paper that the following are exogenous: anthropogenic CH4 emissions and CH4 mitigation (from IMAGE), land-use patterns (except that later the paper discusses optimization of land-use between natural and biomass crops, which seems a bit inconsistent with this statement), and the assumed global mean temperature pathway. Land-use feedbacks on CO2 and CH4 emissions are endogenous to the models used in the paper. Its unclear how radiative forcing and concentrations of CO2 and CH4 are determined. It seems that some adjustment to the IMAGE CH4 concentrations is calculated, which leaves a residual CO2 forcing pathway that is inverted to obtain an anthropogenic fossil CO2 budget? These issues are central to the methodology of the paper and need to be better summarized for the reader. (see also comments below on Figure 3) Is total forcing also fixed to one time path, or is the inverse calcuation focused only on the temperature pathway?

> *Response 1:* *We note the Editor's comment on the improvements made to the paper but that the Editor still finds part of the paper unclear. In response, we have (1) re-written the section in the Introduction describing the study, (2) worked to make our "Approach and Methodology" clear and unambiguous; and (3) use Table 1 to list and provide details on our scenario simulations. In particular, Table 1 provides the notation used throughout the paper. All our text has been checked to ensure references to simulations are entirely consistent with the terminology of Table 1. See also Response 6 on nomenclature.*

> *Change(s) to the Paper: We amend the text (lines 94-160):*

[revised manuscript text omitted]

A second point that is difficult to follow is the methodology for how uncertainty is treated in the paper. One issue is that the paper alternates between discussing uncertainty in some sections, and then discussing factorial runs in others. It might be useful to be more consistent in terminology. The overall approach to addressing uncertainty needs to be more clearly stated early on in the paper (e.g. first part of section 2). There is a clear statement at the beginning of the conclusion section, but this needs to be made much sooner.

*__Response 2:__* *We refer to General Point 2. We now include a paragraph on our treatment of uncertainty at the end of the introduction to Section 2.*

*__Change(s) to the Paper:__ We add the following text, which is also included as the final paragraph of the "Change(s) to the Paper" for Response 1 above.*

Each of the scenarios investigated using the IMOGEN-JULES framework comprises 2 ensembles of 136 members, one ensemble for each of the warming targets. We make use of these ensembles to derive an "uncertainty" in the derived carbon budgets, specifically from climate change (as given by the 34 CMIP5 models) and from key land-surface processes (methane emissions from wetlands and the ozone vegetation damage). The climate change uncertainty comprises both the range of climate sensitivities of the CMIP5 models and the different regional patterns in the models. We use the median of the 136-member ensemble as the central value to derive the carbon budgets and the interquartile range (25-75%) for the uncertainty.

On a specific point, the paper mentions two values of Q10 used as uncertainty bounds (~line 145). Are only the bounding cases used (the text implies this as currently written)? In that case, what do the central values presented in later figures (for example bars figure 9) represent?

*__Response 3:__* *The IMOGEN-JULES runs only make use of the bounding values of $Q_{10}$. As indicated in Response 2, the central value in Figure 8-9, 11-14 is the median of the 136-member ensemble. In Figure 9 (and others), the points represent the results from individual ensemble members. We also refer the Editor to General Point 4 and Response 4 below.*

*__Change(s) to the Paper:__ The required changes are included in Response 2 above and Response 4 below.*

The results of the factorial experiments do not really seem to be fully explored in the discussion of results overall. It would strengthen the paper substantially if these could be more fully examined. For example, what is the range in contributions of each specific component? For example the modeling of methane from wetlands is discussed, but what is the contribution of wetland methane to the allowed carbon budget? Same for ozone damage, etc. What uncertainty factors (and assumed BECCS efficiency) impact these the most? There are some aggregate results shown in Figure 8, but no detailed results are presented or discussed in the paper or the supplement.

*__Response 4:__* *We have taken this request particular seriously and include additional analysis to derive the contribution from climate change and different spatial patterns in the 34 GCMs emulated and the 4 land process factorial runs..*

*__Change(s) to the Paper:__ We add the following text to the paper (lines 526-544):*

[revised manuscript text omitted]

The way that uncertainty in BECCS is considered is difficult to understand since, by the way the calculations are structured, some uncertainties are addressed with a factorial design (natural or "BECCS" as an option), while uncertainty in BECCS potential is addressed by setting the value if "k" (Kapa). There are a few results presented with different values of "K", but most results seems to be based on one value of this scale factor.

> ***Response 5:*** *We now define as scenarios the land-based mitigation options "BECCS" and "Natural" (i.e., no BECCS) (see General Point 2). We introduce the BECCS scale factor $\kappa$ into the post-processing of the IMOGEN-JULES ensemble runs to provide a single parameter to assess the sensitivity to (a) bioenergy productivity and (b) losses from harvest to final storage. We make clear in Section 3.1 that we have used a value of $\kappa = 1$. We will however clarify how the selection of $\kappa$ affects the uncertainty. See also General Point 2 and Reponses 1, 2, 27 and 29.*

> ***Change(s) to the Paper:*** *We amend the text (lines 479-486):*

**3.1 Global Perspective**

We calculate the anthropogenic fossil fuel emission budget to limit global warming to a particular temperature target as the sum of the changes in the carbon stores of the atmosphere, land (vegetation and soil) and ocean between 2015 and 2100 (Sect. 2,4,1, Eq. 5 and 6). We use a BECCS scale factor ($\kappa$) of unity. We present in Fig. 8 the median and  spread of the AFFEB (as box and whiskers) from the 136-member ensemble, and the individual GCM/ESM contributions to the AFFEBs from the four carbon pools shown (points), for each of the main scenarios modelled using the IMOGEN-JULES or derived in the post-processing optimisation step (see Table 1 for description of the scenarios).

> ***We include the following text (lines 700-703) with the Change(s) to the Paper for Response 27 (see later).***

We investigate the efficacy of our "BECCS" scenario by increasing the productivity of BECCS (using a scale factor $\kappa$). From comparison with observed bioenergy crop yields, we argue that the scale factor could be between 1 and 3. We use this range of $\kappa$ as an additional source of uncertainty on the land-based mitigation potential.

There is also some inconsistency in nomenclature:

Table 1 describes a scenario with "Land-based mitigation, including BECCS" and "Land-based mitigation with no BECCS". This is clear, although there is no mention of the assumed BECCS K value, which should be added.

However then in Figure 8, there are a number of other options presented. Such as "Land-based mitigation (Optimized)" and "Land-based mitigation (Natural)" - these should both be identified in Table 1. Then here are "Coupled" scenarios shown that don't correspond with any of the abbreviations given in Table 1.

Then, further, in Figure 9 there is no BECCs scenario presented, only "Land based mitigation", but evidently land-baed mitigation that also contains BECCS (with K =1 and K = 3), which also does not correspond to any label in Table 1. (It seems that these are two variants of "Land-based mitigation including BECCS"?)

The authors need to throughly go through the paper, tables, figures, and supplement and harmonize nomenclature to avoid confusing readers.

*Response 6:* *We refer to General Point 2 where we describe the 6 scenarios investigated using the IMOGEN-JULES framework. With respect to the editor, Table 1 refers to a scenario "4. Land-based mitigation with no BECCS (Natural)", which we denoted "Natural" in that table. We accept that "Land-based mitigation (Optimized)" was not explicitly defined. As requested, we will ensure that a consistent set of terminology is used.*

*In response to the Editor's comment, we add the 2 post-processing scenarios to Table 1 for completeness. As indicated in Response 1, we refer to Table 1 in relevant places to assist the reader.*

*Change(s) to the Paper: We list below the line numbers where we make changes to the manuscript, the line numbers refer to the track change version of paper included in this Author response.*

- *Lines 382, 391, 484, 593, 667, 734 and Table 1: "factorial" to "scenario"*
- *Figure 11: "34 GCMs and 4 factorial runs" to "136-member ensemble"*
- *We make additional reference to Table 1 at lines 325, 485, 684.*
- *Equations renumbered following insertion of new equations 3 and 4*
- *Equations 13-16: "IM1.9" replaced by "BECCS" and "IM1.9N" by "Natural"*
- *Table 3 with $CH_4$ scale factors moved to Supplementary Information*
- *New Table 3 on uncertainty analysis (see also Response 4)*

**Further specific comments:**

Abstract - I presume these results are specific for the IMAGE SSP2 scenario. This should be mentioned.

*Response 7:* **We add "for the SSP2 pathway" to the Abstract.**

*Change(s) to the Paper: We amend lines 24-25:*

 We use consistent data and socio-economic assumptions from the IMAGE integrated assessment model

for the second Shared Socioeconomic Pathway (SSP2)*"*

Line 90
"and methane climate feedbacks) in a climate/Earth System modelling framework to quantify the unrealised potential from the mitigation of land-based options and anthropogenic CH4 sources."

I don't see how the authors can claim here (and elsewhere) that methane mitigation has not been examined. There is substantial literature on methane migration in the context of temperature and forcing targets. Further, all the IAM scenarios such as the SSPs contain representations of anthropogenic methane mitigation, including the IMAGE scenarios from which this paper draws. Clarify this and make sure previous work is properly cited.

For example also the statement here "Although the primary challenge remains mitigation of fossil fuel emissions, these results highlight the unrealised potential of these mitigation options to make the

Paris climate targets more achievable" does not seem to be warranted. Most of the scenarios in the literature, and certainly the SSPs, all include methane mitigation.

*__Response 8:__ We agree that there is an extensive literature on methane mitigation and we discuss some of this in the Introduction (76-85). Both in the scientific literature and in the IAM scenarios, the multiple benefits of methane mitigation are clear.*

*We are using "unrealised" in the sense that society has not yet acted on the potential of methane mitigation. We know that atmospheric methane concentrations are continuing to increase and are currently following a path consistent with the one of the high SSP5 emission scenarios. We will remove "unrealised" but we stand by our results in this and our previous paper (cited paper Collins et al., 2018) that methane mitigation makes the Paris climate targets more achievable (and is probably essential).*

*__Change(s) to the Paper:__*
- *__Abstract (line 38-39):__*

Although the primary requirement remains mitigation of fossil fuel emissions, our results highlight the  potential for the mitigation of $CH_4$ emissions to make the Paris climate targets more achievable.

- *__Introduction (line 94-98). The sentence below has been replaced as part of the "Change(s) to Paper" for Response 1.__*

~~For the first time, we combine these elements (land-based mitigation, anthropogenic CH4 mitigation, and natural carbon and methane climate feedbacks) in a climate/Earth System modelling framework to quantify the unrealised potential from the mitigation of land-based options and anthropogenic CH4 sources. In contrast to previous studies, we use a process-based land surface model to assess these mitigation options by region, yielding policy-relevant information on the optimal mitigation strategy.~~ This paper models the potential for land-based mitigation of greenhouse gases to contribute to meeting the Paris targets of limiting global warming to 1.5°C and 2°C respectively. Specifically, we investigate the effectiveness of mitigation of anthropogenic methane emissions and land-based mitigation (e.g., implementation of BECCS and AR), combining results from three recent papers (Collins et al., 2018; Comyn-Platt et al., 2018; Harper et al., 2018).

- *__Section 3.1 (line 514-515)__*

Although the primary challenge remains mitigation of fossil fuel emissions, these results highlight the  potential of these mitigation options to make the Paris climate targets more achievable.

Line 141 not clear what "(the depth of zero annual amplitude)" means.

*__Response 9:__ This term is widely used in the permafrost community to refer to the depth below the surface, where seasonal changes in ground temperature are negligible (≤0.1 °C).*

*__Change(s) to the Paper: We add the following to line 186-187:__*
we diagnose permafrost wherever the deepest soil layer is below 0°C (assuming that this layer is below the depth of zero annual amplitude, i.e. where seasonal changes in ground temperature are negligible (≤0.1 °C)).

Line 205 This " The increased/reduced atmospheric CH4 concentration will have a corresponding faster/slower atmospheric decay rate than the prescribed concentration pathway. We account for this following the approach of Cubasch et al. (2001). Related changes in atmospheric radiative forcing, in response to altered atmospheric 210 CH4 concentrations, are calculated using the formulation from Etminan et al. (2016). We also include the indirect effect of these CH4 emission changes on the forcing by tropospheric ozone and stratospheric water vapour by multiplying the CH4 forcing by 1.65, based on Myhre et al. (2013)."

is unclear and seems to involve quite a number of assumptions and simplifications that should be further detailed and tested (likely largely in the supplement). First, it is not clear what it means to say "following the approach of Cubasch et al." given that this is a reference to the entire WG I IPCC report. More details should be provided. Changes in wetland emissions are, according to the author's own equations, non-linearly related to temperature. It is not clear at all that this can be represented as a "faster/slower atmospheric decay rate", whatever that means.

It is not clear that "tropospheric ozone and stratospheric water vapour" can be accurately represented by a single multiplicative factor applied to CH4 forcing. Methane concentrations and tropospheric ozone are linked through a number of mechanisms, with tropospheric ozone forcing in Meinshausen etal 2011, for example, being tied more closely to methane emissions not forcing, and with methane and tropospheric ozone forcing linked through changes in atmospheric oxidation capacity. Given that methane is a central focus of the paper, the representation of methane in the atmosphere should be better described. Use of a constant multiplier might introduce some distortions in the results.

> *Response 10:*  *We give more details of our treatment of methane and provide justification. We refer to the extensive uncertainty analysis that we undertook in our earlier study on the climate-methane feedback from wetlands (cited paper by Gedney et al. 2019). We provide a summary in the present paper. We do not think that further testing is therefore needed.*
>
> *The Editor is correct. Wetland emissions are expected to increase significantly with climate change, as discussed in the cited paper by co-author Gedney et al. (2019). This was a key reason for adjusting the IMAGE time series of atmospheric concentrations of methane and its radiative forcing. The IMAGE time series are based on constant natural emissions of 250 Tg per annum.*
>
> *We follow the accepted practice for citing chapters in IPCC Assessment Reports. We now however give the specific page in the IPCC report. The Cubasch et al. reference is to Chapter 9 in the Third Assessment Report. In revising the manuscript, we no longer use this reference.*
>
> *Change(s) to the Paper: We rewrite the paragraph (lines 252-292):*
>
> Our simulations include a CH$_4$ feedback system that captures the climate impacts on CH$_4$ emissions from natural wetland sources. The approach used here follows Comyn-Platt et al. (2018) and Gedney et al. (2019), where the prescribed atmospheric CH$_4$ concentrations, which assume a constant annual wetland CH$_4$ emission (van Vuuren et al., 2017), are modified using the anomaly in the modelled annual wetland CH$_4$ emission. The increased/reduced atmospheric CH$_4$ concentration will have a corresponding faster/slower atmospheric decay rate than the prescribed concentration pathway. We account for this following the approach of Cubasch et al. (2001). Related changes in atmospheric radiative forcing, in response to altered atmospheric CH4 concentrations, are calculated using the formulation from Etminan et al. (2016). We also include the indirect effect of these CH$_4$ emission changes on the forcing by tropospheric ozone and stratospheric water vapour by multiplying the CH$_4$ forcing by 1.65, based on Myhre et al. (2013). The atmospheric CH$_4$ concentrations available from the IMAGE database (see Sect. 2.3.1) assume a constant annual wetland CH$_4$ emission (van

Vuuren et al., 2017). However, these emissions have interannual variability and a positive climate feedback (e.g., Comyn-Platt et al., 2018; Gedney et al., 2019), and their correct representation is a central part of our study. We follow the same approach that we used in our previous studies (Collins et al., 2018; Comyn-Platt et al., 2018; Gedney et al., 2019). As the IMOGEN-JULES modelling framework does not have an explicit representation of the atmospheric chemistry of methane, we represent the oxidation and hence loss of $CH_4$ by a single lifetime ($\tau$).

$$\frac{d([CH_4] - [CH_4]_{IMAGE})}{dt} = C \left\{ \sum F [CH_4] - \sum F [CH_4]_{IMAGE} \right\} - \frac{[CH_4] - [CH_4]_{REF}}{\tau} \qquad (3)$$

where $[CH_4]$ and $[CH_4]_{IMAGE}$ are the atmospheric methane concentrations using our new wetland-based, time varying ($F[CH_4]$) and the constant IMAGE ($F[CH_4]_{IMAGE}$) wetland emissions, respectively. Here parameter C is a constant to convert from Tg $CH_4$ to a mixing ratio in parts per billion by volume (ppbv). Further, higher atmospheric concentrations of $CH_4$ and its oxidation product (carbon monoxide) lower the concentration of hydroxyl radicals, the major removal reaction for $CH_4$, thereby increasing the atmospheric lifetime of $CH_4$. Conversely, lower $CH_4$ concentrations will shorten its atmospheric lifetime. We take account of this feedback of $CH_4$ on its lifetime ($\tau$), using Eq. 4 (Collins et al., 2018; Comyn-Platt et al., 2018; Gedney et al., 2019) as:

$$\ln (\tau/\tau_o) = s.\ln ([CH_4]/[CH_4]_o), \text{ i.e., } \tau = \tau_o \exp (s [CH_4]/[CH_4]_o) \qquad (4)$$

In Eq. 4, $[CH_4]_o$ and $\tau_o$ are the contemporary atmospheric $CH_4$ concentration and lifetime, and s is the $CH_4$-OH feedback factor, defined by $s = \partial \ln(\tau)/\partial \ln (CH_4)$. We take values of $\tau_o = 8.4$ years, $[CH_4]_o = 1,745$ ppbv and s = 0.28, from Prather et al. (2001) (pages 248 and 250). In our earlier study on the climate-wetland methane feedback (Gedney et al., 2019), we investigate the sensitivity to the methane lifetime and the feedback factor, in addition to an analysis of the main drivers on the wetland methane-climate feedback and the main sources of uncertainty. Gedney et al. (2019) conclude that the limited knowledge of contemporary global wetland emissions is a larger source of uncertainty than that from the projected climate spread of the 34 GCMs. We quantify this uncertainty in our experimental design by using two values of $Q_{10}$ (see Sect. 2.1).

In response to our dynamic interactive calculations of atmospheric $CH_4$ concentrations, we derive the related change in methane radiative forcing (RF). We use the formulation from Etminan et al. (2016), which accounts for the short-wave absorption by $CH_4$ and the overlap with $N_2O$. The atmospheric oxidation of methane (by the hydroxyl radical) leads to the production of tropospheric ozone and stratospheric water vapour. We calculate these indirect contributions of methane to the overall radiative forcing, following the approach for methane adopted in our previous work (Collins et al., 2018; Comyn-Platt et al., 2018; Gedney et al., 2019). Collins et al. (2018) represent the forcing contributions from $O_3$ and stratospheric water vapour as linear functions of the $CH_4$ mixing ratio, based on the analysis presented in IPCC AR5 (Myhre et al 2013). The indirect methane forcings amount to $2.36 \times 10^4 \pm 1.09 \times 10^{-4}$ W $m^{-2}$ per ppb $CH_4$ (i.e., $0.65 \pm 0.3$ times the $CH_4$ radiative efficiency). Hence we incorporate the indirect effects of these $CH_4$ emission changes by an approximation, multiplying the $CH_4$ radiative forcing by 1.65.

The discussion above seems to contradict the statement in line 243 that "We use future projections of atmospheric CH4 concentrations and LULUC from the IMAGE SSP2 projections (Doelman et al.,

2018) for both the methane and land-based mitigation strategies. ". From what I'm understanding from the paper it appears that the CH4 concentration is later adjusted for feedback effects - that should be made clearer.

*Response 11:* **We make this clarification.**

*Change(s) to the Paper: We amend lines 323-332.*

We use future projections of atmospheric $CH_4$ concentrations and LULUC (specifically, the areas assigned to agriculture and within that to BECCS) from the IMAGE SSP2 projections (Doelman et al., 2018) as input or prescribed data for both the methane and land-based mitigation strategies (Table 1). This ensures that all projections are consistent and based on the same set of IAM model and socio-economic pathway assumptions. The SSP2 socio-economic pathway is described as "middle of the road" (O'Neill et al., 2017), with social, economic, and technological trends largely following historical patterns observed over the past century. Global population growth is moderate and levels off in the second half of the century. The intensity of resource and energy use declines. We define the upper and lower limits of anthropogenic mitigation as the lowest (RCP1.9, denoted "IM-1.9") and highest ("baseline", denoted "IM-BL") total radiative forcing pathways, respectively, within the IMAGE SSP2 ensemble (Riahi et al., 2017). As described in Section 2.2.1, we modify the atmospheric concentrations of $CH_4$ in the IMOGEN-JULES modelling as the IMAGE scenarios assume constant natural and hence wetland methane emissions.

Also, if CH4 mitigation is from IMAGE this statement in the abstract "Globally, mitigation of anthropogenic CH4 emissions has large impacts on the anthropogenic fossil fuel emission budgets, potentially offsetting (i.e. allowing extra) carbon dioxide emissions of 188-212 GtC" is, therefore, not a result of this work but a conclusion just taken from the IMAGE results. This needs to be made clearer if this is the case.

*Response 12:* *We make clear that this is a result of this study. We are using the IMAGE SSP scenarios to provide realistic and consistent estimates of the methane changes in our CTL and $CH_4$ scenario. The mitigation of anthropogenic $CH_4$ emissions has a large impact on the anthropogenic fossil fuel $CO_2$ emission budgets for two reasons: (1) the reduction in the direct and indirect radiative forcing of methane in response to the lower emissions and hence atmospheric concentration of methane; and (2) the carbon-cycle changes through the increase in the uptake of $CO_2$ by the land and ocean from the higher atmospheric concentrations of $CO_2$ and a reduction in land vegetation ozone damage (lower atmospheric concentrations of methane lead to lower concentrations of tropospheric ozone).*

*Change(s) to the Paper: We amend the Abstract (lines 28-33):*

Globally, mitigation of anthropogenic $CH_4$ emissions has large impacts on the anthropogenic fossil fuel emission budgets, potentially offsetting (i.e. allowing extra) carbon dioxide emissions of 188-212 GtC. This is because of (a) the reduction in the direct and indirect radiative forcing of methane in response to the lower emissions and hence atmospheric concentration of methane; and (b) carbon-cycle changes leading to increased uptake by the land and ocean by $CO_2$-based fertilisation. Methane mitigation is beneficial everywhere, particularly for the major $CH_4$-emitting regions of India, USA and China.

*We also amend the Conclusions (lines 714-721): See also Response 26.*

Stabilising the climate primarily requires urgent action to mitigate $CO_2$ emissions. However, $CH_4$ mitigation has the potential to make the Paris targets more achievable by offsetting up to 188-212 GtC of anthropogenic $CO_2$ emissions, while still meeting the same global-warming targets. This offset is a direct consequence of the reduced radiative forcing by methane and of carbon cycle gains. These balances and related flexibilities have the potential to make the Paris targets more achievable. Our range of additional $CO_2$ emissions broadly applies to both the 1.5° and 2°C warming targets, as the mitigation potential of the $CH_4$ scenario is similar for the two temperature pathways considered.

Line 310 - Is this "the temperature pathway (1.5° versus 2°C warming) having a minor effect." the case even for the high Q10 sensitivity? Presumably this also depends on the temperature pattern used as well?

*Response 13: We refer to General Points 2 and 5, where we explain why the mitigation potential of the different mitigation scenarios are similar for the 1.5°C warming vs. 2°C warming targets. The high $Q_{10}$ methane runs are part of the 136-member ensembles and contribute to the uncertainty range. The solid line is the median of the ensemble and the width of the coloured band is the interquartile range of the ensemble. We will however amend the sentence to make this clear.*

*Change(s) to the Paper: We amend lines (395-397):*

As we use the same input datasets for the two warming targets,  the major control on the modelled atmospheric $CH_4$ concentrations is the $CH_4$ emission pathway followed, with the temperature pathway (1.5° versus 2°C warming) having a minor effect.

Section "2.4.1 Anthropogenic Fossil Fuel Emission Budget and Mitigation Potential"
It would be useful here to clarify that this is only the CO2 emissions budget (not total GHG emissions).

*Response 14: We amend the text to indicate that it is only the $CO_2$ emission budget.*

*Change(s) to the Paper: We amend lines 401-402:*

Following Comyn-Platt et al. (2018), we define the anthropogenic fossil fuel $CO_2$ emission budget (AFFEB) for scenario i as the change in carbon stores from present to the year 2100.

It would also be useful here to add some comment on what is driving each of the terms in the budget. (e.g., which models in Figure 1)

*Response 15: We add these comments. They are however generalisations as the changes are dependent on the scenario. We discuss the changes in the carbon stores in Section 3.1 (lines 487 and Figure 8).*

*Change(s) to the Paper: We amend the text (lines 405-411)*

where $C^{land}(t)$, $C^{ocean}(t)$ and $C^{atmos}(t)$ are the carbon stored in the land, ocean and atmosphere, respectively, in year $t$ and $BECCS(t_1:t_2)$ is the carbon sequestered via BECCS between the years $t_1$ and $t_2$. The atmospheric

carbon store does not include CH$_4$. This is a reasonable approximation, however, given the relative magnitudes of the atmospheric concentrations of CH$_4$ (~2 ppmv at the surface) and CO$_2$ (400 ppmv). where $C^{land}$ (t), $C^{ocean}$ (t) and $C^{atmos}$ (t) are the carbon stored in the land, ocean and atmosphere, respectively, in year $t$ and $BECCS(t_1:t_2)$ is the carbon sequestered via BECCS between the years $t_1$ and $t_2$. The atmospheric carbon store does not include CH$_4$. This is a reasonable approximation, however, given the relative magnitudes of the atmospheric concentrations of CH$_4$ (~2 ppmv at the surface) and CO$_2$ (400 ppmv).

Within the IMOGEN-JULES modelling framework, we use (a) the IMOGEN climate emulator to derive the changes in the ocean and atmosphere carbon stores, and (b) JULES for the changes in the land carbon store and carbon sequestered through BECCS. We discuss the changes in the carbon stores for the baseline and different mitigation scenarios in Sect. 3.1.

Line 378 I assume that the calcuation of H* depends on the assumed loss from farm to final storage? If so the assumed for farm to final storage loss should be stated. (line 445 seems to imply that minimal losses are assumed? 13%).

***Response 16:*** ***The Editor is correct; it does depend on assumed loss. It is implicit in the increase of $\varepsilon$ from 0.6 (40% loss) to 0.87 (13% loss). We make this explicit by repeating the sentence at lines 438-440.***

***Change(s) to the Paper:*** ***We amend the text (lines 469-473):***

In other words, BECCS$^*$ is equivalent to the carbon loss due to the land use change to grow the bioenergy crops. Our IMOGEN-JULES simulations assume a 40% carbon loss from farm to final storage, although other studies have assumed this to be as low as 13% (Harper et al., 2018). To assess the feasibility of meeting this break-even level of BECCS, we calculate the harvest (H$^*$) that would be needed if carbon losses are  minimised, i.e. by increasing ε from 0.6 to 0.87, and assuming in Eq. 13 that:

Line 391 This line is confusing "We find that there is increased uptake of atmospheric CO2 in the land-based mitigation scenarios, although there is a reduction in land carbon from the land-use changes in these scenarios. " Aren't these just are two sides of the same effect? (and see next comment as well)

***Response 17:*** ***This refers to Figure 8 (included for ease).***

***In all the scenarios apart from the BECCS scenario, there is an increase in the land carbon store (shown as positive changes for Coupled (Natural) and Coupled (Optimised) but as smaller negative changes for "CH4", "Natural" and "Optimised" scenarios). In the "BECCS" scenario, the land carbon change becomes more negative than in the "CTL" scenario, as bioenergy crops replace ecosystems with higher carbon content. We now clarify that "the reduction in land carbon from the land-use changes in these scenarios" refers to the scenarios involving BECCS.***

[Figure]

*Change(s) to the Paper: We include the Changes to the Paper with those for the next response (Response 18).*

Line 395 "which is greater than the land carbon lost through land- use changes. " Not clear what this means. How do land-use changes factor into this, given that LULUC is exogenous from IMAGE and therefore constant?

*Response 18: For LULUC, we use the time series of annual areas assigned to agriculture (crops and pasture) and within that the area allocated to BECCS in the "BECCS" land-based mitigation option. These areas are prescribed (i.e., exogenous). The distribution of the natural plant functional types (pfts) and the non-vegetated surface will evolve on the remaining land area in the grid cell. We use the dynamic vegetation model in JULES to calculate the evolution of this distribution. We amend Sections 3.1 and 2.3.2.*

*Change(s) to the Paper: We amend lines 480-495. The changes shown below also include those for Responses 5 and 17.*

**3.1 Global Perspective**

We calculate the anthropogenic fossil fuel emission budget to limit global warming to a particular temperature target as the sum of the changes in the carbon stores of the atmosphere, land (vegetation and soil) and ocean between 2015 and 2100 (Sect. 2.4.1, Eq. 5 and 6). We use a BECCS scale factor ($\kappa$) of unity. We present in Fig. 8 the median and  spread of the AFFEB (as box and whiskers) from the 136-member ensemble, and the individual GCM/ESM contributions to the AFFEBs from the four carbon pools shown (points), for each of the main scenarios modelled using the IMOGEN-JULES or derived in the post-processing optimisation step (see Table 1 for description of the scenarios).

 In all the scenarios apart from the BECCS scenario, there is an increase in the land carbon store (shown as positive changes for Coupled (Natural) and Coupled (Optimised) but as smaller negative changes for "$CH_4$", "Natural" and "Optimised" scenarios. In the "BECCS" scenario, the land carbon change becomes more negative than in the "CTL" scenario, as bioenergy crops replace ecosystems with higher carbon content. In the combined ('coupled') $CH_4$ and land-based mitigation scenarios, the reduction in the emissions and hence atmospheric concentrations of $CH_4$ allow increased atmospheric concentrations of $CO_2$ (Fig. 6). There is increased uptake

of carbon by the land, directly because of the increased atmospheric $CO_2$ concentration and indirectly through the reduction in $O_3$ damage. In the coupled "BECCS" scenario, this increased uptake of atmospheric $CO_2$ is again offset by  the land carbon lost through conversion of the land to bioenergy crops.

*We also amend Section 2.3.2 (lines 351-355)*

2.3.2 **Land-based mitigation: baseline, BECCS and Natural scenarios**

For our land-based mitigation scenarios, we take time series of the annual areas assigned to agriculture (crops and pasture) and within that, the area allocated to bioenergy crops, from the IM-BL and IM-1.9 scenarios (defined at the start of Sect. 2.3). We use the dynamic vegetation module in JULES to calculate the evolution of the natural plant functional types and the non-vegetated surface on the remaining land area in the grid cell (see Land use in Sect. 2.1).

The IM-BL LULUC scenario assumes (a) moderate land-use change regulation; (b) moderately effective land-based mitigation; (c) the current preference for animal products; (d) moderate improvement in livestock efficiencies; and (e) moderate improvement in crop yields (Table 1 in (Doelman et al., 2018)).

Line 446 - "yields of > 30 ton DM ha-1 yr-1 would be more difficult to realise". Not clear what this refers to given that earlier in that sentence H* is estimated to be up to 10-20 tons DM/ha/year..

*__Response 19:__  Where BECCS is replacing ecosystems that have higher carbon content, the productivity required will be potentially unrealistic compared to bioenergy yields currently observed. We amend the text to make this clear*

*__Change(s) to the Paper:__ We amend the text (lines 563-568):*

We calculate for each IMOGEN grid cell the increase in carbon removed via BECCS and the associated increase in bioenergy crop yields ($H^*$ in Sect.2.4.3) required for BECCS to be the preferred mitigation option (Fig. 10(d)), rather than natural land carbon uptake, and assuming minimal amounts of carbon are lost during the BECCS lifecycle (13% carbon loss). In many places, we find that the required yield increases from <10 to 10-20 ton DM ha$^{-1}$ yr$^{-1}$ are achievable, but required yields of > 30 ton DM ha$^{-1}$ yr$^{-1}$ would be more difficult to realise given the range of yields observed (Li et al., 2018).

Line 462 and table 3. The meaning of Table 3 and the associated text is not clear given that cumulative CH4 emissions have no physical meaning since CH4 does not accumulative in the atmosphere on timescales considered here. What is the "regional scale factor" presented in the table (the one line description in the table is not sufficient to understand what this means)? That scale factor also doesn't appear to be used otherwise in this work.

*__Response 20:__  From an atmospheric chemistry perspective, the Editor is correct. Here, although we use the phrase "cumulative emissions", we sum the change in methane emissions between 2020 and 2100. We use these summed emissions to derive a scale factor for each IMAGE region to attribute a fraction of the global atmospheric carbon store to that region. This is equivalent to using the average change in the annual methane emissions between 2020 and 2100. We then*

***use the scale factors in Table 3 to derive Figures 11 and 12 (and Supplementary Information, Figures SI.3 and SI.4).***

***Change(s) to the Paper: We amend the text (lines 587-594):***

For CH$_4$, we  use regional scale factors  to allocate  changes in the global atmospheric CH$_4$ concentration, and therefore the CH$_4$ mitigation potential, to each region, as shown in Table 3. To derive the regional scale factors, we separately sum  the projected anthropogenic CH$_4$ emissions between 2020  and 2100 ,   the IMAGE SSP2-Baseline and SSP2-1.9 scenarios (van Vuuren et al., 2017). We calculate the scale factor as the regional fraction of the global difference in the summed emissions (Table 3). These two CH$_4$ scenarios are consistent with the CH$_4$ concentration pathways considered in the CH$_4$  scenario simulations (Sect. 2.3). We use the scale factors to produce Fig. 11 and 12 (and Supplementary Information, Figures SI.3 and SI.4)".

***We also amend the caption to Table 3, which is now Table SI.3 in the Supplementary Information:***

Table 3 | IMAGE regions , the sum of the projected  anthropogenic CH$_4$ emissions between 2020  and 2100  for the SSP2-Baseline and SSP2-RCP1.9 scenarios and the differences between these summed emissions. The regional scale factor is calculated as the regional fraction of the global difference in the summed  emissions .

Figure 2b - what do the error bars represent? In figure 2a, there is a high/low Q10 bound defined, but Figure 2b has a central point and an error bar for the simulation results. What do each of these represent? Clarify in caption and in paper text.

***Response 21: The error bars denote the lower–upper estimates from the low and high Q$_{10}$ simulations and the symbols represent the mean value between these estimates. We add this clarification.***

[Figure]

***Change(s) to the Paper: We amend the text (lines 195-196):***
The range of uncertainty used in our study (JULES low Q$_{10}$ - JULES high Q$_{10}$) captures the range of uncertainty in the observations (In Fig. 2b, the error bars denote the lower and upper estimates from the low and high Q$_{10}$ simulations. The symbols represent the mean value between these estimates).

*We add to the end of the caption for Figure 2:*

The error bars denote the lower and upper estimates from the low and high $Q_{10}$ simulations. The  symbols represent the mean value between these estimates.

Figure 3. This figure is confusing. There are two temperature paths in 3a. But then there are two forcing paths in panel b that do not appear to be related to the temperature paths in panel a (what does CTRL and CH4 mitigation relate to the 1.5 and 2 degree pathways?). For a given set of climate parameters (climate sensitivity, etc.), wouldn't each forcing path uniquely relate to a temperature pathway?

Also, this seems to be a pivotal figure, but does not appear to be referenced in the text.

> ***Response 22:*** *Figure 3 shows key prescribed or input datasets for the IMOGEN-JULES modelling. We reference Figure 3a in Section 2.2.1 describing the IMOGEN modelling (line 293). Figure 3b is referenced in the same section (line 304). Except for the use of a common colour scheme, the panels are not related.*
>
> *Panel (a) shows the historic temperature change from 1850 to 2015 and the prescribed temperature pathways for 1.5° and 2°C of warming from 2015 to 2100.*
>
> *Panel (b) shows the input non-$CO_2$ GHG radiative forcing time series for the "CTL" and "CH4" mitigation scenario. These are adjusted during the IMOGEN-JULES modelling to take account of the natural methane feedback from wetlands and permafrost thaw.*
>
> ***Change(s) to the Paper:*** *We amend the caption to Figure 3.*
>
> Figure 3 | Time series of key datasets used in the study: (a) the historic temperature record (black) and the prescribed temperature profiles used to represent warming of 1.5°C (blue) and 2°C (orange); (b) the historic (black) and the projected non-$CO_2$ greenhouse gas radiative forcing (W m$^{-2}$) for the control (orange) and methane mitigation (blue) scenarios.
>
> ***We adjust the titles of the panels and use different colours in the plots.***

(a) Time series of the prescribed temperature pathways

[Figure]

(b) Input time series of the input non-$CO_2$ radiative forcing

[Figure]

Figure 8 - This is a central figure, but is unclear. Describe in the legend the difference between the colored bars and the pink/red open bars? It is not clear what the text "Note that the gain in the land carbon store for the CH4 scenario is shown as a reduction from -70.8 GtC in the control run to -1.4 GtC in the "CH4" mitigation option (median of ensemble)." means, particularly given that there is no land (green) bar in the CH4 mitigation bar in this figure.

> ***Response 23: We amend the legend to explain the coloured bars (the contribution of the different carbon stores to the AFFEB for each scenario) and the accompanying pink box and whiskers plot (AFFEB for the scenario, as the sum of the changes in the component carbon stores). The box and whiskers plot were placed to the right of the bars for greater clarity.***
>
> ***In response to comments from the Reviewers, we added a note to explain the gain in the land carbon store for the coupled runs, whereas there appears to be no land carbon component for the CH₄ mitigation scenario. The median value of the change in the land carbon store for the CH₄ mitigation scenario is close to zero. The individual ensemble members are visible as the green points in the second bar.***

**Change(s) to the Paper: We amend the caption to Figure 8.**

Figure 8 | The contribution to the allowable anthropogenic fossil fuel emission budget (AFFEBs, GtC) from the changes in the different carbon stores (atmosphere, ocean, land and BECCS) for the various control and mitigation scenarios, illustrated using the temperature pathway for 1.5°C of warming. The bars are the median of the component 136-member ensembles, with the individual members shown as points. The accompanying pink box and whiskers plots to the right of each set of bars are for the AFFEBs (as the sum of the changes in the component carbon stores). The box and whisker plots show the median, interquartile range, minimum and maximum derived of the resulting AFFEB ensemble. The optimised land based and coupled mitigation options selects the land use option, which maximises the AFFEB for each model grid cell. Note that  the land carbon store for the CH₄ scenario at -1.4 GtC (median of ensemble) is not visible, although the individual ensemble members can be seen as the green points..

**Similarly, we amend lines 501-505:**

In both Figs. 8 and 9, it should be noted that  the land carbon store for the "CH₄" mitigation option at -1.4 GtC (median of ensemble) is not visible in these figures. There has however been a net increase in the land carbon store in the "CH₄" scenario when compared to the land carbon store in the control scenario (

 -70.8 GtC  median of ensemble).

*And the caption to Figure 9:*

Figure 9 | Panels (a & c): The allowable anthropogenic fossil fuel emission budgets (AFFEBs; GtC) for the control (grey), $CH_4$ mitigation (purple), land-based mitigation (green), coupled methane and land-based mitigation (orange) and the linearly summed methane and land-based mitigation (brown), for 2 temperature pathways asymptoting at 1.5°C (left) and 2.0°C (right). (b & d) The mitigation potential (GtC) as the increase in AFFEB from the corresponding control run. The breakdown of each AFFEB and mitigation potential by the changes in the carbon stores is also shown: atmosphere (pale yellow), ocean (light blue), land (dark green) and BECCS (gold) is included alongside each bar. Note that  the land carbon store for the " $CH_4$" scenario at -1.4 GtC (median of ensemble) is not visible. There has however been a net increase in the land carbon store in this scenario when compared to the land carbon store in the control run ( -70.8 GtC, median of ensemble).

Figure 10 - I assume this "The spread of the functions " should be "The width of the lines "?

*Response 24:* ***The Editor is correct. It is the width of the lines.***

*Change(s) to the Paper: We amend the caption to Figure 10:*

The  width of the lines represent the interquartile range of the  136-member ensemble

One of the scientific contributions of this paper is the analysis of BECCS within this larger context, but there's not much detail in the results given. It would enhance the value of the paper to provide in the appendix some tables for the different scenarios of the regional land areas devoted to BECCS and the assumed biomass production rates. This would make the work more readily usable.

*Response 25:* ***We have undertaken further analysis and include in the Supplementary Information additional Tables (SI.3a-3d), which provide information on the areas, biomass production rates and carbon uptake by IMAGE region from the optimised land-based scenario for BECCS scale factors (κ) of 1, 2, 3 and 4.***

*Change(s) to the Paper: We amend the text to refer to these Tables (lines 568-571):*

We provide additional information in the Supplementary Information, Tables SI.3a-SI.3d on the modelled bioenergy yields and the yields required for bioenergy crops to be the preferred land-based mitigation option by IMAGE region. The tables also show that area of bioenergy crops and carbon sequestered by BECCS increases, as expected, with the BECCS scale factor (κ),

*We also amend the text (lines 611-617)*

The carbon uptake by BECCS increases as κ increases from 1 to 3 because there are more grid cells where 'BECCS' is the preferred mitigation option in the optimisation process, as evidenced by the increase in area

of bioenergy crops (Supplementary Information, Tables SI.3a and SI.3c). As κ only affects the 'BECCS' term (Sect. 2.4.3, Eq. 13), the increased carbon removed by BECCS is often accompanied by a decrease in the carbon uptake from the "natural" vegetation that it replaces. This can be seen more clearly in Fig. 12 (and Supplementary Information, Figure SI.3 for 2°C warming) and the Supplementary Information, Tables SI.3b and SI.3d.

*We include new Tables SI.4a-SI.4d in the Supplementary Information. Tables SI.4a and SI.4b for the 1.5°C temperature pathway are included here. Tables SI.4c and SI.4d are the equivalent tables for the 2°C temperature pathway.*

**Table SI.4a** |

| Region | Maximum area of BECCS (Mha) | BECCS Productivity | | Required Scale Factor | Area of BECCS (Mha) in optimised land-based scenario | | | |
|---|---|---|---|---|---|---|---|---|
| | | **Modelled** | **Required to match Natural** | | **BECCS scale factor κ = 1** | **BECCS scale factor κ = 2** | **BECCS scale factor κ = 3** | **BECCS scale factor κ = 4** |
| Canada | 65.9 | 2.99 (0.00-4.75) | 6.39 (0.00-20.16) | 2.93 (0.81-24.94) | 1.02 (0.74-1.24) | 6.69 (4.84-11.41) | 23.49 (14.23-30.93) | 31.93 (25.29-35.80) |
| USA | 39.0 | 5.40 (0.00-6.86) | 3.36 (0.00-10.88) | 1.16 (0.42-3.68) | 10.42 (5.06-15.73) | 23.71 (19.65-32.99) | 29.88 (25.55-37.69) | 35.98 (34.87-38.93) |
| Mexico | 7.1 | 6.86 (2.12-9.30) | 3.09 (0.98-5.84) | 0.73 (0.31-1.26) | 5.25 (5.25-5.81) | 7.28 (7.28-7.28) | 7.28 (7.28-7.28) | 7.28 (7.28-7.28) |
| Central America | 0.5 | 7.61 (0.07-9.60) | 2.64 (0.05-4.63) | 0.59 (0.03-0.92) | 0.56 (0.56-0.56) | 0.56 (0.56-0.56) | 0.56 (0.56-0.56) | 0.56 (0.56-0.56) |
| Brazil | 27.8 | 8.21 (0.00-10.89) | 3.46 (0.00-8.45) | 0.79 (0.34-2.32) | 14.64 (13.72-16.06) | 26.21 (26.21-31.42) | 33.27 (29.36-33.27) | 33.27 (31.89-33.27) |
| Rest of South America | 20.3 | 5.88 (0.01-9.98) | 3.53 (0.01-11.08) | 0.82 (0.40-6.29) | 12.14 (11.21-14.65) | 16.05 (16.03-16.96) | 18.54 (17.62-18.90) | 18.55 (17.62-18.90) |
| Northern Africa | - | - | - | - | - | - | - | - |
| Western Africa | 3.1 | 0.02 (0.00-4.72) | 0.00 (0.00-16.62) | 1.29 (0.33-8.01) | 1.96 (1.89-1.96) | 2.10 (2.10-2.10) | 2.10 (2.10-2.10) | 2.10 (2.10-2.10) |
| Eastern Africa | 33.9 | 4.67 (0.00-7.84) | 3.27 (0.00-35.69) | 2.43 (0.43-53.31) | 2.98 (2.58-3.41) | 5.05 (4.72-5.74) | 8.17 (7.78-8.37) | 8.37 (8.16-11.37) |
| South Africa | 1.0 | 0.00 (0.00-3.16) | 0.00 (0.00-2.71) | 1.03 (0.44-1.40) | 0.72 (0.72-1.02) | 1.02 (0.96-1.02) | 1.02 (1.01-1.02) | 1.02 (1.02-1.02) |
| Western Europe | 23.6 | 4.79 (0.00-5.71) | 3.80 (0.00-7.40) | 1.17 (0.74-2.50) | 4.84 (1.61-6.98) | 19.47 (19.47-19.47) | 22.52 (21.54-23.49) | 23.49 (23.49-23.49) |
| Rest of Southern Africa | 63.7 | 5.38 (0.00-8.83) | 4.42 (0.00-13.68) | 1.31 (0.57-10.90) | 13.17 (8.29-14.28) | 24.76 (24.76-25.25) | 24.76 (24.76-27.22) | 24.76 (24.76-27.59) |
| Central Europe | 19.3 | 5.05 (4.07-6.10) | 4.60 (1.96-11.93) | 1.32 (0.69-3.27) | 3.56 (1.21-6.49) | 9.60 (7.50-11.71) | 13.55 (13.55-13.55) | 17.14 (13.55-19.62) |
| Turkey | - | - | - | - | - | - | - | - |
| Ukraine Region | 11.4 | 4.78 (3.63-5.38) | 4.73 (2.87-41.12) | 1.35 (0.82-13.81) | 2.46 (0.00-5.20) | 5.20 (5.20-5.20) | 5.20 (5.20-5.20) | 8.06 (5.20-8.06) |
| Central Asia | 0.5 | 2.14 (0.00-4.58) | 0.00 (0.00-0.00) | - | 0.47 (0.47-0.47) | 0.47 (0.47-0.47) | 0.47 (0.47-0.47) | 0.47 (0.47-0.47) |
| Russia region | 146.1 | 3.39 (0.03-4.50) | 6.52 (0.02-44.97) | 3.34 (1.14-33.58) | 2.51 (0.48-4.59) | 24.36 (13.68-36.87) | 49.21 (36.39-69.33) | 75.28 (68.13-81.69) |
| Middle East | - | - | - | - | - | - | - | - |
| India | 6.0 | 6.92 (6.72-7.22) | 2.50 (1.38-9.07) | 0.53 (0.29-1.85) | 0.14 (0.14-6.08) | 6.08 (6.08-6.08) | 6.08 (6.08-6.08) | 6.08 (6.08-6.08) |
| Korea region | 4.3 | 6.09 (5.86-6.29) | 4.42 (3.41-5.54) | 1.07 (0.84-1.30) | 2.15 (0.00-4.30) | 4.30 (4.30-4.30) | 4.30 (4.30-4.30) | 4.30 (4.30-4.30) |
| China region | 58.1 | 5.08 (0.05-7.06) | 3.00 (0.04-7.64) | 0.89 (0.48-4.41) | 18.13 (13.20-22.02) | 35.11 (32.61-36.85) | 38.84 (37.77-39.79) | 46.08 (41.53-48.81) |
| Southeastern Asia | 24.5 | 7.19 (0.03-9.65) | 4.22 (0.02-16.68) | 0.83 (0.48-11.75) | 6.92 (6.92-6.92) | 7.30 (7.30-7.30) | 7.30 (7.30-7.30) | 7.30 (7.30-7.30) |
| Indonesia region | - | - | - | - | - | - | - | - |
| Japan | 2.7 | 5.59 (1.61-6.27) | 6.38 (3.33-23.76) | 1.56 (0.87-18.64) | 1.08 (0.00-2.46) | 2.46 (2.46-2.46) | 2.46 (2.46-2.46) | 2.46 (2.46-2.46) |
| Rest of South Asia | - | - | - | - | - | - | - | - |
| Oceania | 78.7 | 2.66 (0.00-7.52) | 3.26 (0.00-10.71) | 1.88 (0.79-4.67) | 18.36 (17.71-19.58) | 33.78 (30.20-39.76) | 64.41 (59.14-67.85) | 77.58 (74.43-81.72) |

**Table SI.4b** |

| Region | Carbon Uptake (GtC) | | | | | | | |
|---|---|---|---|---|---|---|---|---|
| | **BECCS** | **Land** | **BECCS** | **Land** | **BECCS** | **Land** | **BECCS** | **Land** |
| | **BECCS scale factor $\kappa = 1$** | **BECCS scale factor $\kappa = 1$** | **BECCS scale factor $\kappa = 2$** | **BECCS scale factor $\kappa = 2$** | **BECCS scale factor $\kappa = 3$** | **BECCS scale factor $\kappa = 3$** | **BECCS scale factor $\kappa = 4$** | **BECCS scale factor $\kappa = 4$** |
| Canada | 0.02 (0.01,0.02) | 0.15 (0.12,0.21) | 0.77 (0.59,1.41) | -0.49 (-0.87,-0.40) | 3.88 (2.58,4.61) | -2.63 (-2.89,-1.83) | 6.61 (5.88,6.95) | -3.91 (-4.00,-3.52) |
| USA | 0.70 (0.36,1.07) | 1.30 (0.58,1.88) | 2.76 (2.67,3.22) | 0.25 (-0.37,0.44) | 4.72 (4.53,5.44) | -0.30 (-0.68,0.01) | 7.39 (7.26,7.51) | -1.01 (-1.15,-0.86) |
| Mexico | 0.44 (0.41,0.48) | 1.00 (0.83,1.05) | 1.25 (1.21,1.32) | 0.76 (0.71,0.80) | 1.88 (1.82,1.98) | 0.76 (0.71,0.79) | 2.50 (2.42,2.64) | 0.76 (0.71,0.79) |
| Central America | 0.06 (0.05,0.06) | 0.24 (0.20,0.28) | 0.11 (0.11,0.12) | 0.24 (0.20,0.28) | 0.17 (0.16,0.18) | 0.24 (0.20,0.28) | 0.23 (0.22,0.24) | 0.24 (0.20,0.28) |
| Brazil | 1.72 (1.63,2.03) | 5.84 (5.28,6.58) | 5.21 (5.02,5.35) | 4.84 (4.51,5.33) | 8.05 (7.68,8.21) | 4.66 (4.28,5.13) | 10.80 (10.44,10.96) | 4.63 (4.28,5.06) |
| Rest of South America | 1.32 (1.20,1.47) | 4.20 (3.73,4.56) | 3.41 (3.29,3.55) | 3.69 (3.41,3.94) | 5.28 (5.10,5.48) | 3.57 (3.28,3.84) | 7.05 (6.81,7.32) | 3.57 (3.27,3.83) |
| Northern Africa | - | - | - | - | - | - | - | - |
| Western Africa | 0.02 (0.01,0.02) | 17.63 (16.30,19.04) | 0.06 (0.06,0.06) | 17.62 (16.28,19.02) | 0.09 (0.09,0.09) | 17.62 (16.28,19.02) | 0.12 (0.12,0.12) | 17.62 (16.28,19.02) |
| Eastern Africa | 0.12 (0.09,0.15) | 1.97 (1.88,2.14) | 0.50 (0.41,0.58) | 1.76 (1.61,1.97) | 1.11 (1.05,1.22) | 1.44 (1.38,1.63) | 1.48 (1.39,2.04) | 1.40 (1.19,1.63) |
| South Africa | 0.00 (0.00,0.01) | -0.13 (-0.14,-0.13) | 0.03 (0.02,0.03) | -0.15 (-0.15,-0.13) | 0.04 (0.04,0.04) | -0.15 (-0.15,-0.13) | 0.05 (0.05,0.06) | -0.15 (-0.15,-0.13) |
| Western Europe | 0.39 (0.10,0.59) | -0.27 (-0.43,-0.04) | 3.13 (3.08,3.19) | -1.83 (-2.25,-1.61) | 4.97 (4.90,5.05) | -2.15 (-2.46,-1.84) | 6.69 (6.57,6.80) | -2.16 (-2.57,-1.84) |
| Rest of Southern Africa | 1.49 (0.96,1.59) | 4.87 (4.68,5.60) | 4.84 (4.74,4.93) | 3.57 (3.47,3.71) | 7.27 (7.11,7.44) | 3.57 (3.46,3.70) | 9.70 (9.48,9.92) | 3.56 (3.43,3.70) |
| Central Europe | 0.24 (0.07,0.46) | -0.33 (-0.44,-0.22) | 1.33 (1.09,1.61) | -0.90 (-0.96,-0.88) | 2.62 (2.58,2.66) | -1.66 (-1.76,-1.07) | 3.87 (3.52,4.16) | -1.71 (-1.76,-1.62) |
| Turkey | - | - | - | - | - | - | - | - |
| Ukraine Region | 0.14 (0.00,0.31) | -0.13 (-0.25,-0.03) | 0.64 (0.63,0.66) | -0.36 (-0.47,-0.25) | 0.96 (0.95,0.99) | -0.44 (-0.47,-0.25) | 1.57 (1.32,1.64) | -0.54 (-0.57,-0.47) |
| Central Asia | 0.00 (0.00,0.00) | 0.02 (0.01,0.03) | 0.00 (0.00,0.00) | 0.02 (0.01,0.03) | 0.01 (0.01,0.01) | 0.02 (0.01,0.03) | 0.01 (0.01,0.01) | 0.02 (0.01,0.03) |
| Russia region | 0.15 (0.03,0.28) | 0.84 (0.64,1.02) | 3.15 (1.95,4.32) | -1.43 (-2.08,-0.52) | 7.85 (6.43,9.22) | -3.81 (-4.38,-3.36) | 13.05 (12.41,13.60) | -5.84 (-6.55,-5.49) |
| Middle East | - | - | - | - | - | - | - | - |
| India | 0.00 (0.00,0.12) | 0.18 (0.15,0.20) | 0.25 (0.24,0.26) | 0.02 (-0.01,0.12) | 0.38 (0.37,0.39) | 0.01 (-0.02,0.08) | 0.51 (0.49,0.52) | 0.01 (-0.02,0.08) |
| Korea region | 0.13 (0.00,0.26) | -0.11 (-0.23,-0.01) | 0.54 (0.53,0.55) | -0.29 (-0.36,-0.23) | 0.81 (0.79,0.83) | -0.29 (-0.36,-0.23) | 1.09 (1.05,1.11) | -0.29 (-0.36,-0.23) |
| China region | 1.36 (1.03,1.61) | -0.23 (-0.43,-0.01) | 5.42 (5.27,5.56) | -2.16 (-2.47,-2.07) | 8.54 (8.42,8.65) | -2.63 (-3.06,-2.19) | 12.10(11.61,12.49) | -3.18 (-3.23,-3.12) |
| Southeastern Asia | 0.88 (0.86,0.90) | 1.52 (1.33,1.65) | 1.82 (1.77,1.85) | 1.49 (1.31,1.61) | 2.72 (2.66,2.78) | 1.49 (1.31,1.61) | 3.63 (3.55,3.71) | 1.49 (1.31,1.61) |
| Indonesia region | - | - | - | - | - | - | - | - |
| Japan | 0.10 (0.00,0.23) | -0.04 (-0.16,0.05) | 0.47 (0.45,0.48) | -0.24 (-0.32,-0.16) | 0.70 (0.68,0.72) | -0.24 (-0.32,-0.16) | 0.93 (0.91,0.96) | -0.24 (-0.32,-0.16) |
| Rest of South Asia | - | - | - | - | - | - | - | - |
| Oceania | 0.20 (0.19,0.27) | 1.71 (1.60,1.88) | 2.37 (1.62,2.94) | 0.47 (-0.14,1.15) | 6.17 (5.72,6.90) | -1.54 (-1.88,-1.38) | 9.18 (8.51,10.26) | -2.52 (-2.81,-2.17) |
| Global | 9.31 (7.19,11.93) | 41.82 (36.12,46.59) | 37.83 (35.74,41.73) | 27.50 (23.51,31.04) | 67.96 (65.96,71.25) | 18.82 (16.30,21.23) | 98.62(95.98,100.77) | 12.61 (11.16,13.67) |

The title and abstract of the paper focus on the combination of CH4 and land-based mitigation, but the conclusion section doesn't integrate these nor give context from the literature. The abstract indicates that methane mitigation offsets 88-212 GtC, while land-based mitigation offsets 51-100 GtC, which is much smaller. The methane mitigation analysis is exogenous to this paper, but is this range robust across models that do include this? Does the methane range include the offsetting effects from wetland emissions?

> ***Response 26: We refer to Responses 1 and 13, where we explain why the methane mitigation analysis is not exogenous. We add further text to the Conclusions to address the integration of the mitigation scenarios and comparison with relevant literature.***
>
> ***The analysis does include the offsetting effect of wetland methane emissions. We state in the Abstract (lines 25-26) "The analysis includes the effects of the methane and carbon-climate feedbacks from wetlands and permafrost thaw, which we have shown previously to be significant constraints on the AFFEBs". We also state in the Conclusion (line 533-534) "We utilise the detailed JULES land-surface model, which includes the temperature sensitivity of methanogenesis (Comyn-Platt et al., 2018) and the effect of CH$_4$ emissions on land carbon storage via ozone impacts on vegetation …".***

> ***Change(s) to the Paper: We amend the sentence in the Conclusion (line 663-665) to make explicit that we are referring to wetlands (and permafrost thaw).***

We utilise the detailed JULES land-surface model, which includes the temperature sensitivity of  methane production from wetlands and permafrost thaw (Comyn-Platt et al., 2018) and the effect of CH$_4$ emissions on land carbon storage via ozone impacts on vegetation …

***We amend the text to address the integration (lines 669-683):***

This analysis quantifies the regional differences in potential CH$_4$ and/or land-based strategies to aid mitigation of climate change.  We present our findings  within a full probabilistic framework, capturing uncertainty in climate projections across the CMIP5 ensemble, as well as process uncertainties associated with the strength of natural CH$_4$ climate feedbacks from wetlands and ozone-induced vegetation damage. Globally, mitigation of anthropogenic CH$_4$ emissions and the optimised land-based mitigation can potentially offset (i.e. allow extra) fossil fuel carbon dioxide emissions of 188-212 GtC and 51-100 GtC, respectively. These bounds are almost independent of the eventual global-warming target, or the climate sensitivity of the climate models emulated. As shown in Sect.3.1, the CH$_4$ and land-based mitigation strategies show little interaction and their potential can be summed to give a comparable result to the corresponding coupled simulation. This decoupling is despite the CH$_4$ emissions from the agricultural sector being influenced by land use choices. We can therefore treat the two mitigation strategies as independent, and sum their individual potentials. Such linearity enables simpler and more direct comparisons between the carbon budgets of methane and land-based mitigation strategies. Some caveats remain however. Land surface models still require refinement, alongside improved characterisation of the assumptions inherent in the socio-economic pathways and IAM modelling. Further, we do not allow for the reduced emissions from fossil fuel combustion due to the bioenergy crop being grown (or the converse when bioenergy crops are replaced in the Natural model run), as this would require energy sector modelling that is beyond the scope of this study.

*We also amend the text (lines 716-730):*

Stabilising the climate primarily requires urgent action to mitigate $CO_2$ emissions. However, $CH_4$ mitigation may offset up to 188-212 GtC of anthropogenic $CO_2$ emissions, while still meeting the same global-warming targets. This offset is a direct consequence of the reduced radiative forcing by methane and of carbon cycle gains. These balances and related flexibilities have the potential to make the Paris targets more achievable. Our range of additional $CO_2$ emissions broadly applies to both the 1.5° and 2°C warming targets as the mitigation potential of the $CH_4$ scenario is similar for the two temperature pathways considered. Although there are differences in the precise methane emission scenarios used, our mitigation potential is similar to that given in Collins et al. (2018). That paper presents values of 155 or 235 GtC for offsetting $CH_4$ mitigation from a high to a medium or from a high to a low emission scenario, respectively. Our value, and those of Collins et al. (2018), can be compared to the increase of 130 GtC in the carbon budget between a no and a stringent $CH_4$ emission mitigation scenario estimated by Rogelj et al. (2015). More recently, Harmsen et al. (2020) have also investigated the mitigation potential of methane, although their results are expressed in terms of changes in radiative forcing and temperature, rather than carbon budgets. An advantage of our analysis remains the inclusion of climate response to altered radiative forcing, enabling understanding in terms of actual $CO_2$ emissions. We conclude that $CH_4$ mitigation would be effective globally as a contribution to constraining global warming, and especially so for the major $CH_4$-emitting regions of India, USA and China.

For the land-based mitigation, it would be useful if the conclusion section could provide some context for these results. What factors impact this range (e.g., under what conditions is this low, and under what conditions is this high?). How does this compare with what is assumed in the literature from IAM scenarios? Seems this might be much smaller than commonly assumed, why is that?

*Response 27: We provide text comparing our results to those from IAM and identify possible reasons why our results might be lower, drawing on our previous work (cited paper by Harper et al, 2018).*

*Change(s) to the Paper: We amend the text (lines 684-710):*

For the "Natural" land-based scenario (see Table 1), we find a mitigation potential of 50-55 GtC (183-201 $GtCO_2$). The land-based mitigation estimates vary over wide ranges, partly related to different assumptions on land use and carbon pools. Our results are within the wide range of the overall deployment of $CO_2$ removal by Agriculture, Forestry and Other Land Use (including afforestation and reforestation) to 2100 of 200 [0-550] $GtCO_2$ (Page 2.40 in IPCC, 2018) and of estimates of the cumulative potential to 2100 from 80 to 260 $GtCO_2$ (Table 2) in Minx et al. (2018). In the "BECCS" scenario, we obtain a geological carbon storage via BECCS (27±1 GtC median, interquartile range) similar to that (30±1 GtC) derived by Harper et al. (2018), for the same land use scenario (IM-1.9). Our result is lower as we include the natural methane feedbacks from wetlands and permafrost thaw. Inclusion of this better process description leads to ~10% reduction in carbon budgets (Comyn-Platt et al., 2018). These estimates for the geological carbon storage via BECCS are much lower than the corresponding value derived by the IMAGE IAM (130 GtC). Harper et al. discuss this difference, identifying a number of reasons for the lower value: the use of initial

above ground biomass harvested in boreal forests for BECCS, the replacement of fossil-fuel based emissions in the energy system, as well as specific assumptions about crop yields, conversion efficiency, use of residues, the proportion of bioenergy crops used with CCS. Estimates of the BECCS contribution in the literature vary over a wide range (from 178 to >1000 $GtCO_2$, according to Minx et al., 2018), but in recent studies these result are typically revised downwards taking into account among others sustainability constraints (e.g. Fuss et al. 2018 suggests a potential of 0.5-5 $GtCO_2$ per year in 2050).

We investigate the efficacy of our "BECCS" scenario by increasing the productivity of BECCS (using a scale factor κ). From comparison with observed bioenergy crop yields, we argue that the scale factor could be between 1 and 3. We highlight how using this range of κ provides characterisation of an additional source of uncertainty on the land-based mitigation potential, and is therefore a key feature of our manuscript. In our optimised land-based mitigation scenario, which maximises the land carbon uptake (Sect 2.4.2, Eq. 10), the increased carbon removed by BECCS is often accompanied by a decrease in the carbon uptake from the "natural" vegetation that it replaces (as discussed in Sect. 3.3 and shown in Figure 12). This concern is equivalent to the statement in Harper et al. that the "use of BECCS in regions where bioenergy crops replace ecosystems with high carbon contents could easily result in negative carbon balance". Hence the particularly novel feature of our paper is that our optimal approach accounts explicitly for that trade-off, only suggesting BECCS where there is a net gain.

Are both of these ranges across 1.5 and 2.0 targets? Does a 1.5 or 2.0 degree target shift either of these ranges significantly?

*__Response 28:__ We refer to General Point 5 (1.5°C warming vs. 2°C warming targets). The ranges are similar for the two warming targets. For that reason, we presented a single range across the two warming targets.*

*__Change(s) to the Paper:__ We amend the Abstract (28-36):*

Globally, mitigation of anthropogenic $CH_4$ emissions has large impacts on the anthropogenic fossil fuel emission budgets, potentially offsetting (i.e. allowing extra) carbon dioxide emissions of 188-212 GtC. Methane mitigation is beneficial everywhere, particularly for the major $CH_4$-emitting regions of India, USA and China. Land-based mitigation has the potential to offset 51-100 GtC globally, the large range reflecting assumptions and uncertainties associated with BECCS. The ranges for $CH_4$ reduction and BECCS implementation are valid for both the 1.5° and 2°C warming targets. That is the mitigation potential of the $CH_4$ and of the land-based scenarios is similar whether society aims for one or other of the final stabilised warming levels. Further, both the effectiveness and the preferred land-management strategy (i.e., AR or BECCS) have strong regional dependencies.

*__We amend the Conclusions (720-721):__*

Our range of additional $CO_2$ emissions broadly applies to both the 1.5° and 2°C warming targets, as the mitigation potential of the $CH_4$ scenario is similar for the two temperature pathways considered.

I noticed that the upper end of land mitigation given in the abstract is lower than the high end of the line provided in Figure 10. (Presumably you haven't allowed the BECCS scale factor to be as high as 6 in the main results?).

***Response 29: The Editor is correct. This also links to the earlier comment about bioenergy productivity (Response 27). The lower end of the range is based on results using a BECCS scale factor of 1 and the upper end of the range is based on a BECCS scale factor of 3. We make this clear in the Conclusions. We already state in the Abstract that the large range of the land-based mitigation reflects "assumptions and uncertainties associated with BECCS" (line 30).***

***Change(s) to the Paper: We include the following text with those to Response 27 (see above).***

We investigate the efficacy of our "BECCS" scenario by increasing the productivity of BECCS (using a scale factor κ). From comparison with observed bioenergy crop yields, we argue that the scale factor could be between 1 and 3. We use this range of κ as an additional source of uncertainty on the land-based mitigation potential.

Note that the journal's data policy requests that data and code associated with articles to be deposited in public data repositories. See: https://www.earth-system-dynamics.net/policies/data_policy.html. The paper does not currently follow best practices here. Please consider depositing key input and output data in a public data repository.

***Response 30: We have created a github project, where we make available (a) the IMOGEN-JULES source code (as a zipped tarball); and (b) the post-processing scripts, which we use in the analysis and with which we generate the Figures. We will lodge key model outputs in a publically accessible repository.***

***Change(s) to the Paper: We amend the Code and Data Availability section (lines 559-565):***

Code and Data Availability

The IMOGEN-JULES source code used in this work is available from the JULES code repository (https://code.metoffice.gov.uk/trac/jules/browser/main/branches/dev/annaharper/r7971_vn4.8_1P5_DEGREES_CCS, at JULES revision 14477, user account required). The rose suites used for the specific scenario and factorial runs are: u-as624, u-at010, u-at011, u-at013, u-av005, u-av007, u-av008, u-av009, u-ax327, u-ax332, u-ax455, u-ax456, u-ax521, u-ax523, u-ax524, u-ax525, u-bh009, u-bh023, u-bh046, u-bh081, u-bh084, u-bh098, u-bh103 and u-bh105. These can be found at https://code.metoffice.gov.uk/trac/roses-u/ (user account required).

The IMOGEN-JULES source code is also available as a zipped tarball from https://github.com/GarryHayman/Regional_Mitigation_Paper, as are the python scripts used for post-processing. Relevant outputs from the IMOGEN-JULES runs will be made available through a publically accessible data repository (tbc) .

Requests for information about the All code, data and or parameterisations are available on requestcan be made to the corresponding author.

---

## Author Response (AR3)

"**Regional variation in the effectiveness of methane-based
and land-based climate mitigation options**"
Hayman et al. (ESD-2020-24)

**Editor Decision:**
Publish as is (05 Mar 2021) by Steven Smith

Comments to the Author:
The revisions to the paper are well done and have significantly enhanced the readability and depth of analysis. I have only a few minor editorial comments that could probably be addressed in page proofs.

Line 114: "reflecting the different thermal sensitivities of existing climate models"
I assume the authors mean climate sensitivity? (thermal sensitivities is not a standard term). Suggest revision to make clear to reader.

Line 116: Some words appear to be missing here
"activity, and for scenarios taken from the IMAGE integrated assessment model"

Line 117: "for CH4, we use an understanding of its atmospheric lifetime to translate methane emissions into their atmospheric concentrations"

"use an understanding" is a bit vague, but it would be sufficient to revise with a reference to section 2.2.1 that contains the new Equation 3 and text there that describes the methane calculations.)

**Author Response**:

We submit the version accepted for publication, with the following changes:

1.  We amend lines 114-117 to address the Editor's comment above.

    "Hence our Radiative Forcing (RF) trajectories have uncertainty bounds, reflecting the different  climate sensitivities of existing climate models.

    For each radiative forcing pathway, we subtract the individual RF components for non-$CO_2$ and non-$CH_4$ radiatively-active gases that are perturbed by human activity,  using baseline and mitigation scenarios taken from the IMAGE integrated assessment model. Then, for $CH_4$, we  represent its atmospheric chemistry by a single atmospheric lifetime to translate the methane emissions into  atmospheric concentrations"

2.  We correct the text where 'κ' or '°' is inadvertently shown as '□'.

3.  We amend the data availability statement to give locations and doi's for:
    *   The IMOGEN-JULES source code and python processing scripts
    *   Datasets and relevant output from the study.
    *   The IMOGEN pattern scaling datasets and parameters for the IMOGEN Energy Balance model, for each of the 34 CMIP5 models emulated. We include the reference (Comyn-Platt et al., 2018) and doi for these data.

4.  We abbreviate journal names in the list of references, using the accepted abbreviations. We add pages numbers to those references, where these were either missing or incomplete.